# The TNFR Wengen regulates the FGF pathway by an unconventional mechanism

Annalisa Letizia [1], Maria Lluisa Espinàs [1], Panagiotis Giannios [1,2] & Marta Llimargas [1] ✉

Unveiling the molecular mechanisms of receptor activation has led to much understanding of development as well as the identification of important drug targets. We use the *Drosophila* tracheal system to study the activity of two families of widely used and conserved receptors, the TNFRs and the RTK-FGFRs. Breathless, an FGFR, controls the program of differentiation of the tracheal terminal cells in response to ligand activation. Here we identify a role for Wengen, a TNFR, in repressing the terminal cell program by regulating the MAPK pathway downstream of Breathless. We find that Wengen acts independently of both its canonical ligand and downstream pathway genes. Wengen does not stably localise at the membrane and is instead internalised—a trafficking that seems essential for activity. We show that Breathless and Wengen colocalise in intracellular vesicles and form a complex. Furthermore, Wengen regulates Breathless accumulation, possibly regulating Breathless trafficking and degradation. We propose that, in the tracheal context, Wengen interacts with Breathless to regulate its activity, and suggest that such unconventional mechanism, involving binding by TNFRs to unrelated proteins, may be a general strategy of TNFRs.

Receptors receive information from the environment (e.g. as signalling molecules or mechanical forces) and transmit it to the cell to elicit changes. Their activity regulates many kinds of biological events during development and homoeostasis, ranging from migration or cell differentiation to immunity or regulation of metabolism. Receptor activation needs to be exquisitely controlled to provide an outcome only when and where it is required. Thus, misregulation of receptor activity (excess or defect) frequently leads to malignant transformation, diseases or malformations.

Fibroblast Growth Factor Receptors (FGFRs), which belong to the Receptor Tyrosine Kinase (RTK) superfamily, are involved in diverse processes, ranging from organ morphogenesis to injury repair and regeneration. Consequently, FGFR malfunction leads to severe diseases, such as chronic kidney disease, dwarfism syndromes or obesity and it is also involved in cancer, especially in breast, lung, prostate and ovarian cancers[1–3].

FGFRs are activated by Fibroblast Growth Factors (FGF) ligands. Ligand binding promotes receptor dimerisation and trans-phosphorylation, initiating the activation of downstream cascades, namely AKT, PLCγ, STAT and ERK-MAPK, by phosphorylation[4]. Correlating with the importance of FGFRs in health and disease, their activity is finely regulated by a variety of mechanisms, including synthesis and secretion, stabilisation of FGF/FGFR, interactions with cofactors/adaptors, subcellular localisation, endocytosis and intracellular trafficking[5].

Tumor Necrosis Factor Receptors (TNFRs) also play key roles in development and homoeostasis, and are particularly involved in the regulation of the immune system, inflammation and cell death. Misregulation of TNFR activity also leads to several serious pathologies such as autoinflammatory diseases and cancer[6–8]. TNFRs are activated by Tumor Necrosis Factor (TNF) ligands, resulting in trimeric TNFR-TNF complexes. Through oligomerisation, the TNFR-TNF complexes

[1]Department of Cells and Tissues. Institut de Biologia Molecular de Barcelona, IBMB-CSIC. Parc Científic de Barcelona, Baldiri Reixac, 10-12, 08028 Barcelona, Spain. [2]Institute for Research in Biomedicine (IRB Barcelona), The Barcelona Institute of Science and Technology (BIST), Baldiri Reixac 10, 08028 Barcelona, Spain. ✉e-mail: mlcbmc@ibmb.csic.es

recruit adaptor proteins like TRADD or TRAFs that initiate a cascade to regulate downstream signalling by JNK, NF-kB and Complex-II mediated apoptosis[6,9–11].

Because of the multifunctional nature of most receptor families in health and disease, it is urgent to understand the cross-talk between the different receptors, their signalling pathways, and their downstream outputs in "in vivo" conditions. *Drosophila* gives us many genetic tools in the approach to such complex problems. Here we use the embryonic tracheal system of *Drosophila* as a model to investigate the roles and interactions of two different types of receptors, the FGFR-Breathless (Btl), and the TNFR-Wengen (Wgn). It was previously known that the FGFR-Btl is central to tracheal development, regulating different steps including migration and cell differentiation (for reviews see[12,13]). In contrast to the extensive knowledge of the roles of FGFR-Btl, there is no information about possible roles of TNFR-Wgn in tracheal formation. TNFR-Wgn was identified several years ago as a receptor for the unique TNF in *Drosophila* Eiger (Egr)[14–17]; however, its function became unclear and controversial[18]. Here we describe that both FGFR-Btl and TNFR-Wgn are required for the specification of the tracheal terminal cells, which extend fine terminal branches that are responsible for gas exchange with the target tissue once the tracheal network becomes physiologically functional[12,19–21]. Terminal cell differentiation depends on the activation of FGFR-Btl by its ligand FGF-Branchless (Bnl), in such a way that in the absence of FGFR-Btl activation no terminal cells form, while in conditions of FGFR-Btl overactivation extra terminal cells are observed[21–24]. We find that in *TNFR-wgn* loss of function conditions extra terminal cells are detected, while in gain of function conditions less terminal cells form. Due to the similarity of the phenotypes in terminal cell differentiation produced by *TNFR-wgn* and *FGFR-btl* manipulations, we investigated their interactions and crosstalks during the process. We show that *TNFR-wgn* and *FGFR-btl* are expressed in the same cells and regulate the same process. We find that TNFR-Wgn works in an unconventional manner to regulate the activity of FGFR-Btl, adding another layer of regulation of this critical receptor.

## Results

### TNFR-Wgn restricts tracheal terminal cell differentiation

We identified the *TNFR-wgn* in the course of a genetic screen for new factors regulating tracheal development.

We used a null allele of *TNFR-wgn* (*TNFR-wgn^{KO,18}*) to investigate *TNFR-wgn* tracheal requirements during embryogenesis. The early steps of tracheal formation and branching were not affected (Supplementary Fig. 1a–g), however, we detected adventitious terminal branches throughout the whole tracheal tree (i.e. in dorsal branches (DBs), lateral trunk (LT), ganglionic branches (GBs) and visceral branches (VB)) (Fig. 1a,b, and Supplementary Fig. 1a–g).

To determine the origin of these terminal branches we stained the embryos with Drosophila Serum Response Factor (DSRF), a marker for terminal cell differentiation[19,20]. We observed excess of DSRF positive cells that generated these adventitious terminal branches (Fig. 1c–f and Supplementary Fig. 1a–g). The *TNFR-wgn^{KO}* phenotype was fully penetrant as all embryos displayed extra terminal cells. We analysed the DBs, which in normal conditions contain 1 terminal cell at the tip (Fig. 1e), to investigate the phenotype of *TNFR-wgn* depletion. We found around 90% of DBs containing more than one terminal cell, with a high proportion of them containing 3 or more (Fig. 1i). In addition, we found many cases in which terminal cells also appeared in the stalk of the branch (Fig. 1b, f, i and Supplementary Fig. 1e, f). Quantification of terminal cells in other branches, like GBs, also indicated a significant increase with respect to the control (2,1% of GBs contained more than one terminal cell in the control, $n = 475$ branches from 32 embryos, 57% of GBs contained more than one terminal cell in *TNFR-wgn^{KO}* mutants, $n = 415$ branches from 31 embryos), indicating a general effect in terminal cell differentiation.

Downregulation of *TNFR-wgn* in the tracheal system using RNAi reproduced the same phenotype of null mutants (Fig. 1g, i), indicating an autonomous effect and that TNFR-Wgn is required in the tracheal cells to regulate the number of terminal cells.

The tracheal overexpression of a wild type form of *TNFR-wgn* (*TNFR-wgn-Flag*)[25] produced a highly penetrant phenotype (Fig. 1j) that was the opposite to that observed in *TNFR-wgn^{KO}* and *TNFR-wgnRNAi*: a loss of DSRF expressing cells (and terminal branches) throughout the tracheal system (Fig. 1h).

*TNFR-wgn* manipulations specifically affected the differentiation of terminal cells, and neither the absence nor the overexpression of *TNFR-wgn* affected the differentiation of other tracheal tip cell types such as the fusion cells (Fig. 1k–n). We found that the extra terminal cells in *TNFR-wgn* loss of function conditions did not derive from extra cell proliferation in the tracheal system (Supplementary Fig. 1h). Cell counts in DBs indicated a comparable number of cells in conditions of *TNFR-wgn* downregulation and control conditions (Supplementary Fig. 1i). These results indicated that in *TNFR-wgn* loss of function conditions presumptive stalk cells acquire the terminal cell identity (Supplementary Fig. 1j).

Altogether these results show that *TNFR-wgn* activity is specifically required to limit the number of terminal cells that generate the terminal branches.

### TNFR-Wgn accumulates in intracellular vesicles in trachea

*TNFR-wgn* is expressed in different tissues, including the tracheal system (BDGP).

We analysed the accumulation and localisation of TNFR-Wgn protein in tracheal cells using a validated antibody[18,26]. In contrast to our expectations for a membrane receptor, we could not detect TNFR-Wgn in the membrane of tracheal cells. Instead, we detected Wgn in intracellular punctae (Fig. 2a,b), as previously described in imaginal discs[26].

We reasoned that maybe the endogenous levels of TNFR-Wgn were not high enough to be detected at the membrane by the antibody. For this reason, we overexpressed TNFR-Wgn in the tracheal cells. We detected increased levels of TNFR-Wgn in these conditions, but again the protein localised mostly in intracellular punctae (Fig. 2c). However, we also observed, on occasions, accumulation of TNFR-Wgn in the apical membrane upon overexpression (Fig. 2d). This result suggested that TNFR-Wgn has the ability to localise to the membrane, at least when we saturate the system.

To identify the nature of TNFR-Wgn punctae, we co-stained with different intracellular trafficking markers. We found colocalisation with markers of late endosomes and multivesicular bodies (i.e. Rab7 and Hrs, Fig. 2e,f), with lysosomal markers (Arl8, Fig. 2h) and with the fast recycling pathway component Rab4 (Fig. 2g). Thus, the results suggested that TNFR-Wgn traffics through the endocytic pathway and can be degraded or recycled back to the membrane.

Because we found TNFR-Wgn in endocytic compartments but could not detect TNFR-Wgn stabilised at the membrane, we speculated that TNFR-Wgn could be constitutively internalised. To test whether TNFR-Wgn is internalised and traffics through the endocytic pathway, we compromised endocytic uptake by downregulating Rab5 activity. In these conditions, we now found a clear accumulation of TNFR-Wgn (endogenous and overexpressed) at the apical, basal, and lateral membrane (Fig. 2i, j). This change of localisation of TNFR-Wgn upon perturbing endocytic internalisation indicates that this transmembrane receptor is normally internalised. Strikingly, we also found that when internalisation is compromised, TNFR-Wgn overexpression can no longer prevent terminal cell differentiation, as assessed by the presence of DSRF expressing cells in all DBs (Fig. 2j). This result suggests that TNFR-Wgn must be internalised to exert its activity.

Altogether these results show that TNFR-Wgn reaches the membrane but is normally internalised and accumulates in endocytic vesicles, preventing it to stably localise at the membrane.

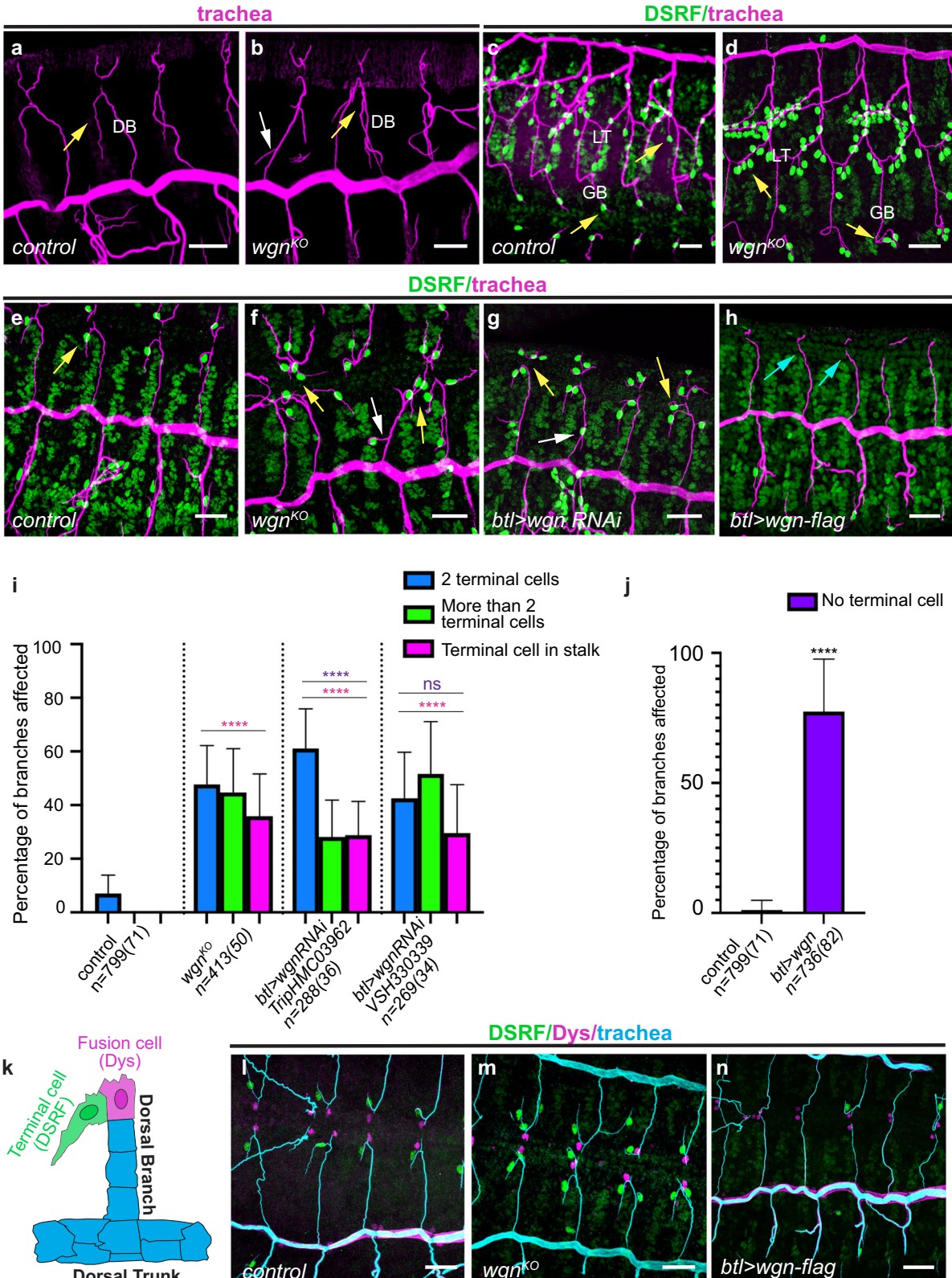

### TNFR-Wgn acts independently of its canonical signalling pathway

TNFR-Wgn was proposed to transduce the Egr signal through the JNK pathway[27]. Thus, we investigated the contribution of this pathway to *TNFR-wgn* tracheal requirements (Fig. 3a).

We downregulated the pathway using different tools (*bsk^DN*, *Tak1^DN*, *hepRNAi*, *Traf2RNAi*, or *UASpuc*). We detected a proportion of DBs with 1 extra-terminal cell, producing a mild phenotype when compared to the control (Fig. 3b, c). However, downregulation of the JNK pathway did not reproduce the *TNFR-wgn* loss of function phenotype: the proportion of DBs with 1 extra terminal cell was low, we never detected more than 2 terminal cells per DB, presence of terminal cells in the DB stalk, or a significant excess of terminal cells in GBs. Thus, the JNK downregulation phenotype did not correlate quantitatively and qualitatively with that of *TNFR-wgn* loss of function.

**Fig. 1 | Functional requirements of *TNFR-wgn* and *TNFR-egr* during tracheal formation. a**–**h** Dorso-lateral (**a**, **b**; **e**–**h**) or ventro-lateral (**c**, **d**) views of stage 15/16 embryos stained with CBP to visualise the tracheal tubes (magenta) and with DSRF to visualise the terminal cells (green). One single terminal cell and terminal branch in dorsal (DB) and ganglionic branches (GB) form in wild type conditions (yellow arrows in (**a**, **c**, **e**)). In *TNFR-wgn* mutants more terminal branches and cells arise from the tip of DBs (yellow arrows in (**b**, **f**, **g**)) or in the stalk (white arrows in (**b**, **f**, **g**)) and in GBs and Lateral trunk (LT) (yellow arrows in (**d**)). In *TNFR-wgn* over-expression, terminal cells and terminal branches are not present (blue arrows in (**h**)). **i**, **j** Quantification of the percentage of dorsal branches with indicated phenotypes. All embryos analysed showed defects (100% penetrance). We compared the proportion of branches showing two or more terminal cells per DB at the tip, and proportion of branches with terminal cells in the stalk in *TNFR-wgn^{KO}* and two independent RNAi lines. One of them (*UASwgn-VSH330339*) showed a comparable expressivity to that of *TNFR-wgn^{KO}* mutants, indicating that *TNFR-wgn* is required in tracheal cells. The other line (*UASwgn-Trip.HMC03962*) showed a milder phenotype, indicating a weaker interference of *TNFR-wgn*. The Error bars indicate SD of mean. *n*, number of DBs analysed, in brackets number of embryos analysed. ****$p < 0.0001$; *$p < 0.05$, ns not significant, Chi-squared test, two-sided. In (**i**), magenta asterisks refer to the *p* values comparing each mutant condition to the control. Purple asterisks refer to the *p* values comparing each RNAi line with the *TNFR-wgn^{KO}* mutant condition. **k** Scheme showing the tip of a DB with one terminal cell expressing DSRF (forming a terminal branch) and one fusion cell expressing Dys engaged in branch fusion. **l**–**n** Dorsal views of stage 15 embryos stained with CBP to visualise the tracheal tubes (cyan), DSRF to visualise the terminal cells (green) and Dys to visualise the fusion cells (magenta). Fusion cells are normally specified in gain or loss of *TNFR-wgn* function. Scale bar: 20 μm. Source data and details of statistical tests used and *p* values are provided as a Source Data file.

In line with this result, overactivation of the pathway, using the overexpression of *hep* or a constitutively active form of *hep* (*hep^{CA}*), did not prevent terminal cell specification (Fig. 3d, e). Similarly, the overexpression of Traf2 (the most upstream component of TNFR-mediated JNK pathway[27] that interacts with Wgn[16]) did not prevent terminal cell specification either (Fig. 3f).

Different reporters that typically indicate JNK activity (*puc-lacZ* or *Tre-GFP*) were not detectably expressed in tracheal cells (Fig. 3g, h), suggesting no role or a minor role of the pathway in the trachea. However, *puc-lacZ* tracheal expression was detected upon activation of the pathway with *hep^{CA}* (Fig. 3i). In contrast, upon overexpression of *TNFR-wgn*, *puc-lacZ* was not expressed in the tracheal cells (Fig. 3j), indicating that *TNFR-wgn* does not detectably activate the pathway.

Activation of JNK pathway by TNFR signalling is known to promote cell death in different cellular contexts[14,17]. We found massive cell death in the trachea upon expression of *hep^{CA}* (Fig. 3k), but not when we overexpressed *TNFR-wgn* (Fig. 3l). Altogether, these results indicate that *TNFR-wgn* overexpression does not reproduce the effects of JNK activation.

Finally, we found that the overexpression of *hep* cannot rescue the excess of terminal cells in *TNFR-wgn^{KO}* mutants (Fig. 3m), and that *bsk^{DN}* did not revert the loss of terminal cells produced by *TNFR-wgn* overexpression (Fig. 3n).

Altogether our results argue that *TNFR-wgn* does not act through the JNK pathway to regulate terminal cell differentiation in normal conditions.

## TNFR-Wgn acts independently of its canonical ligand
TNFR-Wgn was proposed to transduce the signal of the unique TNF in *Drosophila*, TNF-Egr[15,16]. Therefore, we investigated the possible involvement of TNF-Egr in the differentiation of terminal cells.

When we analysed null mutants for *TNF-egr* we could not detect DBs with more than 2 terminal cells, presence of terminal cells in the stalk, or a significant excess of terminal cells in GBs as in *TNFR-wgn* mutants (Fig. 4a, b). We detected a phenotype of 1 extra-terminal cell in DBs (Fig. 4a), similar to the effects of JNK pathway downregulation (Fig. 3b). These quantitative and qualitative phenotypic differences suggested that TNFR-Wgn may regulate terminal cell differentiation independently of its ligand TNF-Egr. In agreement with this hypothesis, we found that the overexpression of *TNFR-wgn* was still able to prevent terminal cell specification in the absence of *TNF-egr* (Fig. 4c).

We also analysed the pattern of TNF-Egr accumulation at the stages of terminal cell specification. We detected TNF-Egr expression in the amnioserosa and ventral nerve cord (BDGP database,[16]) (Fig. 4d, e). We then analysed in detail the accumulation of TNF-Egr with respect to the trachea and to TNFR-Wgn accumulation. We could not detect tracheal expression of TNF-Egr (Fig. 4f) or TNF-Egr colocalising with TNFR-Wgn in the tracheal cells (Fig. 4g), although we detected some co-localisation of TNF-Egr and TNFR-Wgn in tissues other than the trachea (Fig. 4g). This result suggested that TNF-Egr may not reach the tracheal cells to activate TNFR-Wgn.

We also investigated the effect of *TNF-egr* overexpression in the trachea. Instead of producing a phenotype comparable to that of *TNFR-wgn* overexpression (i.e. absence of terminal cells), it produced a similar effect to the *TNFR-wgn* loss of function (i.e. excess of terminal cells, also in the DB stalk) (Fig. 4h). *TNF-egr* tracheal overexpression did not lead to *puc-lacZ* expression in the tracheal cells (Fig. 4i) and did not promote cell death (Fig. 4j), indicating that *TNF-egr* does not detectably activate the JNK pathway under these conditions.

Our results fit in a model in which TNF-Egr interferes with an activity of TNFR-Wgn independent of JNK. In this scenario, TNF-Egr in the trachea would bind and sequester TNFR-Wgn, preventing it from performing its independent activity (see below). In agreement with this hypothesis, we found that increasing TNFR-Wgn levels suppressed the phenotype of extra terminal cells observed in conditions of TNF-Egr tracheal overexpression, producing lack of terminal cell specification (Fig. 4k, l).

Altogether our analysis of the phenotypes of the gain and loss of TNF-Egr function, its pattern of accumulation and genetic experiments strongly suggest that TNFR-Wgn regulates terminal cell differentiation independently of its ligand TNF-Egr, although we cannot completely discard a minor contribution.

## TNFR-Wgn regulates ERK-MAP kinase signalling pathway
We asked how TNFR-Wgn regulates terminal cell differentiation. Terminal cell differentiation was previously shown to depend on FGF-Bnl/FGFR-Btl[22–24,28], which transduce the signal through the ERK-MAPK cascade[29] (Supplementary Fig. 2a). FGFR-Btl is known to be the sole contributor for ERK-MAPK activation at the tips of the migrating tracheal branches[29]. Thus, in wild type conditions, FGFR-Btl activation leads to phosphorylation of ERK that enters the nucleus and activates the terminal cell program in the tip cell (Fig. 5a).

Because TNFR-Wgn was also affecting terminal cell differentiation, we asked whether TNFR-Wgn was regulating the ERK-MAPK pathway and we found it did. In *TNFR-wgn* mutant conditions many more cells accumulated phosphorylated ERK/MAPK in the nucleus, correlating with the excess of DSRF cells (Fig. 5b). In contrast, in *TNFR-wgn* overexpression conditions, no phosphorylated ERK/MAPK accumulated in the nucleus of tip cells, correlating with absence of DSRF expressing cells (Fig. 5c). These results indicated that TNFR-Wgn restricts terminal cell number by downregulating the ERK-MAPK cascade activated by FGF-Bnl/FGFR-Btl.

Constitutive activation of Ras leads to supernumerary terminal cells (Fig. 5d). This phenotype was not reverted when simultaneously overexpressing TNFR-Wgn (Fig. 5e), suggesting that TNFR-Wgn acts upstream or in parallel to Ras.

We also analysed whether TNFR-Wgn controls ERK-MAPK by regulating the levels or expression pattern of *FGF-bnl*. We found no differences in *TNFR-wgn* mutants respect to control when we analysed

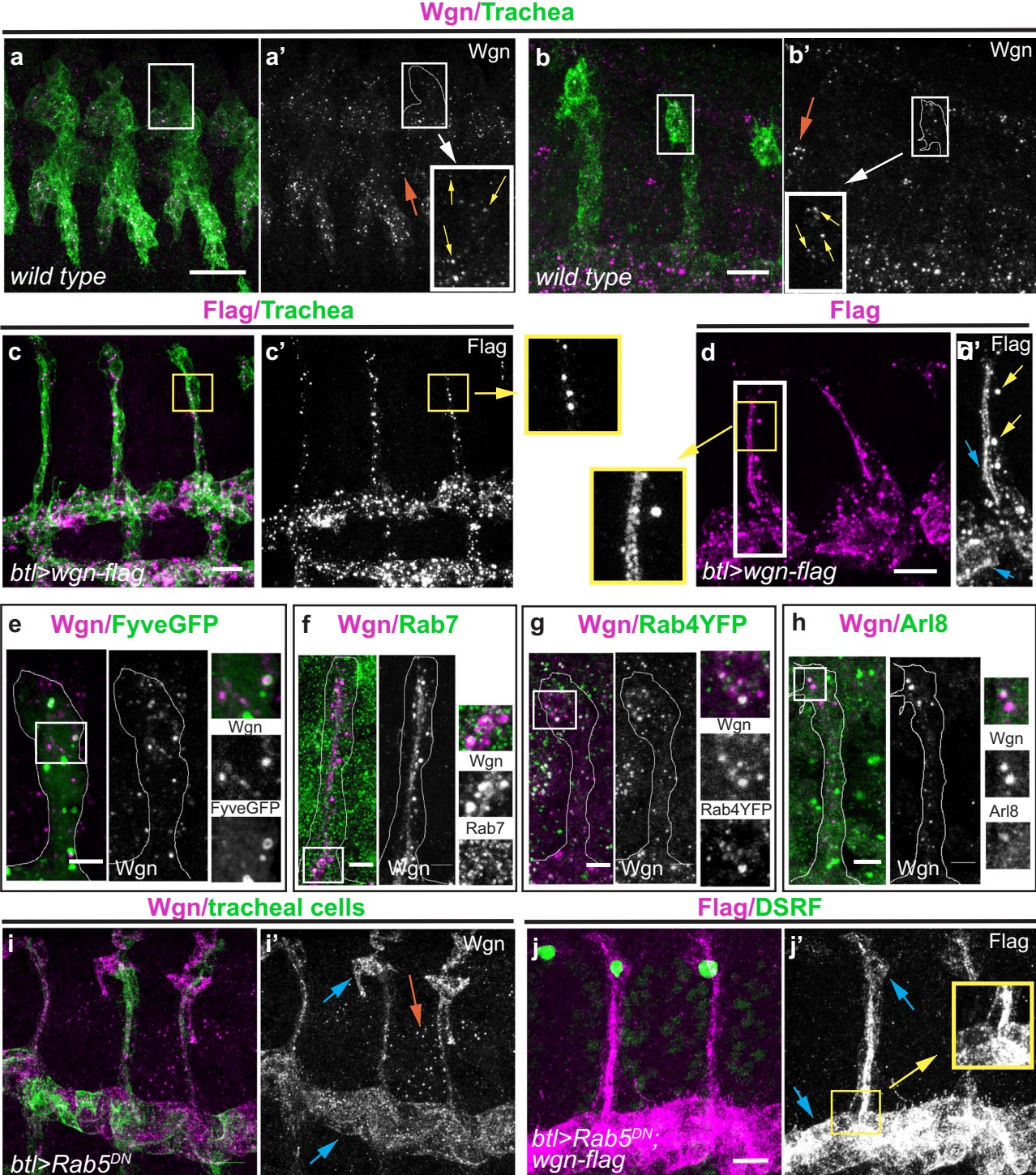

**Fig. 2 | Pattern of TNFR-Wgn protein in the tracheal system. a–j** All images show dorso-lateral views of stage 14 embryos focused to visualise the DBs, except a, which shows a general view of a stage 13 embryo. TNFR-Wgn (in magenta) is visualised using a Wgn antibody or a Flag antibody that recognises overexpressed *TNFR-wgn-Flag*. TNFR-Wgn accumulates in the tracheal cells in wild-type conditions in intracellular vesicles (yellow arrows in insets in (**a′**, **b′**); the tracheal system is outlined). TNFR-Wgn overexpression shows a similar pattern (**c** and inset), although, besides the accumulation in vesicles (yellow arrows in **d′**), accumulation in the apical region can be detected (blue arrows in **d′** and inset). **e–h** Colocalisation analysis indicates that TNFR-Wgn colocalises with late endosomal markers (Fyve-GFP, which labels the Fyve domain of the ESCRT-0 component Hrs, and Rab7), with the fast recycling marker Rab4 and with lysosomal markers (Arl8). Dorsal branches are outlined, and insets display colocalising vesicles. **i, j** When internalisation is compromised in tracheal cells, endogenous or overexpressed TNFR-Wgn localises to the membrane (blue arrows in (**i, j**) and inset in (**j′**)), although TNFR-Wgn still accumulates in punctae in other tissues (orange arrow in (**i**)). In these conditions, overexpression of TNFR-Wgn cannot prevent terminal cell differentiation (note presence of terminal cells in (**j**)). Scale bar: a 20 μm; (**b**,–**d**, i) 10 μm; (**e–h**) 5 μm.

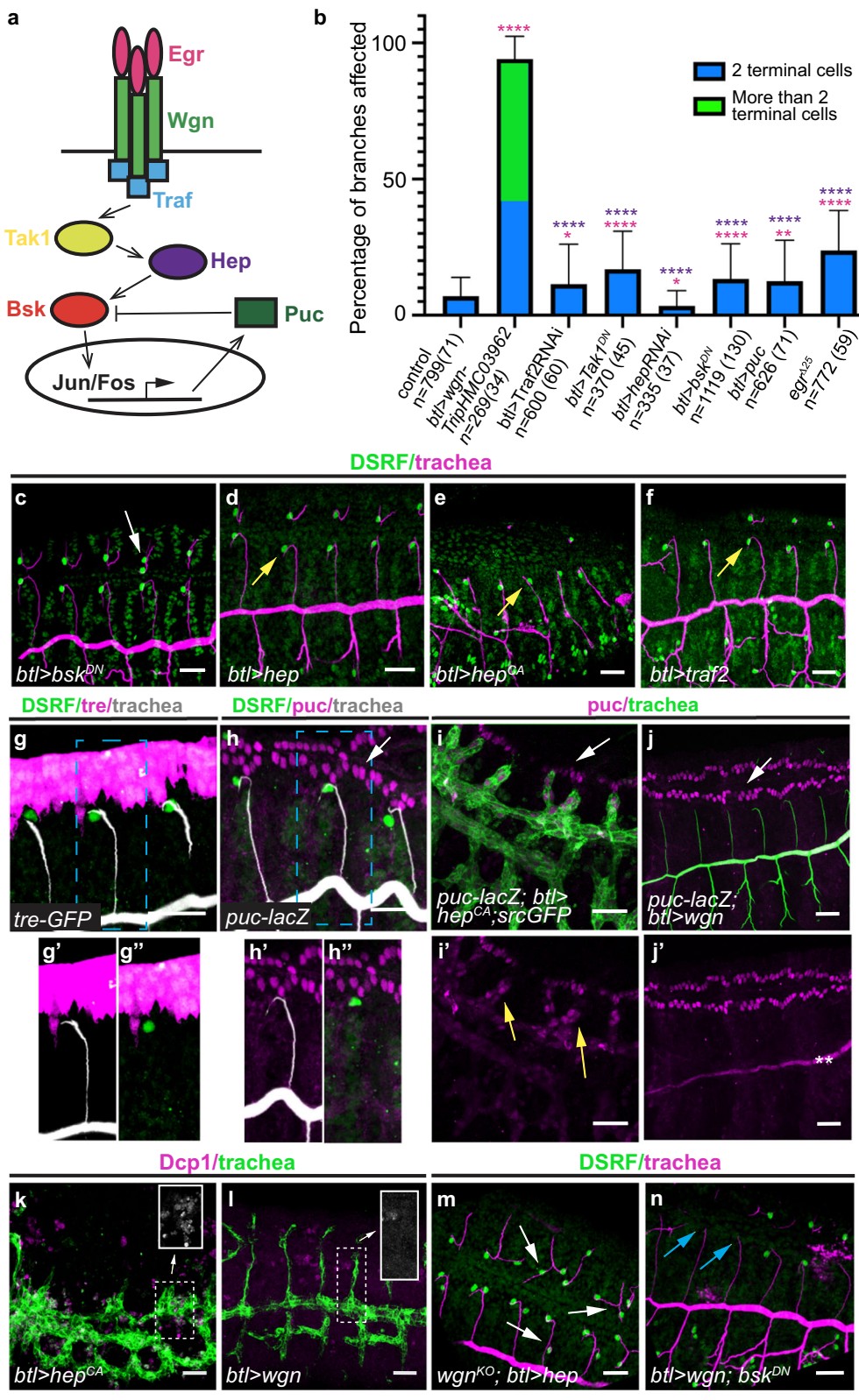

the pattern of *FGF-bnl* using the lexA-lexO system as reporter[30], or the transcriptional levels of *FGF-bnl* (Supplementary Fig. 2b–d), further supporting that TNFR-Wgn is required in the tracheal cells to regulate terminal cell number (Fig. 1g, i). Nevertheless, due to the very dynamic and complex pattern of *FGF-bnl*[30], we cannot completely discard minor effects of TNFR-Wgn on *FGF-bnl* pattern.

## TNFR-Wgn forms a complex with FGFR-Btl receptor

As TNFR-Wgn seemed to act upstream (or in parallel) of Ras and downstream of FGF-Bnl, we considered the possibility that it regulates the FGFR-Btl receptor. To investigate this possibility, we used tagged alleles of *FGFR-btl* (*FGFR-btl^{GFP}* and *FGFR-btl^{endoRFP,31}*,). *FGFR-btl*-tagged forms reproduced the pattern of expression of the gene (with higher

**Fig. 3 | Analysis of JNK pathway role in terminal cell specification. a** Scheme of the proposed Egr-Wgn-JNK signalling pathway. **b** Quantification of the percentage of dorsal branches that show the indicated phenotypes. Error bars indicate SD of mean. *n*, number of DBs analysed, in brackets number of embryos analysed. ****$p < 0.0001$; $0.001 <$**$p < 0.01$, *$p < 0.05$; Chi-squared test, two-sided. Magenta asterisks refer to the *p* values comparing each mutant condition to the control. Purple asterisks refer to the *p* values comparing each mutant condition to the *TNFR-wgn* downregulation condition. **c–f, m, n** Lateral views of stage 15 embryos stained with CBP to visualise the tracheal tubes (magenta) and with DSRF to visualise the terminal cells (green). JNK pathway downregulation produce a mild tracheal effect (white arrow in (**c**)). Note the presence of terminal cells in embryos with over-expressed/overactivated JNK or Traf2 (yellow arrows in (**d–f**)). Overexpression of *hep* cannot rescue the excess of terminal cells (white arrows) produced by *TNFR-*

*wgn* depletion (m). Downregulation of JNK cannot rescue the lack of terminal cells (blue arrows) produced by an excess of *TNFR-wgn* (*n*). **g, h** Lateral views of stage 14 embryos stained with GFP or β-Galactosidase (magenta) to visualise the JNK activity reporters, with CBP to visualise the tracheal tubes (white) and with DSRF to visualise the terminal cells (green). Note that the reporters are not expressed in the trachea. **i–l** Lateral views of stage 14/15 embryos stained in magenta with β-Galactosidase (**i, j**) to visualise the *puc-lacZ* reporter or with Dcp1 (**k, l**) to visualise cell death and in green to visualise the trachea. Note that *puc-lacZ* and cell death are activated upon JNK overactivation (yellow arrows in (**i'**), inset in (**k**)) but not upon *TNFR-wgn* overexpression (**j'** and inset in (**l**)). Asterisks in (**j'**) indicate staining background in the lumen of the trachea. White arrows in (**h–j**) point to *puc-lacZ* expression in the leading edge. Scale bar: 20 µm. Source data and details of statistical tests used and *p* values are provided as a Source Data file.

levels at the tip during the specification of tip cells[32], Supplementary Fig. 2e) and showed that the protein accumulated at the cell membrane, with a clear presence at the basal membrane (Fig. 5f and Supplementary Fig. 2d, e) from where the ligand FGF-Bnl is received[24,30]. In addition, a detailed subcellular analysis detected FGFR-Btl in intracellular vesicles (Fig. 5f). These vesicles likely reflect the normal intracellular trafficking and recycling of FGFR-Btl to ensure its proper localisation and activity[5,33–35]. Co-staining with TNFR-Wgn indicated that many of these FGFR-Btl vesicles also contained TNFR-Wgn (Fig. 5g, l, n). The presence of the two receptors in the same vesicles could just indicate that they traffic together, but it could also indicate a more direct interaction.

To test a possible interaction between TNFR-Wgn and FGFR-Btl we performed co-IP experiments. We expressed TNFR-Wgn and FGFR-Btl in salivary glands and we found that the two proteins also colocalised in intracellular vesicles (Supplementary Fig. 2f). TNFR-Wgn co-immunoprecipitated full length FGFR-Btl as well as a constitutively active form of FGFR-Btl in which the extracellular domain has been replaced with the dimerisation domain of the bacteriophage λ, λBtl (Fig. 5h, I and Supplementary Fig. 2g, h). These results indicate that TNFR-Wgn and FGFR-Btl form a complex and that the transmembrane and/or the intracellular domains of FGFR-Btl are sufficient for this interaction.

To further test the interaction between TNFR-Wgn and FGFR-Btl we performed in situ PLA (Proximity Ligation Assay). The PLA technology uses a pair of secondary antibodies labelled with oligonucleotides which when localised within very close proximity, are able to hybridise and undertake rolling circle amplification to generate a specific fluorescent signal after the addition of labelled probes (see Methods and[36]). Using Flag and GFP primary antibodies in the trachea of larvae expressing *TNFR-wgn-flag* and *FGFR-btl-GFP* (Fig. 5j, k), the PLA experiments suggested that the two proteins reside at a maximum distance of 0–40 nm, consistent with the hypothesis that they participate in the formation of a complex.

In a previous section we have hypothesised that the overexpression of *TNF-egr* produced a *TNFR-wgn* loss of function phenotype because TNF-Egr interferes with a TNF-Egr-independent activity of TNFR-Wgn. This activity could be related to this interaction with FGFR-Btl. In agreement with this hypothesis, we found that the proportion of common FGFR-Btl/TNFR-Wgn vesicles decreased when we overexpressed TNF-Egr (Fig. 5l–n). In addition, phosphorylated ERK/MAPK accumulated in more cells, correlating with the excess of DSRF cells (Supplementary Fig. 2i), indicating the activation of FGFR-Btl in these extra terminal cells, as it occurs in *TNFR-wgn* loss of function conditions (Fig. 5b).

**TNFR-Wgn regulates FGFR-Btl accumulation**

As we found that TNFR-Wgn forms a complex with FGFR-Btl, colocalises with FGFR-Btl in vesicles, and regulates its downstream activity, we asked whether TNFR-Wgn regulates FGFR-Btl in tracheal cells. We found that this was the case.

We first measured the levels of FGFR-Btl in tracheal cells. We observed a clear increase of FGFR-Btl levels in *TNFR-wgn* mutants, which was strongly detected at the basal membrane (Fig. 6a and Supplementary Fig. 3a, b). In contrast, we found a clear decrease of FGFR-Btl upon *TNFR-wgn* overexpression, which was very conspicuous particularly at the membrane of the tips (Fig. 6a and Supplementary Fig. 3a, c).

We then quantified the presence of FGFR-Btl intracellular vesicles. We detected a significant increase of FGFR-Btl vesicles in *TNFR-wgn* mutants and a decrease in *TNFR-wgn* overexpression conditions (Fig. 6b).

Thus, TNFR-Wgn regulates the general levels of FGFR-Btl accumulation and the presence of FGFR-Btl vesicles. Our results could indicate a role for TNFR-Wgn in promoting FGFR-Btl degradation. In this scenario, lack of TNFR-Wgn activity would lead to decreased FGFR-Btl degradation resulting in more vesicles and higher FGFR-Btl levels. In contrast, TNFR-Wgn overexpression would promote degradation resulting in less vesicles and lower levels. To test this possibility, we analysed the colocalisation of FGFR-Btl vesicles with Arl8 as a marker for lysosomes[37] in conditions of TNFR-Wgn overexpression. While there was a certain variability, we detected a significant increase of FGFR-Btl vesicles positive for Arl8 (Fig. 6c–e). This result suggests that a mechanism by which TNFR-Wgn could regulate FGFR-Btl trafficking and levels is by promoting its degradation.

**TNFR-Wgn regulates FGFR-Btl activity in terminal cells**

To further investigate whether TNFR-Wgn regulates terminal cell differentiation by regulating FGFR-Btl we performed different genetic experiments.

First, we asked whether decreasing FGFR-Btl levels or activity reverted the strong effect of *TNFR-wgn* loss of function on terminal cell differentiation, as predicted if TNFR-Wgn negatively regulates FGFR-Btl levels and activity. We found this was the case. The expression of *FGFR-btlRNAi* in tracheal cells produced an extreme phenotype of lack of terminal cells (Fig. 6f, h), presumably due to insufficient levels of receptor to activate the downstream signal. Expression of *FGFR-btlRNAi* completely reverted the excess of terminal cells of *TNFR-wgn* mutants (Fig. 6g,h), consistent with TNFR-Wgn acting by regulating FGFR-Btl. The FGF-Bnl is a haploinsufficient locus[21,24], and we observed a clear effect of lack of terminal cells when removing 1 copy of the gene (Supplementary Fig. 3d, g), presumably due to insufficient levels of receptor activation. Reducing the dose of *FGF-bnl* reverted the excess of terminal cells of *TNFR-wgn* mutants (Supplementary Fig. 3e–g), consistent with TNFR-Wgn regulating activated FGFR-Btl.

Then, we asked whether the overexpression of *FGFR-btl* could bypass the effect of *TNFR-wgn* overexpression as our model would predict. We first tested the effect of the overexpression of *FGFR-btl-GFP*, which rescues the lack of FGFR-Btl activity[38] and undergoes normal trafficking[33]. We found that the overexpression of *FGFR-btl-GFP* alone was not able to produce an excess of terminal cells (Supplementary Fig. 3i). *FGFR-btl-GFP* has been shown to be expressed at low levels and posttranscriptionally regulated by the

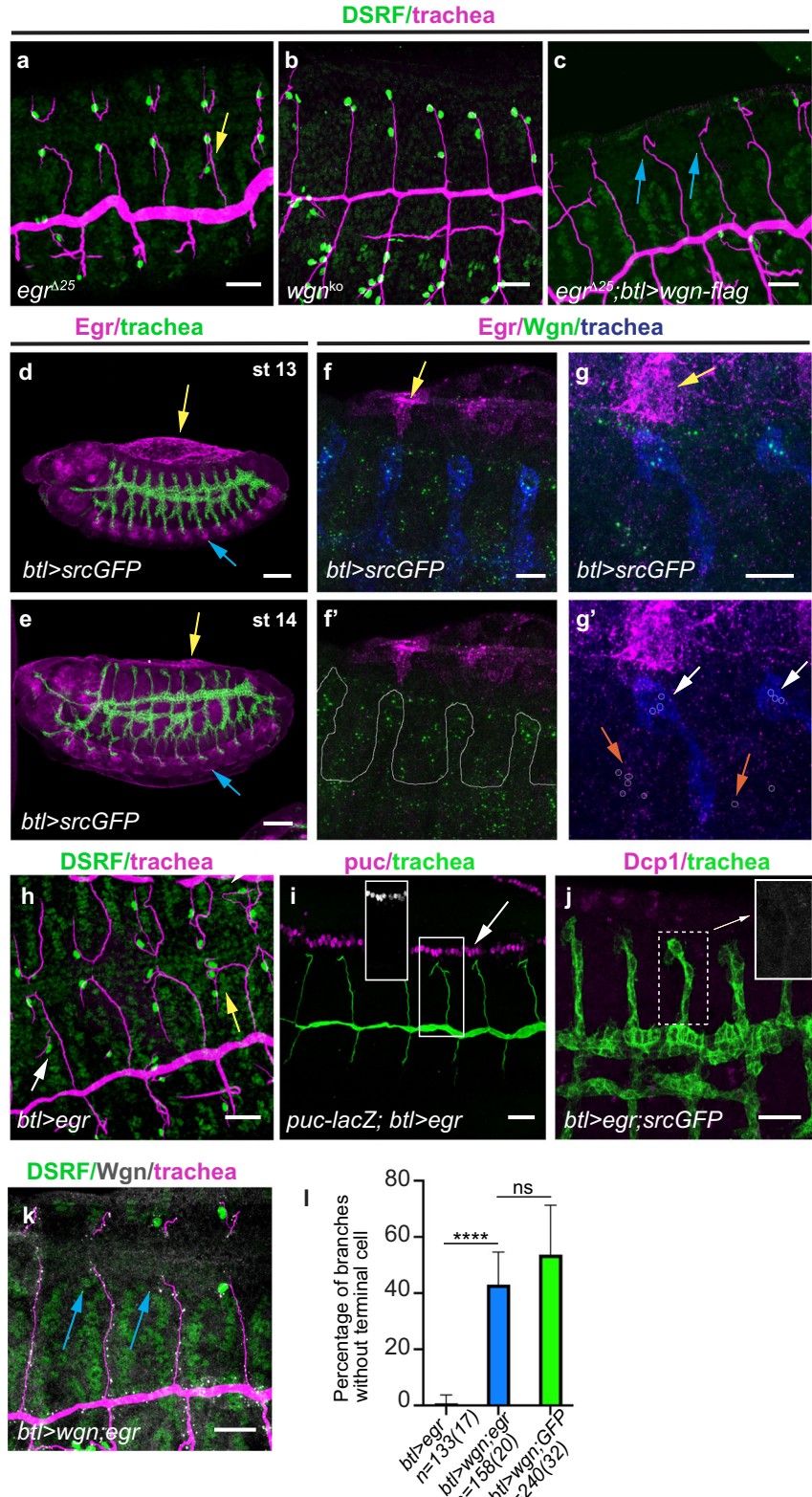

same mechanisms that control endogenous FGFR-Btl[33,34]. This may explain its lack of effect on terminal cell differentiation (see Discussion). In agreement with this lack of effect, the co-overexpression of *TNFR-wgn* and *FGFR-btl-GFP* produced a phenotype of lack of terminal cells (Supplementary Fig. 3h), indicating that the overexpression of *FGFR-btl* cannot rescue the defects produced by excess of *TNFR-wgn*. In contrast to this, we observed that the

overexpression of *FGF-bnl* (Supplementary Fig. 3j) or activated *FGFR-λbtl*, (Fig. 6i, k) led to extra terminal cells, showing that terminal cell differentiation depends on the levels of activated FGFR-Btl. We then asked whether the overexpression of *FGFR-λbtl* could bypass the effect of *TNFR-wgn* overexpression, and we found it did. The co-overexpression of *FGFR-λbtl* and *TNFR-wgn* produced a rescue of the lack of terminal cells and partially reverted the effects of

**Fig. 4 | Analysis of TNF-Egr in terminal cell specification. a–c** Lateral views of stage 15 embryos stained with CBP to visualise the tracheal tubes (magenta) and with DSRF for terminal cells (green). *TNF-egr* mutants show a mild phenotype of extra terminal cells (yellow arrow in a), compared to *TNFR-wgn* mutants (**b**). Overexpression of *TNFR-wgn* prevents terminal cell differentiation in the absence of *TNF-egr* (blue arrows in (**c**)). **d, e** Lateral views of whole embryos stained with TNF-Egr (magenta) and GFP to visualise the tracheal system (green). TNF-Egr is expressed in amnioserosa (yellow arrows) and CNS (blue arrows). **f, g** Lateral views of stage 14 embryos focused in dorsal branches stained with TNF-Egr (magenta), TNFR-Wgn (green) and GFP (tracheal system, blue). Note the expression of TNF-Egr in amnioserosa cells (yellow arrows in (**f, g**)) but not in trachea (oulined in (**f'**)) and some co-localisation of TNF-Egr and TNFR-Wgn (orange arrows in (**g'**)) in tissues other than the trachea (white arrows in (g')). **h–k** Lateral views of embryos overexpressing TNF-Egr, which leads to extra terminal cells (yellow arrows) also in the stalk (white arrow) (**h**). TNF-Egr does not activate *puc-lacZ* in tracheal cells (**i**, inset with puc-lacZ expression alone, white arrow points to expression in the leading edge) or cell death (**j**). Co-overexpression of TNFR-Wgn suppresses the extra terminal cell phenotype of TNF-Egr overexpression (blue arrows in (**k**)). **l** Quantification of the percentage of dorsal branches that lack terminal cells. Co-overexpression of TNFR-Wgn and TNF-Egr is slightly milder than the phenotype produced by TNFR-Wgn alone (together with a control UAS line), likely indicating the genetic interactions. The Error bars indicate SD of mean. *n*, number of DBs analysed, in brackets number of embryos analysed. ****$p < 0.0001$; ns not significant, Chi-squared test, two-sided. Scale bar: (**d, e**) 50 µm, (**a–c, h–k**) 20 µm, (**f, g**) 10 µm. Source data and details of statistical tests used and *p* values are provided as a Source Data file.

*FGFR-λbtl* (Fig. 6j, k), consistent with a model in which *TNFR-wgn* restricts terminal cell differentiation by regulating the activity of FGFR-Btl.

To further explore the role of TNFR-Wgn in regulating active FGFR-Btl, we analysed FGF-Bnl distribution in tracheal cells using an endogenously tagged-*bnl* allele (*bnl$^{endoGFP}$*,[31],). In wild type conditions we observed the pattern of FGF-Bnl punctae close to the tracheal cells[30]. In addition, we also detected FGF-Bnl accumulated in large and conspicuous intracellular vesicles containing also FGFR-Btl and TNFR-Wgn; these were mainly in the terminal cell (Fig. 7a). This intracellular FGF-Bnl signal may correspond to the internalisation of active ligand-receptor complexes that signal through ERK to trigger the terminal cell program. In contrast to the wild type, we found the presence of FGF-Bnl intracellular vesicles in several cells of dorsal branches in *TNFR-Wgn* mutants which activate DSRF (Fig. 7b). The results suggest that TNFR-Wgn prevents the accumulation/maintenance of FGF-Bnl/FGFR-Btl complexes, and likely FGFR-Btl activation, to more proximal regions in the branch.

### FGFR activity regulates TNFR-Wgn intracellular trafficking

We noticed that, during terminal cell differentiation, TNFR-Wgn intracellular vesicles were more abundant at the tip of the DBs than in the proximal part. Quantification indicated that 88% of TNFR-Wgn vesicles localised at the tip of the branches (*n* = 15 DBs analysed, Fig. 7c).

This result could indicate a faster or higher degradation of TNFR-Wgn at proximal regions and/or increased accumulation (transcriptional or posttranscriptional) at the tips. To investigate this further we aimed to block endocytic maturation. *shrub* encodes Vps32/Snf7, a subunit of the ESCRT III complex that regulates the endocytic sorting of cargoes leading to lysosomal degradation[39,40]. Overexpression of *shrub-GFP* phenocopies *shrb* loss of function[35,41], and we expressed *shrub-GFP* in the trachea to interfere with the trafficking to degradation. We found the presence of large vesicles containing TNFR-Wgn at both tip and proximal regions (Fig. 7d), suggesting that in normal conditions TNFR-Wgn is differentially processed throughout the dorsal branch, and likely degraded at the proximal region. Thus, our results suggest that regulated degradation along the dorsal branches likely contributes to TNFR-Wgn pattern, although other mechanisms may also contribute.

Is TNFR-Wgn protected from degradation at the tips? As we had found that at the tips TNFR-Wgn intracellular vesicles contained FGF-Bnl, we asked whether FGFR-Btl activation could affect TNFR-Wgn trafficking. We found that in conditions of FGF-Bnl overexpression, overexpressed TNFR-Wgn was significantly increased in intracellular vesicles compared to control (Fig. 7e, f). These results point to a regulatory feed-back loop in which FGF-Bnl presence (or activated FGFR-Btl receptor) stabilises TNFR-Wgn protein, while TNFR-Wgn regulates FGFR-Btl activity.

### Discussion

Our work illustrates the diversity of molecular mechanisms that implement receptor's activities. We propose a model in which two different receptors, each acting in a different way, regulate one physiological event. It was previously known that FGFR-Btl, by FGF-Bnl ligand activation, acts through the ERK-MAPK cascade to regulate the differentiation of tracheal terminal cells. Here we find that the TNFR-Wgn also regulates this process and propose that it does so in an unconventional manner, independently of its canonical ligand and downstream pathway, by directly regulating the activity of FGFR-Btl. Our biochemical data and in situ PLA experiments show that TNFR-Wgn and FGFR-Btl form a complex. In addition, our cellular analysis shows that while TNFR-Wgn can localise at the membrane, in normal conditions it is internalised into intracellular vesicles. Furthermore, we demonstrate that TNFR-Wgn regulates FGFR-Btl downstream signalling ERK-MAPK cascade and FGFR-Btl accumulation. Therefore, our results strongly suggest that through its constant trafficking and its ability to interact with FGFR-Btl, TNFR-Wgn regulates the activity of FGFR-Btl in terminal cell differentiation.

How does TNFR-Wgn regulate FGFR-Btl activity? It is known that the strength, the duration and also the subcellular localisation of activated FGFRs can determine the cellular outcome (reviewed in ref. [42]). For instance, internalisation of activated FGFR1 does not attenuate the signal but instead promotes stronger signalling through the ERK pathway, while AKT activation is independent of FGFR internalisation[43,44]. In the trachea, we propose a model in which FGFR-Btl is activated by the presence of FGF-Bnl at the tips, which stimulates its internalisation and signalling through the ERK pathway to regulate the differentiation of terminal cells. TNFR-Wgn, by forming a complex with FGFR-Btl and promoting its degradation, regulates FGFR-Btl signalling (Fig. 7g). Upon FGF-Bnl binding, internalised FGF-Bnl/FGFR-Btl could signal from endosomes activating the ERK pathway, and TNFR-Wgn could regulate the intensity or duration of this endosomal signal. Alternatively, or additionally, TNFR-Wgn could be promoting FGFR-Btl internalisation and its degradation modulating the availability of receptor to respond to ligand binding. TNFR-Wgn could promote degradation by recruiting or binding components of the cellular machinery involved in protein degradation. Interestingly, TNF activity has been shown to be modulated by NOPO[27,45], which encodes a TRIP (Traf interacting protein) E3 ubiquitin ligase that interacts with TRAF proteins[46] and is involved in the lysosome-dependent degradation of Traf3[47]. It will be interesting to investigate a possible involvement of NOPO in tracheal formation.

Our genetic experiments revealed that the overexpression of *FGFR-btl-GFP* cannot rescue the lack of terminal cells produced by *TNFR-wgn* overexpression, while the overexpression of activated *FGFR-λbtl* can do it. This could simply be due to the fact that FGFR-Btl-GFP is not expressed at sufficient levels to overcome the repressor effect of TNFR-Wgn[33,34]. Alternatively, or additionally, it could also indicate that TNFR-Wgn restricts terminal cell differentiation by regulating FGFR-Btl activity rather than the absolute levels of FGFR-Btl. Actually, the overexpression of wild type FGFR-Btl-GFP does not lead to extra terminal cell differentiation, suggesting that the levels of the receptor are not limiting. In this

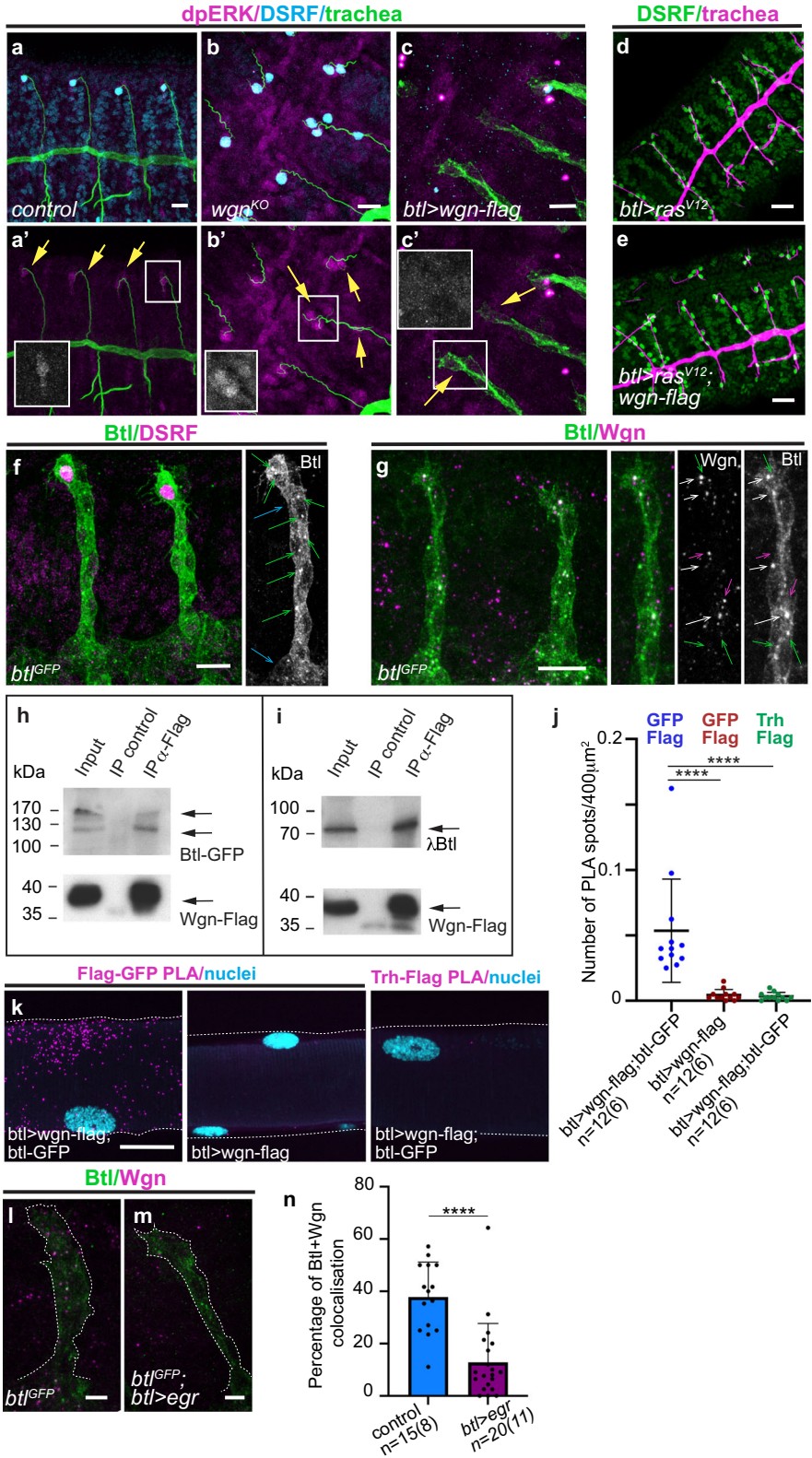

respect, the haploinsufficient nature of the FGF-Bnl loss of function[24] suggests that the limiting factor for FGFR-Btl activation is the ligand availability. We propose that TNFR-Wgn can interact with activated and non-activated FGFR-Btl receptors, but this interaction only has phenotypic consequences for terminal cell differentiation in the case of the activated receptor (Supplementary Fig. 4a). Several reports have shown that, although FGFRs can dimerise and

internalise in the absence of ligand, FGFRs activation stimulates its endocytosis (reviewed in[2]). Thus, it is possible that the activated FGFR-Btl (either *FGFR-λbtl*) or when activated by *FGF-Bnl)* is involved in a more dynamic trafficking, and this could facilitate interactions with TNFR-Wgn, which also undergoes trafficking. In summary, TNF-Wgn, by fine-tuning FGFR-Btl activity, would restrict the differentiation of terminal cells in tracheal branches. Finding new

**Fig. 5 | TNFR-Wgn forms a complex with FGFR-Btl and regulates its activity.**
**a**–**c** Dorso-lateral views of stage 14/15 embryos stained with dpERK (magenta),
DSRF (cyan) and a tracheal marker (green). dpERK accumulates in the nuclei of
the tip cell and activates DSRF. In *TNFR-wgn* mutants more cells accumulate
nuclear dpERK, while in *TNFR-wgn* overexpression no dpERK accumulates in tip
tracheal cells. **d**, **e** Dorso-lateral views of stage 14/15 embryos stained with DSRF
(green) and CBP (trachea, magenta). **f**, **g** Dorso-lateral views of two dorsal
branches of stage 14 embryos stained to detect FGFR-Btl (green) and DSRF or
TNFR-Wgn (magenta). Note the accumulation of FGFR-Btl at the basal mem-
brane (blue arrows in (**f**)) and in intracellular vesicles (green arrows in (**f**)). Many
FGFR-Btl containing vesicles contain also TNFR-Wgn (white arrows in (**g**)), but
TNFR-Wgn and FGFR-Btl single vesicles are also detected (magenta and green
arrows respectively). **h**, **i** Co-immunoprecipitation experiments. Western blot
using αBtl (upper panels) and αFlag (lower panels) of third-instar larvae salivary
gland extracts expressing either *TNFR-wgn-Flag* and *FGFR-btl-GFP* (**h**) or *TNFR-
wgn-Flag* and *FGFR-λbtl* (**i**). Extracts were immunoprecipitated using αFlag or a
control antibody (αAbd-B). Input corresponds to 5% of the immunoprecipitated

material. Note that αBtl recognises two specific bands. **j**, **k** In situ PLA experi-
ments. **k** Tracheal metamere 6 of larval DTs. Note the presence of PLA interac-
tions (magenta) when FGFR-Btl and TNFR-Wgn are expressed in the trachea.
**j** Quantification of PLA signal (number of PLA spots per 400 µm² region) indicate
a significant presence of interactions in the experimental condition (αGFP and
αFLAG in *btlGal4>btl-GFP; wgn-flag*) compared to control conditions (αGFP and
αFLAG in *btlGal4> wgn-flag*; αTrh and αFLAG in *btlGal4>btl-GFP;wgn-flag*). *n*,
number of DT regions analysed (metameres 5 and 6), in brackets number of
larvae analysed. Bars show mean ± SD. ****$p < 0.0001$, Kruskal–Wallis by ranks
followed by Dunn's multiple comparisons test. **l**, **m** Representative example of
lateral views of stage 14 embryos showing accumulation of FGFR-Btl (green) and
TNFR-Wgn (magenta). **n** Quantification of the percentage of FGFR-Btl vesicles
that contain TNFR-Wgn. *n*, number of DBs analysed, in brackets number of
embryos analysed. Bars show SD of mean. ****$p < 0.0001$, non-parametric Mann-
Whitney two-tailed test. Scale bar: (**a**–**c**, **f**, **g**) 10 µm; (**d**, **e**, **k**) 20 µm, **l**, **m** 5 µm.
Source data and details of statistical tests used and *p* values are provided as a
Source Data file.

regulators of FGFR internalisation, trafficking and activity is critical
for the development of new FGFR-directed therapies for disease and
cancer treatments[2,3,5].

TNFR-Wgn was identified as the receptor for a unique TNF in
Drosophila, TNF-Egr[14–17]. However, this finding was subsequently
questioned: it was shown that TNFR-Wgn can act independently of
TNF-Egr in photoreceptor axon pathfinding[25]. In addition, complicat-
ing the issue, a second TNFR was identified, named TNFR-Grindelwald
(Grnd), and it was proposed that this is the receptor that transduces
TNF-Egr functions[18]. In a further development it has now been
demonstrated that TNF-Egr can bind both TNFRs, TNFR-Grnd and
TNFR-Wgn, but with very different affinities. TNFR-Grnd binds TNF-Egr
with a much higher affinity than TNFR-Wgn, suggesting they have
different cellular functions[26]. Thus, a role for TNFR-Wgn, particularly in
physiological conditions, in the transduction of TNF-Egr activity,
became controversial. Here we find a role for TNFR-Wgn during normal
development, independent of TNF-Egr and the JNK pathway, in reg-
ulating the activity of the FGFR-Btl. However, we cannot discard a role
for TNF-Egr and the JNK pathway in the trachea mediated by TNFR-
Wgn in stress conditions, as the JNK and TNF/TNFR pathway play a key
role in the process[6,48,49].

In contrast to TNFR-Grnd which localises to the apical
membrane[18,26], we find that TNFR-Wgn does not stably localise to the
membrane of the tracheal cells and is found instead in intracellular
vesicles. This unusual localisation for a receptor seems to be a general
feature of TNFR-Wgn, as it was described to be localised in intracellular
vesicles also in imaginal tissues[26]. We observed this same pattern in
other tissues in which TNFR-Wgn is expressed (Supplementary
Fig. 4b). Our analysis indicates that these intracellular vesicles mostly
correspond to endosomes. When endocytic uptake is generally com-
promised, TNFR-Wgn is stabilised at the membrane, strongly sug-
gesting that in normal conditions the receptor is internalised after
reaching the membrane. In addition, when TNFR-Wgn internalisation is
compromised, the capacity of TNFR-Wgn to prevent terminal cell dif-
ferentiation is lost. Thus, we propose that TNFR-Wgn is constitutively
internalised, as it is the case for transferrin receptors[50], and that this
internalisation is absolutely required for its activity. Future analysis of
the intracellular domain of TNFR-Wgn should help to identify the
signals that promote this internalisation. Interestingly, TNFR-Wgn
contains a dileucine motif in the intracellular domain, which is not
present in TNFR-Grnd receptor and could act as a recognition motif for
internalisation[51,52].

We find that TNFR-Wgn forms a complex with FGFR-Btl. It was
previously shown that TNFR-Wgn can also physically interact with
Moesin in the context of photoreceptor axon guidance[25]. Thus, TNFR-
Wgn has the ability to bind diverse proteins, besides its canonical
adaptor protein Traf2[16], and thereby regulate their activity.

Interestingly, not only TNFR-Wgn, but also the other *Drosophila* TNFR,
TNFR-Grnd, was shown to be able to bind an unrelated protein, Veli[18].
Strikingly, Fn14, a rat TNFR superfamily member, was shown to phy-
sically interact with FGFR1[53]. Altogether these results indicate that
TNFR members have the ability to bind unrelated proteins, and we
propose that binding unrelated proteins and regulating their function
may be a general mechanism of activity for TNFRs. The participation of
TNF-TNFRs in cancer and inflammatory diseases is well-documented,
and TNF-therapies directed to control their activity have been devel-
oped, but improved therapies are needed to be more effective and
avoid undesirable side effects[6,7]. A better understanding of the mole-
cular mechanisms of TNFRs is key to developing improved therapies.

## Methods

### *Drosophila* strains and maintenance
*Drosophila melanogaster* embryos, larvae or flies of the different gen-
otypes specified in the manuscript were used. *Drosophila melanogaster*
studies are not subjected to ethical regulation. Data was obtained from
large pools of embryos or larvae, which contained an equivalent pro-
portion of both sexes. All *Drosophila* strains were raised at 25 °C under
standard conditions. Balancer chromosomes were used to follow the
mutations and constructs of interest in the different chromosomes.
For overexpression experiments, we used the Gal4 drivers *btlGal4* (in
all tracheal cells) and *fkhGal4* (in salivary glands). The overexpression
experiments were performed using the Gal4/UAS system[54]. To max-
imise the expression of the transgenes, crosses were kept at 29 °C. The
fly strains used are the following: stocks obtained from the Bloo-
mington Drosophila Stock Center: *y¹w¹¹¹⁸* (# 6598), *UAS-Cherry.NLS*
(#38425), *UASwgn-Trip.HMC03962* (#55275), *UAS-Traf2* (#58991),
*UAS-Tak1 DN* (#58811), *UAS-hepCA* (#9306), *UAS-bskDN* (#6409), *tre-GFP*
(#59010), *UAS-FyveGFP-myc* (#42716), *Rab4^EYFP* (#62542), *UAS-Ras^V12*
(#4847), *UAS-btlGFP* (#41802), UAS*btl.λ* (#29046), *UAS-btl RNAi*
(#60013), *bnl^008S7* (#6384), *UAS-bnl* (#64232), *UAS-shrubGFP* (#32559);
stocks from the Vienna Drosophila Resource Center: *UASwgn-
VSH330339* (#330339), *UAS-Traf2 RNAi* (#16125), *UAS-hepRNAi*
(#109277), *btl^GFP* (#318302). *btl-Gal4* was kindly provided by Prof. S.
Hayashi; *fkh-Gal4* was kindly provided by Prof. D. Andrew; *UAS-bnlGFP,
bnl-LexA, lexO-CAAXmcherry, bnl^endoGFP, btl^endoRFP* were kindly provided
by Prof. S. Roy[31]; *UAS-Rab5^DN* was kindly provided by M. González-
Gaitán; *wgn^KO* is described in;[18] *UASwgn-flag* is described in[25]; *egr^A25* is
described in[55]; *UAS-egr* is described in[17]; *UAS-puc* and *puc lacZ* are
described in[56]; *btl-Gal4,UAS-srcGFP* and *UAS-sdk-V5* were generated in
our laboratory.

### Immunohistochemistry and antibodies
Embryos were stained following standard protocols and staged as
described[57]. Embryos were fixed in 4% formaldehyde (Sigma-Aldrich)

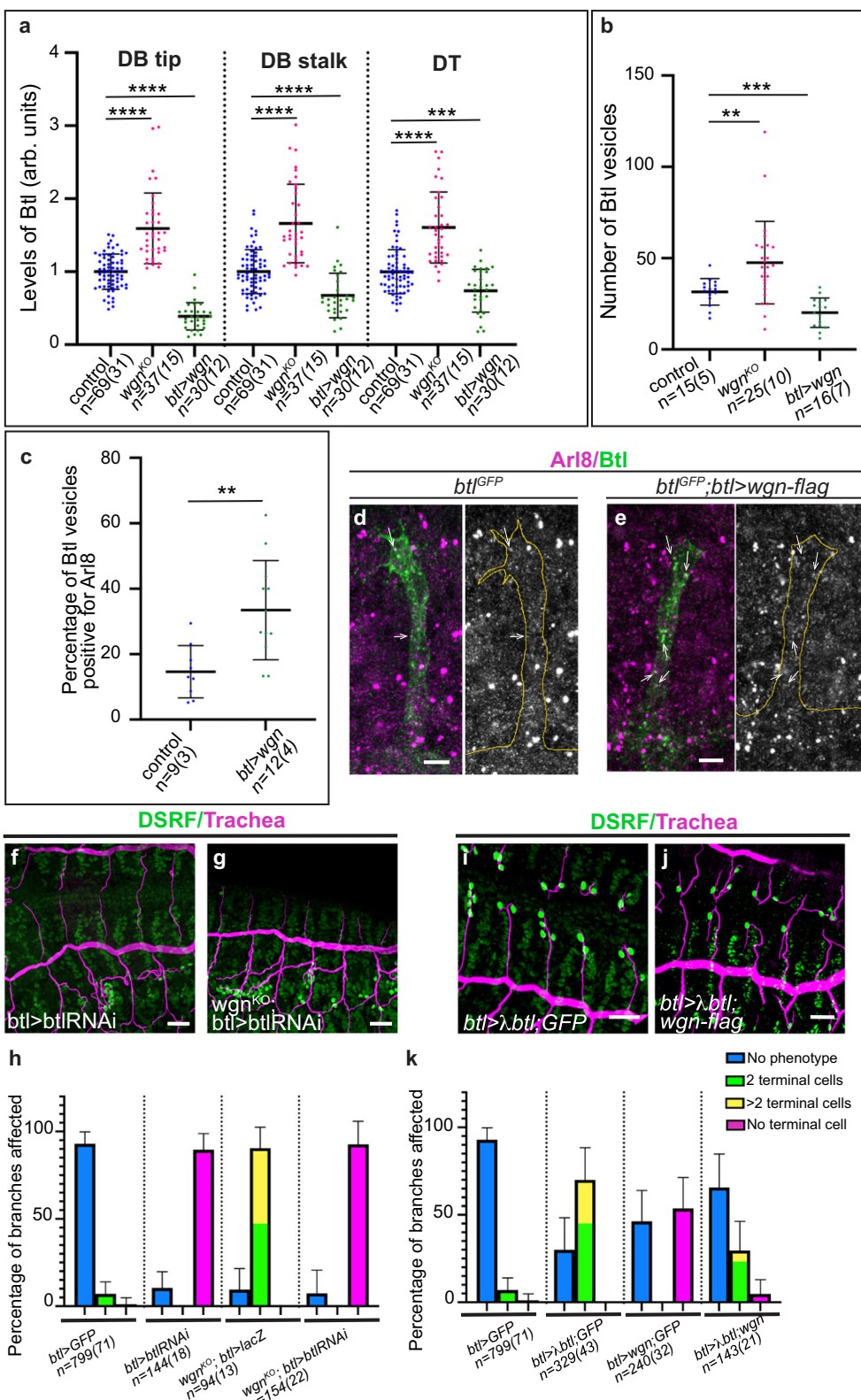

in PBS1x-Heptane (1:1) for 20 min. Embryos transferred to new tubes were washed in PBT-BSA blocking solution and shaken in a rotator device at room temperature. Embryos were incubated with the primary antibodies in PBT-BSA overnight at 4 °C. Secondary antibodies diluted in PBT-BSA (and for the CBP staining) were added after washing and were incubated at room temperature for 2–5 h in the dark. Embryos were washed, mounted on microscope glass slides and

covered with thin glass slides. The following primary antibodies were used: rabbit anti-Arl8 (1:100, AB_2618258, DSHB); mouse anti-Wgn (1:200, kindly provided by K. Basler); rabbit anti-DSRF (1:400, kindly provided by N. Martín, Prof. J.Casanova lab); goat anti-GFP (1:600, ab6673, AbCam); rabbit anti-GFP (1:600, A11122, ThermoFisher Scientific–Invitrogen); mouse anti-flag (for IF 1:200, for WB 1:10000, A00187, clone 5A8E5, GenScript); mouse anti-dpERK (1:100, M8159,

**Fig. 6 | Interactions TNFR-Wgn/ FGFR-Btl. a–c** Scatter plots quantifying the levels of FGFR-Btl accumulation at the tip or stalk of the DBs or in the DT (**a**), the number of FGFR-Btl intracellular vesicles (**b**), or the proportion of FGFR-Btl intracellular vesicles that are positive for Arl8. Genotypes are indicated. *n*, number of DBs or DTs analysed, in brackets number of embryos analysed. Bars show mean ± SD. ****$p < 0.0001$; $0.0001 < ***p < 0.001$, $0.001 <**p < 0.01$, unpaired *t* test two-tailed with Welch's correction (**b, c**) or non-parametric Mann–Whitney two-tailed test (**a, b**). **d, e** Lateral views of stage 14 embryos showing a representative example of the colocalisation of FGFR-Btl (green) and Arl8 (magenta) in the indicated geno- types. White arrows point to colocalising FGFR-Btl/Arl8 vesicles. **f, g, i, j** Dorso-

lateral views of embryos stained with DSRF (green) and CBP as a tracheal marker (magenta). **h, k** Quantification of the percentage of dorsal branches that show the indicated phenotypes. Note the rescue of extra terminal cells in *TNFR-wgn* mutants when *FGFR-btl* is downregulated. Note the phenotype of extra terminal cells when FGFR-Btl is activated and the lack of terminal cells when TNFR-Wgn is over- expressed; the combination of the two conditions results in a rescue of each phe- notype. Bars show SD of mean. *n*, number of DBs analysed, in brackets number of embryos analysed. Scale bar: (**f, g, i, j, g**) 20 μm, d,e 5 μm. Source data and details of statistical tests used and *p* values are provided as a Source Data file.

clone MAPK-YT, Sigma); rabbit anti-btl (1:2000 for WB, kindly pro- vided by J. Casanova lab); rabbit anti-Dys (1:500, kindly provided by L. Jiang); rabbit anti-Rab7 (1:1000, kindly provided by T. Tanaka[58].); chicken anti-β Gal (1:600, ab9361, AbCam); rabbit anti-RFP (1:300, ab62341, AbCam); mouse anti-Abd-B (1A2E9, AB_528061, DSHB); rabbit anti-Cleaved Drosophila Dcp1 (1:100, 9578S, Cell Signaling Technol- ogy); rabbit anti-egr (1:50, described in[59]); rabbit anti-trh (1:100, kindly provided by J. Casanova); rabbit anti-P-Histone H3 (1:100, 9701S, Cell Signaling Technology); rat anti-Dcad2 (for WB 1:4000, AB_528120, DSHB); mouse anti-V5 (for WB 1:8000, R960-25, clone SV5-Pk1, Ther- moFisher Scientific – Invitrogen); Chitin Binding Probe fluorescently labelled CBP (1:300, kindly provided by N. Martín, Prof. J.Casanova lab). The following secondary antibodies were used at 1:300: Cy3 AffiniPure Donkey Anti-Mouse IgG (H + L), 715-165-150; Cy3 AffiniPure Donkey Anti-Chicken IgY (IgG) (H + L), 703-165-155; Cy2 AffiniPure Donkey Anti-Chicken IgY (IgG) (H + L), 703-225-155; Cy5 AffiniPure Donkey Anti Rabbit IgG (H + L), 711-175-152; Cy5 AffiniPure Donkey Anti-Goat IgG (H + L), 705-175-147; Cy3 AffiniPure Goat Anti-Mouse IgG (H + L), 115-165-003; Cy2 AffiniPure Goat Anti Rabbit IgG (H + L), 111- 225-144; Cy5 AffiniPure Goat Anti-Mouse IgG (H + L), 115-175-146; Cy5 AffiniPure Goat Anti-Rabbit IgG (H + L), 111-175-144 (Jackson Inmunor- esearch) and Alexa Fluor® 647 Donkey anti mouse, A31571; Alexa Fluor Plus 488 Donkey anti-Goat IgG (H + L), A32814 (Life Technologies /Thermofisher Scientific).

## Image acquisition

Images from fixed embryos were taken using Leica TCS-SPE or Leica DMI6000 TCS-SP5 laser confocal microscopes, using Leica AF soft- ware, with the ×20 and ×63 immersion oil (1.40-0.60; Immersol 518F- Zeiss oil) objectives and additional zoom. Settings were adjusted for the different channels prior to image acquisition. Z-stack sections of 0.24–0.5 μm were acquired. The images were imported and processed using Fiji (ImageJ2, version 2.9.0/1.53t)[60] for measurements and adjustments, and assembled into figures using Adobe Photoshop 2020 and Illustrator 2020 (Adobe Inc.).

## Image analyses

**Quantification of terminal cells.** The number of terminal cells in dorsal or ganglionic branches was calculated using the nuclear factor DSRF as a marker for the nuclei of terminal cells. CBP was used to mark the lumen of the tracheal system to identify the different branches. The Max Intensity projections of confocal sections of late stage 14- stage 15 embryos, from different immunostaining experiments, were analysed using Fiji. DSRF positive nuclei were manually selected with the wand tool in Fiji and counted for each branch/embryo.

**Quantification of vesicles.** To quantify the number of vesicles, Max Intensity projections of late stage 14 embryos were taken and analysed using Fiji. After substracting the background, a Region Of Interest (ROI) was drawn to select the dorsal branch. A binary mask was created using the threshold tool and the watershed segmentation tool. Num- ber of vesicles were counted using the Analyse particles tool and the parameters were set to 0.05–1.7 mm$^2$ size, 0–1 circularity; the number of vesicles and a mask of the result were obtained.

Quantification of Wgn vesicles along the DB was performed using ImageJ plugin Analyse particles. For each dorsal branch, two regions of comparable area were analysed: the most distal, corresponding to the tip of the branch, and the most proximal, corresponding to the base of the branch.

**Quantification of levels.** To analyse the levels of Btl protein in the tracheal cells and to compare control and *wgn* mutant conditions we performed different independent experiments in which control and mutant embryos were collected, fixed and stained together. Confocal images of late stage 14 embryos were acquired with the same laser settings for each individual experiment. We then generated a projec- tion from the different stacks using the Max Intensity tool in the Fiji software and subtracted background. Three different ROIs were con- sidered and compared: the tip of a dorsal branch, a part of the stalk of the same branch and a part of the dorsal trunk near the dorsal branch. To measure the total Btl fluorescence at each ROI we obtained the "integrated density" in manually drawn areas at the tip, stalk and adjacent dorsal trunk with the freehand selection tool (see Supple- mentary Fig. S3a). The integrated density of each region was normal- ised to the average of the integrated densities calculated for the corresponding region (tip, stalk and dorsal trunk) in the control of each experiment. The obtained values were compared between con- trol and *wgn* mutant conditions using the Scatter Plot tool of GraphPad Prism.

**Colocalisation of vesicles.** Colocalisation analysis was performed using the ImageJ plugin Colocalisation highlighter, considering colo- calisation when the ratio of fluorescence intensities between the two channels analysed was above 0,5. Those fluorescence intensities above the threshold appear in a binary image colour as white (colocalised points). From this mask, we selected manually each vesicle with colo- calisation with the wand tool in Fiji and added it in the ROI Manager to be counted.

## Proximity ligation assay (PLA)

For FGFR-Btl-GFP-TNFR-Wgn-flag PLAs, *btlGal4>btl-GFP;-wgn-flag* and *btlGal4>wgn-flag* L3 wandering larvae were dissected in ice cold 1X PBS, removing the larval and imaginal tissues, while leaving the tra- cheal dorsal trunks attached to the larval cuticle throughout the pro- cedure. The tissues were then fixed in 4% formaldehyde for 20 min and rinsed 3 times with 0.1% Triton X-100 in PBS. The PLA was performed using the Duolink In Situ Red Starter Kit Mouse/ Rabbit according to the manufacturer's instructions with some modifications. After per- meabilization, the samples were blocked for 1 h at 37 °C, left at room temperature for 1 h more and then incubated overnight at 4 °C with the appropriate antibody combination; against GFP and FLAG for the *btlGal4>btl-GFP; wgn-flag* (experimental condition) and for the *btlGal4> wgn-flag* (control for PLA specificity; absence of one of the interacting partners) or against Trh and FLAG for the *btlGal4>btl-GFP; wgn-flag* (control for PLA specificity; use of an antibody that recognises a protein expressed in the tissue, not expected to interact with TNFR- Wgn-flag). The following day, the samples were incubated with the MINUS and PLUS PLA probes corresponding to the primary antibodies

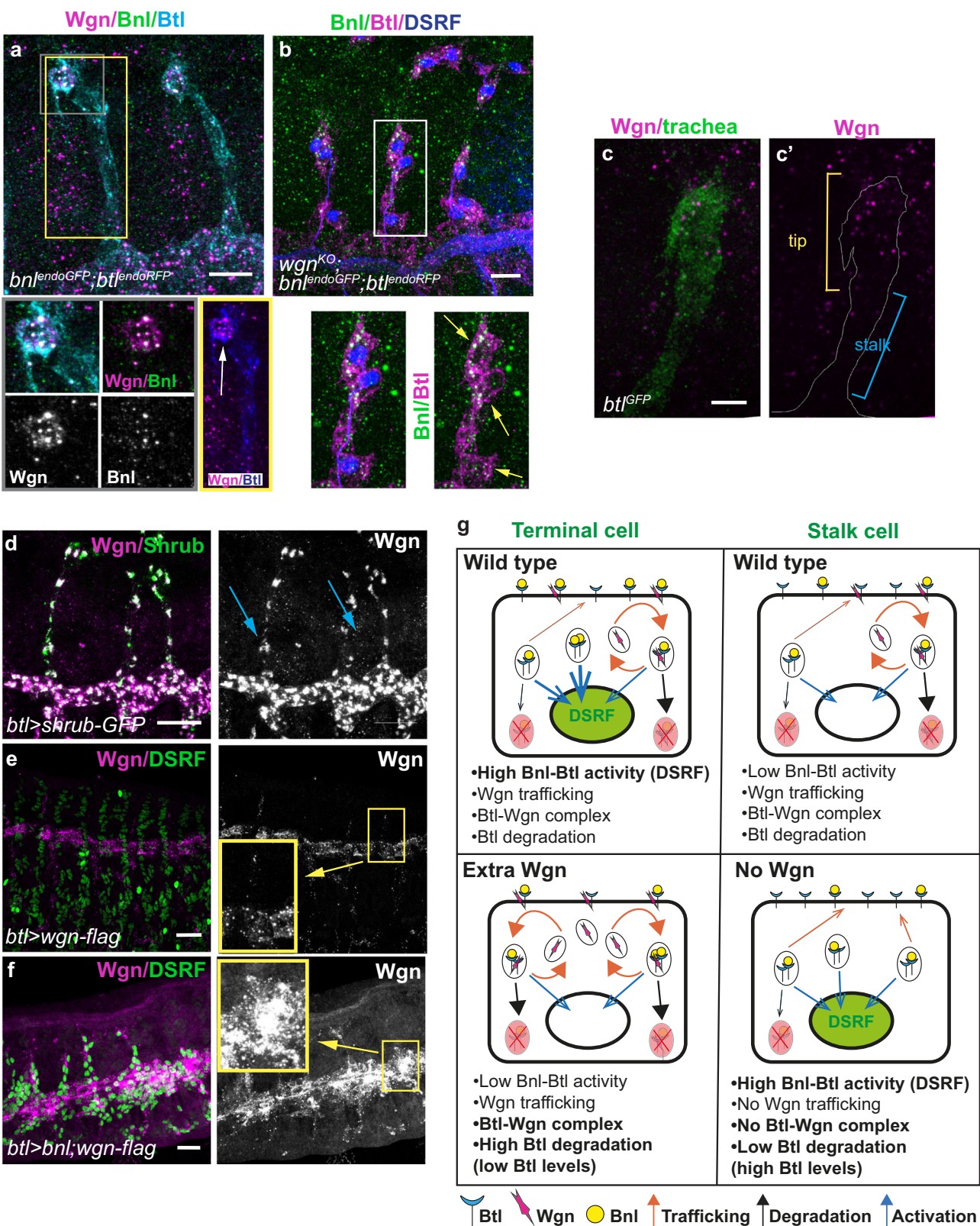

used, followed by 45 min ligation and 100 min amplification, using Texas-Red labelled oligos to generate the signal. Finally, the tracheal dorsal trunks were dissected in 1 X PBS and were mounted in Vectashield DAPI-containing medium. All incubations were performed in a humidity chamber using a volume of 20–40 µL per well. For imaging, a 63x/1.4 Oil DIC M27 objective of the Zeiss 880 confocal microscope was used. For quantification of PLAs, the number of PLA spots were counted in Fiji software across maximum intensity projections of raw files for each stack produced. In the image projections, subtract background with a rolling bar radius of 30 followed by a gaussian blur filter with the sigma set at 2 was used. The find maxima tool with prominence set at 9 was used to count the spots in 400 µm$^2$ regions of interest randomly selected within the cell cytoplasm of Tr5 and Tr6 tracheal metameres of the larval dorsal trunk.

**Fig. 7 | Modulation of TNFR-Wgn and model. a, b** Dorso-lateral views of a stage 14 embryos. **a** FGF-Bnl accumulates in intracellular vesicles that also contain TNFR-Wgn and FGFR-Btl particularly in the terminal cells (grey square). More TNFR-Wgn vesicles are detected at the tip of DBs (white arrow) compared to the base (yellow square). **b** In *TNFR-wgn* mutants, FGF-Bnl is detected in more cells of the DBs (yellow arrows). **c**–**f** Dorso-lateral views of a stage 14 embryos. **c** TNFR-Wgn vesicles are more abundant at the tip of the branches. **d** When the endocytic maturation is compromised, TNFR-Wgn is found also at high levels in the stalk cells at the base (blue arrows). **e, f** The overexpression of FGF-Bnl leads to an increased accumulation of TNFR-Wgn protein. **g** Model. TNFR-Wgn protein is constantly trafficking and can form a complex with FGFR-Btl receptor. Through this interaction, TNFR-Wgn promotes FGFR-Btl degradation. In the wild type, the terminal cell receives huge

amounts of FGF-Bnl ligand (due to source proximity), activating FGFR-Btl, which leads to activation of the ERK pathway and to transcriptional DSRF activation. This activation can bypass the negative effect of TNFR-Wgn. The stalk cell receives lower levels of FGF-Bnl ligand, leading to a weaker activation of the pathway. This, combined with the negative effect of TNFR-Wgn, prevents DSRF activation. When TNFR-Wgn is overexpressed, it promotes FGFR-Btl degradation, preventing DSRF activation in spite of the presence of high levels of FGF-Bnl ligand. This also leads to lower levels of FGFR-Btl. When TNFR-Wgn activity is lost, FGFR-Btl degradation decreases, and FGFR-Btl levels increase. Under these conditions, weak activation of the FGFR-Btl in stalk cells can lead to DSRF activation. Scale bar: (**a, b**) 10 μm; (**c**) 5 μm; (**d**–**f**) 20 μm. Source data are provided as a Source Data file.

## Co-immunoprecipitation assay
Assays were performed with extracts prepared from salivary glands of *Drosophila* third-instar larvae that were lysed in RIPA buffer (50 mM Tris-HCl pH8,150 mM NaCl, 0.1% SDS, 0.5% sodium deoxycholate,1% Triton X-100, 1 mM PMSF and protease inhibitors (cOmplete Tablets, Roche, 04693159001). Extracts were immunoprecipitated using αFlag antibodies or a control antibody (αAbd-B), followed by incubation with Protein G Dynabeads (Invitrogen, 10003D). Immunoprecipitates were washed with RIPA buffer and analysed by Western blot using either αBtl, αV5, DCAD2 or αFlag antibodies and the Immobilon ECL reagent (Millipore, WBKLS0100). Uncropped and unprocessed scans of blots are provided in the Source Data file or in the Supplementary Information.

## Quantitative RT-qPCR
Total RNAs were extracted with TRIzol Reagent (Life Technologies, 15596018), purified with NZY total RNA isolation kit (NZYTech, MB13402) and treated with DNAseI. cDNAs were prepared from 0.8 μg of RNA using RevertAid H Minus First Strand cDNA Synthesis Kit and oligo-dT primers (Life Technologies, K1632). -RT controls were included in qPCR reactions to discard genomic DNA contamination. qPCR was performed on Roche LightCycler 480 System using SYBER Green Master Mix (ThermoFisher, K0221). Transcriptional levels were normalised to ribosomal protein RpL23.

Primers used:
Bnl F GGATGCAAGTACCACCACCA
Bnl R CCCTATCGCTGGTTTCGCTA
RpL23 F GACAACACCGGAGCCAAGAACC
RpL23 R GTTTGCGCTGCCGAATAACCAC

## Quantification and statistical analysis
Data from quantifications was imported and treated in the Excel software and in GraphPad Prism 9.5.0 (GraphPad Software), where graphics were finally generated. Graphics shown in this work are scatter dot plots or columns, where bars indicate the mean and the standard deviation (S.D.). Statistical analyses comparing the different conditions were performed in GraphPad Prism 9.5.0 using for the comparison of two groups unpaired two-tailed student's *t-test* applying Welch's correction and two-tailed non-parametric Mann-Whitney test when data is not normally distributed, and for comparisons of three groups, Kruskal Wallis *H* test followed by Dunn's multiple comparisons test. Chi-squared test was used for comparisons of distributions of categorical variables. Differences were considered significant when $p < 0.05$. Significant differences are shown in the graphics as $*p < 0.05$, $**p < 0.01$, $***p < 0.001$, $****p < 0.0001$. Sample size ($n$) is provided in the figures or legends.

## Statistics and reproducibility
All confocal pictures presented in the figures show representative specimens from several acquisitions (>8) obtained from at least three independent experiments/samples. Data for analysis and phenotype quantification was acquired from at least three different independent experiments/samples. Each Co-IP experiment was performed three

times and a representative one is shown in Figures. For RT-qPCR experiments, three independent biological replicates for each genotype were performed. The PLA experiments were performed twice, using 3 biological replicates per genotype (a total of 6 individuals per genotype).

## Reporting summary
Further information on research design is available in the Nature Portfolio Reporting Summary linked to this article.

## Data availability
Flybase database was used (http://flybase.org/). The authors declare that all data supporting the findings of this study are available within the article and its supplementary information files and in the Source Data File. Source data are provided with this paper.

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

## Acknowledgements

The authors thank N. Martín for technical help and J. Ferrandiz for contributions at the initial stages of this work. We thank K. Basler for kindly providing the Wgn antibody, and S. Roy for kindly providing *bnl* and *btl* tagged-alleles. We also thank M. Milan, T. Tanaka, L. Jiang, S. Hayashi and D. Andrew for kindly providing flies and antibodies. We acknowledge the Bloomington Stock Centre and the Developmental Studies Hybridoma Bank for fly lines and antibodies. We thank the members of the Llimargas and Casanova labs for helpful discussions. We thank P.A. Lawrence for help, support and advice, and J. Casanova and M. Furriols for critical reading of the manuscript. PG is a researcher in Prof. Jordi Casanova's lab funded by Spanish Ministerio de Ciencia e Innovación (PGC2018-094254-B-100 grant) and the CERCA Program of the Catalan Government. This work was supported by funds (grants PGC2018-098449-B-I00 and PID2021-126689NB-I00) to ML from the Sp anish Ministerio de Ciencia e Innovación, MICINN (https://www.ciencia.gob.es/) and Agencia Estatal de Investigación, AEI (https://www.aei.gob.es/).

## Author contributions

Conceptualisation: A.L., M.L.; Methodology: A.L., M.L.E., P.G.; Investigation: A.L., M.L.E., P.G., M.L.; Formal Analysis: A.L., M.L.E., P.G., M.L.; Supervision: M.L.; Writing—original draft: M.L.; Writing—review & editing: A.L., M.L.E., P.G., M.L.; Funding acquisition: M.L.

## Competing interests

The authors declare no competing interests.
