## [Peer Review File · Nature Communications]

The TNFR Wengen regulates the FGF pathway by an unconventional mechanismReviewers' comments:

Reviewer #1 (Remarks to the Author):

The manuscript by Letizia et al describes a very interesting potential association between a TNF receptor as well as an FGF receptor in the development of the *Drosophila* tracheal system. This involves the modulation of terminal tracheal cell genesis by the FGF/FGFR signaling cascade. Here, we aim to have a direct interaction between the TNFR Wengen and the FGFR Btl, accelerating its degradation and thus directly affecting signaling. As said, the manuscript is interesting and the results are relevant, but some questions remain that should be clarified.

- 1) The obtained result describing that ectopic overexpression of Btl cannot compensate for the effect of Wengen is very difficult to understand with the presented model. An adjustment of the model would be required.
- 2) What exactly does Wengen bind to, the btl monomer, the activated dimer, both, the differentiation is not clear.
- 3) Central to the model is the hypothesis that activated btl is degraded more rapidly - here direct evidence supporting this hypothesis would be extremely helpful.
- 4) Alternative signaling systems should be considered. Downstream of Wengen, dTRAF2 could be acting and mediating the effects - right? Is Grindelwald relevant in this context?
- 5) How should we think of the logic of the interaction - does the effect of Wengen depend solely on its expression level, what other effects are mediated. The finding that JNK signaling does not seem to be active under normal conditions does not at all mean that it is unimportant as it is a classical stress signaling pathway.

Reviewer #2 (Remarks to the Author):

Authors report a TNFR, Wengen, and its role in *Drosophila* tracheal terminal branch specification. The authors performed mostly genetic experiments, in combination with a biochemical analysis to suggest a model where Wgn acts unconventionally by apparently interacting with Btl/FGFR. While the idea and some of the observations are interesting, I think the paper mostly reports very preliminary data with several conceptual and technical caveats. A cell biological mechanism is implicated based on just indirect genetic experiments, and a biochemical co-IP assay shown has problems in the experimental design. I believe, more deeper thinking and rigorous/thorough analyses are required to prove/disprove the model proposed. A better writing organization will help the reader. It is unclear what is the broad implication of this study. Conclusions in this paper appear to be claiming a lot more than what is shown. Overall, the results appear to be very preliminary, but with proper experiments, it can reveal an important biology, which controls tracheal terminal branching.

- 1) A general conceptual block in the paper is that it reads as if the authors are trying to just explain the mechanism of extra terminal branching observed in the Wgn-KO embryo and the this process is not critical to the overall growth of the tracheal system. What is the role of this extra terminal branching? Does it affect the organism/development? While authors invoke molecular crosstalk between Wgn and Btl/FGFR, the alternative probabilities that can also explain Wgn-KO phenotypes are not carefully examined. This is important, as the interactions between Btl and Wgn do not fully explain phenotypes. To my understanding and based on the Introduction written, not much is known about TNF/TNFR expression and activity in tracheal development. So, understanding Wgn's role is a new direction in the field. However, the authors did not provide sufficient evidence of TNFs/TNFRs expression/roles in tracheal branching before starting to speculate signaling crosstalk. A careful

analysis of the autonomous and non-autonomous roles of Wgn and other TNFR and their ligands is a prerequisite to carefully thinking and designing experiments to speculate molecular/cellular mechanisms leading to a phenotype that originates far downstream from these events.

Let me elaborate on some of these above comments going through line-by-line and by Figures.

2) In Line 24, sufficient evidence is not provided that Wgn works unconventionally. Still, Egr expression provided a phenotype. No data was shown of Egr or any other TNF binding/not-binding to Wgn.

3) Line 25 and 26 repeat the concept of Wengen working unconventionally. Secondly, it is confusing to read that Wgn represses terminal cell fates in one sentence, and then in the next sentence, it interacts with FGFR to activate terminal cell differentiation. The abstract does not give any clue of the mechanism or significance, and it appears to report very preliminary observations.

A general comment: I guess we do not know the exact mechanism that activates terminal cell differentiation (or any cell other fate specification), it would be ideal to state that a certain genotype is required for the terminal cell fates. A statement like FGFR signaling activates terminal cell differentiation might be wrong, but it is true that FGFR is required for differentiation.

4) Line 68 – citation needed.

Line 73-74 – the reason for investigating TNFR and FGFR needs to be clarified. What is known and what is unknown about the TNF-TNFR signaling in the tracheal is critical for the reader of a different field. Signaling cross-talks are not surprising, but why these two receptors but not others? Why not Hh or Dpp, which are also critical for the dorsal branch migration and terminal fates?

5) Figure 1: A: and Line 103, I think an introduction of all the tracheal branch names is helpful to general readers.

a. 1B: please provide the significance of terminal branching and what happens if more terminal branching occurs. What does it mean in terms of the organism? Do we have more cells that need oxygen?

b. Line 116 : quantification is based on only 2 embryos?

c. Since tracheal terminal branching involves a trachea-extrinsic control of branching from the target cells, and since wgn-KO affects all cells, it is critical to analyze autonomous and non-autonomous effects on trachea to correctly predict a molecular mechanism from the end-point genotype-phenotype analyses. For instance, Fig. 1D - wgnKO shows higher DSRF outside the trachea relative to control. Similarly, in Fig 1G, wgnRNAi expression in the trachea has fewer extra branches (a milder phenotype) and terminal cells in comparison to wgn-KO. There might also be a non-autonomous effect on the trachea, promoting terminal branches in the wgn-KO condition. Alternative non-autonomous effects are not shown.

d. Fig. 1D, where is the dorsal trunk? Similarly, 1G, has a dorsal trunk broken in one place. Authors mentioned that the early development is not affected by wgn-KO, but these images might suggest an effect and an important role of TNF/TNFR

e. Line 122: "tracheal cells to regulate terminal cell differentiation". I think a statement like --'regulate terminal cell number' is more directly based on the data. Authors should avoid speculative sentences, especially on mechanisms, like wgn regulate differentiation. Based on the picture presented, wgn caused an increase in terminal cell number.

f. No mention of Fig. 1M-O in the text.

g. Line 125: highly penetrant phenotype – please provide justification for 'highly' with quantitation.

h. A conceptual problem that Figure 1 presents - at this embryonic stage shown in figures, tracheal cells do not divide. So, if multiple terminal cells appear, where are they originating? More terminal cells could be due to the re-activation of cell division of a mother terminal cell or differentiation of the existing cells, or could it also be an early specification of terminal fates which get reorganized later into branched patterns? A number count of all cells in a branch would be informative. It is also important to examine an alternative possibility if wgn-KO can increase terminal cells by suppressing apoptosis of extra terminal cells that might be there before and that are not normally detected in wild type genotype-phenotype analyses. It is also possible that the apoptosis is blocked in the terminal branch recipient cells to which the branch will adhere, thereby stabilizing the extra branch that we can see now. Since wgn is involved in apoptotic pathway, this point needs to be clarified.

I think a rigorous analysis of expression of TNFs and TNFRs and their binding (like shown for Btl-Bnl) needs to be verified. There are reagents available to do so. As nothing is known about TNF-TNFR in trachea, this characterization is critical before speculating complex mechanisms and interactions between the wgn and other pathways.

5. Fig. 2 and related text:

a. Fig. 2B: significance of stats to be indicated.

b. "Downregulation of the pathway using different tools (bskDN, Tak1DN, hepRNAi, Traf2RNAi, or UASpuc) did not reproduce the TNFR-wgn loss of function phenotype." – it is possible that wgn-KO condition is due to non-autonomous effect. The comparison between an entire embryo wgn-KO with tissue-specific knockdown of JNK signaling is inaccurate. Authors should compare KO of some of these genes with wgn-KO. Alternatively, compare all the conditions with *btl>wng-RNAi*. Please examine thoroughly where else wgn is expressed and whether wgn can be complex with ligands.

c. Authors mentioned already focusing on a dorsal branch for simplicity, but why refer to another branch GB repeatedly without proper quantitation and reason?

d. Since authors are talking about an interaction of FGFR and TNFR within the tracheal cells, they clarify that the effect they see in wgn-KO is not due to an effect from outside the trachea. Many of the results (.e.g, Fig 2J; can be explained by non-autonomous effects in the trachea). Expression of Egr in the trachea inducing more branches could be due to the non-autonomous effect from cells that received Egr.

e. The way phenotypes are described is confusing. Based on the control image of DB, all branches robustly have 1 terminal cell, but more than 1 terminal cell in conditions like egr expression of bskDN is termed as not effective. They might not completely phenocopy a whole embryo KO phenotype, but they clearly produced more terminal cells than normal. In genetic interactions, it might be important not to focus on milder phenotypes, but the minor phenotypes often can give very important ideas of a molecular mechanism.

f. Fig 2K/L: wng-flag expression reduced DSRF expression, but we still see the extra branch in L popping out from the dorsal trunk, and we see the abnormal fusion. Why is it so?

Line 177-183: It is better not to assume what is physiological or not based on just knock-out and overexpression analyses. Wgn can have multiple roles through multiple pathways. However, classifying one as physiological but not the other based on phenotypic analyses of gene perturbation can potentially accumulate wrong information or the future. Secondly, if excess egr sequesters Wgn,

an overexpression of wgn should suppress overexpression phenotypes. Show Egr/other TNF binding to TNFR, as the authors suggest that the binding is not critical for the wgn-KO-induced phenotypes.

Line 185: Please mention how native Wgn is detected in the text. How specific is this Wgn antibody, and which part of Wgn is probed? Why is Wgn not in other tissues in Fig. 3A, unlike what is mentioned in the text? This might be important.

4) Fig.3A:

a. the ROI box on the trachea is not in focus. More importantly, please show a comparable stage with terminal cells turning away from the fusion cell. Wgn expression can be stage-specific as early-stage wgn-ko embryos have normal development of most primary branches. Note that endogenous receptors can be faintly localized on the membrane, and often the strong signal in the intracellular vesicles under confocal can mask the detection of faint cell surface localization. Since this is the first report of TNFR expression in the trachea, better imaging (e.g., airyscan, if available, or non-permeabilized Wgn staining for surface distribution) might be critical for this paper as it talks a lot about the difference between surface and intracellular distribution of the receptor.

b. Also, please mark the cell/tissue outlines and focus on the parts required to be shown. This is important because the authors claim that Wgn apparently has little function in membrane-localized forms.

c. In Fig. 3B,C dorsal trunk, wgn is more in the fusion cells? Fig 3F, Rab7 is everywhere, and specific colocalization is hard to predict.

d. Fig.3I, a clear increase in membrane Wgn with Rab5DN condition and membrane localization of Wgn correlated with its loss of terminal cell repression activity. However, a control expt must be done. Is this a condition-specific for Wgn, or all membrane proteins, including Btl/BMPR, or anything else expressed in these branches? Internalization of all these receptors is critical for canonical signaling pathways. A recent paper suggests that Btl in membrane-localized form can also act as a cell adhesion molecule for the GPI-anchored Bnl and the CAM activity (which does not require activation of canonical MAPK) is required for terminal branching (Du et al. 2022). So, all these different pieces of information suggest that we have a limited understanding of terminal branching and to clearly understand the role of TNF/TNFR's in this, a thorough analysis of these new signaling proteins is a pre-requisite.

e. "...traffics through the endocytic pathway and it is then degraded or recycled back to the membrane." - Authors previously mentioned Wgn, not in the membrane, but now concluded that Wgn recycled back to the membrane. Is this just based on lysosomal and Rab4/7 localization? Where is strong evidence for degradation or recycling?

f. Wgn clearly is enriched at the branch tip cells compared to other cells in a branch. Authors should focus and display images properly. Fig 3E-H; which branch are we looking at, and what is the outline of the trachea?

g. "Altogether, these results show that TNFR-Wgn is a highly dynamic protein that reaches 216 the membrane but is constantly internalised, preventing it to stably localise at the 217 membrane."

h. Authors repeatedly suggested dynamic activities, but no data was shown to support dynamism. Dynamism is inferred from just fixed-tissue localization in some specific compartments.

5) Fig. 4:

a. Since dpERK is a probe for terminal cells, even if Wgn-KO removes the terminal cell by an alternative route, it will not show dpERK-positive cells, as shown in Fig 4. How could authors conclude that Wgn-KO affected Btl-Bnl signaling without eliminating the other possibilities?

b. Line: 244 – cite these reagents for validation.

c. Fig. 4H: IP did with overexpression of both proteins in the salivary gland. Overexpression can cause false positive co-IP. This might not indicate a physiological interaction. Why is the salivary gland used? It is a non-essential organ in larvae. Secondly, no clear idea of what is the IP control in panel H. Is it the membrane GFP co-IP with Wgn? Can there be a control overexpression membrane protein that does not co-IP with Wgn?

Salivary glands normally do not have tracheal supply and Btl/Bnl expression. Does it express TNF/TNFR? Do we get any overexpression phenotype due to Wgn expression in the salivary gland?

d. Fig S2H is important data and needs to be in the main Figure, and the graph should be accompanied by a representative Figure panel. Why are authors not showing dpERK (activity) if the concern is Btl activation? If the hypothesis presented in this section is correct - Do you see Wgn localization on the membrane and more Btl in the cytoplasm (required for MAPK) when is Egr overexpressed? I think the data here is not strong enough to be confident about the model proposed.

e. Fig. 4K-L, is very important and needs to be supported by proper image panels. Secondly, the no. of btl puncta or levels could simply be a function of no. of cells in the tip or stalk or their size/shape/amount of membrane surface. A normalization/cell would be helpful to conclude.

f. Authors must examine whether the effect of Wgn is specific to Btl by examining control receptors/proteins, which might remain unchanged. Conclusions on degradation and so on are just based on Arl8 localization. Degradation needs to be verified with biochemical assay if this is important.

h. Fig S2, authors must focus on showing the same stage of embryos where there are terminal cells differentiating from fusion cells and also to the point of vesicles and membrane localization wrt to terminal cells. I guess these Btl reagents are published (please cite) and well-verified, and showing a validation of these reagents is less important than validating Wgn localization/expression (which needs to be reinforced with data presented in the paper).

i) Line 301-315: In a previous section, authors already showed that RasV12 induced ectopic DSRF positive terminal cells, and this cannot be suppressed by Wgn overexpression, so Wgn must act in parallel to Ras pathway or upstream between FGFR and Ras. Since activated Btl has no ligand binding domain, and the intracellular Kinase domains are permanently dimerized, so constitutively inducing Ras-MAPK signaling, it is not surprising that the const. activated Ras pathway is not suppressed by Wgn overexpression. All these phenotypes collectively suggest a parallel pathway of FGFR and Wgn, but not clear-cut interactions between the two pathways,

j) Btl:GFP overexpression can induce MAPK signaling and tracheal responses shown by other groups. It is also known that Btl level and localization to the membrane are dependent on Bnl signaling, so overexpression of Btl:GFP did not cause an extra terminal branch, but overexpression of Bnl did seem to be not matched with any of the prior reports on Btl-Bnl signaling and the existing model. It sounds like the authors are suggesting that overexpression of Btl:GFP did not activate signaling.

5) Fig 5:

a. Fig 5D, I do not understand if a Btl:GFP or Btl overexpression alone did not show an extra terminal branch. Why did authors need to examine something upstream to Btl? It is important that the authors focus on the cross-talks based on the available BtlDN (lacks intracellular kinase domains), loss of Btl (RNAi clones), and gain of Btl (clones). E.g. btl-RNAi clone and see what happens to Wgn. Unless there is an increase in the number of Bnl source cells in wgn KO condition (which is not verified), that can extrinsically control terminal cell fates. Also, a direct measure of MAPK signaling is required if activation of Btl is the critical component in the pathway predicted. A Btl-Bnl endosome might not always mean signaling activation. So a direct measure is more important than an indirect measure to

prove an outstanding claim.

b. Fig. 3H, All the conditions, like overexpressing a highly potent ligand like Bnl in the trachea, destroy tracheal branch growth. So, not sure what the authors are really comparing.

c. Line 333: Intracellular Wgn more in terminal cells - quantitation? Authors suggest degradation of Wgn based on more signal detected in the vesicular form inside. More signals could be due to more expression.

d. Line 338- what is shrub?

e. Line 339: "using shrub-GFP 33,34. We found the presence of large vesicles containing TNFR-Wgn at both tip and proximal regions (Fig 5F), suggesting that in normal conditions TNFR Wgn is differentially processed throughout the dorsal branch, and likely degraded faster at the proximal region, probably contributing to the TNFR-Wgn pattern." what is the basis of such an important conclusion about differential processing?

f. Feedback loop? The evidence is not strong enough to support the feedback loop. Too many claims without showing convincing data and analyzing the roles of TNF/TNFRs in tracheal very carefully!

6) Discussion:

353 "We evidence a model in which two different receptors, each acting by a different mechanism, regulate one physiological event." It would be best to first show a mechanism of Wgn, before claiming different mechanisms at the molecular level.

358: "In addition, our cellular analysis shows that while TNFR-Wgn can localise at the membrane, in normal conditions it is constantly and rapidly internalised into intracellular vesicles." - There were no experiments showing dynamics or kinetics and many normal conditions in the paper.

394: MAPK signaling needs to be shown under wgn conditions. A direct measure of btl activation is more important than Btl-Bnl colocalization. Can Btl activation be non-canonical, induced by TNFR, without Bnl?

Line 404: nothing was shown about Egr binding to Wgn in the trachea. How about Grnd expression in the trachea?

I think similarly, TNFR-Btl interaction if any needs to be carefully verified with proper experiments. Also, please remove all the claims of temporal dynamics which were not tested.

Reviewer #3 (Remarks to the Author):

Letizia and colleagues report an unconventional TNFR signaling in Drosophila tracheal development. Drosophila Egr-JNK signaling is conserved and plays various roles in immune response, apoptosis and cell growth. In stark contrast to this canonical TNF pathway, Wengen, a TNFR, is in a complex with Btl (FGFR), and repress the differentiation of the tracheal terminal cell. Mechanistically, Wengen undergoes constant internalization. It colocalizes with Btl in intracellular vesicles and regulates the intracellular transport and turnover of Btl. This study also reveals a feedback loop in which FGF/Bnl stabilizes Wgn protein.

The messages in this manuscript are noteworthy and informative. The conclusions are experimentally addressed. This manuscript would alert the field the involvement of TNFR in RTK-FGFR signaling-dependent tubulogenesis. It would be of help if authors could clarify some points below.

- 1 The phenotype of extra terminal branches and/or terminal cells is very striking. The mechanism in which Wgn interferes Btl is reasonable. There is also possibility although unlikely that Wgn alters FGF, Bnl. It would be even more convincing, if the authors could show that expression of bnl is unaltered, by bnl-lacZ, Bnl staining, or RT-PCR?
- 2 Authors show that Wgn colocalizes with Rab4, Rab7 and Arl8 and that perturbation of Rab5 suppresses TNFR-Wgn-dependent phenotype. Wonder if depletion of other endosomal components (e.g. Rab4) generates similar phenotype?
- 3 In addition to anti-Flag, is it possible to perform the co-IP experiment with anti-Wgn antibody to assay the interaction between Wgn and Btl at physiological level?
- 4 It has been reported that Awd (abnormal wing discs) regulates the endocytosis of Btl. Wonder if authors have genetic interaction between awd, wgn and btl.
- 5 The authors show that overexpression of Egr reduces Btl/Wgn vesicle. Because TNF/Egr triggers apoptosis in various tissues, it is good to know if this happens in tracheal system?
- 6 In line 326-327, the authors claim that Wgn restricts the maintenance of Bnl/Btl complexes in terminal cells. They also show that Wgn is in a complex with Btl. Are those three together in a complex? Or Wgn competes for Btl, and thus prevents its activation by Bnl? Please clarify.

Minor points:

- 1 It would be informative to provide p value and/or statistical significances in Fig. 2B, since the authors claim that egr mutants and JNK downregulation gave low penetrant phenotype in line 169, 170.
- 2 The terminal cell phenotypes of wgn mutants can be suppressed by expression of btlRNAi or btlDN?
- 3 In line 258, the authors claim that btl is not expressed in salivary gland. This statement is not needed for this co-IP experiment. The expression of btl-Gal4 in salivary gland is observed.
- 4 It would be of help if authors could provide more insights regarding how the membrane receptor, Wgn, promotes the degradation of Btl in the discussion?

Point-by-point response to reviewer's comments

Reviewer 1:

The manuscript by Letizia et al describes a very interesting potential association between a TNF receptor as well as an FGF receptor in the development of the *Drosophila* tracheal system. This involves the modulation of terminal tracheal cell genesis by the FGF/FGFR signaling cascade. Here, we aim to have a direct interaction between the TNFR Wengen and the FGFR Btl, accelerating its degradation and thus directly affecting signaling. As said, the manuscript is interesting and the results are relevant, but some questions remain that should be clarified.

We are happy to see that the reviewer finds our work relevant and interesting.

1) The obtained result describing that ectopic overexpression of Btl cannot compensate for the effect of Wengen is very difficult to understand with the presented model. An adjustment of the model would be required.

We agree with the reviewer that in our previous version of the manuscript the effect of the overexpression of Btl in a wild type background and in a background of Wgn overexpression was a bit confusing. We have tried to clarify this point in the results section (lines 381-389) and in the discussion section (lines 481-491). In addition, we have adjusted the model in Fig 7G and we have added a scheme in Fig S4A to help to discuss this point. We hope that these explanations and additions help to clarify this point.

2) What exactly does Wengen bind to, the btl monomer, the activated dimer, both, the differentiation is not clear.

In our work we show that Wgn forms a complex with a constitutive active form of FGFR-Btl (Fig 5I) in which the extracellular domain has been replaced with the dimerisation domain of the bacteriophage λ fused to the transmembrane (TM) and cytoplasmic (IC) domains of Btl. This form spontaneously dimerises being a constitutively active receptor (Lee et al., 1996). Therefore, our results indicate that Wgn forms a complex with Btl dimers.

On the other hand, since previous analyses have shown that FGF receptors form dimers through contacts between the TM domains in the absence of ligand (Sarabipour and Hristova, 2016), our work is not conclusive on whether Wgn interacts only with the dimer or both, monomer and dimer.

We have tried to address this point by analysing a direct interaction of Wgn with Btl monomers. To this end we have performed GST-pulldown experiments using a fusion of GST to either the IC or the TM+IC domains of Btl and protein extracts from salivary glands that express wgn-Flag. Unfortunately, both GST-Btl proteins (GST-TM+IC and GST-IC) are mostly insoluble when expressed in bacteria, suggesting

that they are not properly folded, and cannot be used to analyse protein-protein interactions (see Fig 1 in this document).

Different experimental approaches should be carried out to fully determine whether Wgn also interacts with the monomeric Btl receptor, for instance structural analyses. However, in the frame of our work, we clearly demonstrate that Wgn forms a complex with the active (dimer) form of Btl, which fits with our cellular and genetic data.

3) Central to the model is the hypothesis that activated btl is degraded more rapidly - here direct evidence supporting this hypothesis would be extremely helpful.

In our initial submission we proposed that Wgn is a highly dynamic protein that is rapidly internalised regulating in this way Btl trafficking. However, as pointed by Reviewer #2, we have not specifically addressed the dynamic behaviour of Wgn. With the results we obtained we can propose that Wgn protein is trafficked, but we cannot imply aspects related to dynamism. We have addressed this point in the text throughout all the manuscript in this revised version.

Our results indicate that Wgn regulates the activity and the levels of Btl. We also find that when Wgn is overexpressed, more Btl vesicles colocalise with a lysosomal marker, and this correlates with a decrease in Btl levels and number of vesicles. From these results we suggested that Wgn regulates Btl trafficking and degradation. Different scenarios could account for our observations. For instance, Wgn could regulate the proportion of Btl protein destined for degradation, or it could regulate the speed of trafficking to degradation, among others. We aim to disentangle the exact mechanism in future work that we are planning. In this line, we would like to argue that finding direct evidence for Btl degradation in the trachea upon Wgn manipulations is a hard task. To accurately address this point, the levels of Btl in tracheal cells should be measured after preventing transcription. This would probably require, in addition, sorting the tracheal cells from the whole embryo. An alternative approach that we are considering is to generate a photoconvertible allele of btl. This would allow us to photoconvert Btl protein at a desired spatio-temporal point and track the protein in time-lapse experiments to investigate its dynamic trafficking. In parallel we are also considering generating a wgn-photoconvertible allele. This approach requires multiple time-consuming steps (generation of a photoconvertible knock-in allele by CRISPR techniques, validation of the allele, setting the parameters for the experiment, among others) that we plan to address in future work in the lab but that we feel it is beyond the scope of this work at this stage.

4) Alternative signaling systems should be considered. Downstream of Wengen, dTRAF2 could be acting and mediating the effects - right? Is Grindelwald relevant in this context?

As the reviewer points, Trafs are downstream mediators of TNFR. Traf2 has been proposed as the adaptor protein that mediates Egr signaling in Drosophila regulating JNK (Moreno et al., 2002; Xue et al., 2007) and to physically interact with Wgn (Kauppila et al., 2003). Thus, Traf2 could be mediating the role of wgn in tracheal

formation. In our initial submission we already presented results showing that the downregulation of Traf2 (using a validated Traf2 RNAi, Traf2^{GD7146}, VDRC#16125) did not reproduce the defects of wgn loss of function (i.e. excess of terminal cells) (Fig 3B). In this revised version we have extended our analysis of Traf2 and we are now showing that the overexpression of Traf2 does not reproduce the defects of Wgn overexpression (i.e. loss of terminal cells). This result is now shown in the text (lines 211-213) and figures (Fig 3F). These results indicate that Traf2 does not mediate the effects of Wgn in terminal cell differentiation.

The focus of our work in this manuscript was to investigate the role of wgn during tracheal formation, as we identified wgn as a gene required for proper tracheal formation. We did not address the role of other TNFR in tracheal morphogenesis as we considered that this analysis was beyond the scope of this work and could represent, in itself, a topic for a parallel investigation. Nevertheless, we understand the interest of Reviewer #1 and Reviewer #2 in the other known TNFR in Drosophila, Grindelwald (Grnd), and we have performed several experiments to address its possible implication in tracheal development.

First, we analysed the pattern of accumulation of the receptor using antibody staining (antibody kindly provided by Prof Pierre Léopold, (Andersen et al., 2015)). The antibody indicated accumulation of Grnd in the apical membrane of hindgut and foregut (in agreement with the data of in situ hybridisation in BDGP, <https://insitu.fruitfly.org/cgi-bin/ex/insitu.pl>). At the stages of tracheal morphogenesis, we did not detect prominent and clear accumulation of the receptor in the tracheal tissue, however, on occasions we detected a very faint pattern of Grnd in the apical domain of the tracheal epithelium, which is difficult to distinguish from background. We then addressed the functional requirement of Grnd during tracheal development, analysing, with particular interest, the differentiation of the terminal cells. The analysis of grnd mutants did not reveal consistent and reproducible defects in the trachea. To circumvent possible maternal contributions and also possible non-autonomous effects of Grnd affecting tracheal development, we expressed Grnd-extra (which acts as a dominant negative form, (Andersen et al., 2015)) in tracheal cells. Again, we did not find gross defects in tracheal morphogenesis, and branching and branch fusion events proceeded correctly in these embryos. We detected a mild phenotype of extra terminal cells, in line with the defects observed by different loss of function conditions of JNK pathway. The overexpression of wild type Grnd in the trachea did not produce any phenotype.

In summary, we did not detect any clear effect of Grnd in embryonic tracheal formation, although we observed a minor phenotype (in line with effects of JNK) that could be analysed in more detail in a parallel work. We are showing these results in Fig 2 in this document. Because we consider that these results do not provide relevant information related to the role of Wgn in tracheal formation and would add more data that could distract from the main point, we decided not to include these results in the revised version. Nevertheless, in case it is considered necessary, we could add these data.

5) How should we think of the logic of the interaction - does the effect of Wengen depend solely on its expression level, what other effects are mediated. The finding

that JNK signaling does not seem to be active under normal conditions does not at all mean that it is unimportant as it is a classical stress signaling pathway.

The reviewer poses interesting questions.

In this work we find that Wgn is required for tracheal morphogenesis in normal conditions. Our cellular, genetic and biochemical data indicate that this effect depends on the regulation of the activity of Btl, and is not mediated by the JNK pathway. In this specific activity in tracheal terminal cell differentiation, we find that the levels of Wgn play a key role, as in the absence of Wgn we detect an overactivation of Btl activity while in conditions of extra levels of Wgn the pathway is downregulated. This may reflect the molecular mechanism employed that we are proposing, in which Wgn and Btl would form a complex and in this way Wgn would regulate Btl

We did not identify in our analysis other requirements for Wgn during embryonic tracheal development besides the regulation of the number of terminal cells. However, we cannot discard that Wgn also serves other functions that we have not been able to unveil in our experimental settings in normal conditions. As the reviewer points, the TNF pathway has been shown to signal through the JNK pathway, and it is known that the JNK pathway is activated upon stress conditions. Thus, we do not discard that the TNF pathway, through the presence of Wgn receptor in the tracheal tissue, activates the JNK pathway in stress conditions. We have introduced this information in our revised version (lines 228-229, 514-519)

Reviewer #2 (Remarks to the Author):

Authors report a TNFR, Wengen, and its role in Drosophila tracheal terminal branch specification. The authors performed mostly genetic experiments, in combination with a biochemical analysis to suggest a model where Wgn acts unconventionally by apparently interacting with Btl/FGFR. While the idea and some of the observations are interesting, I think the paper mostly reports very preliminary data with several conceptual and technical caveats. A cell biological mechanism is implicated based on just indirect genetic experiments, and a biochemical co-IP assay shown has problems in the experimental design. I believe, more deeper thinking and rigorous/thorough analyses are required to prove/disprove the model proposed. A better writing organization will help the reader. It is unclear what is the broad implication of this study. Conclusions in this paper appear to be claiming a lot more than what is shown. Overall, the results appear to be very preliminary, but with proper experiments, it can reveal an important biology, which controls tracheal terminal branching.

1) A general conceptual block in the paper is that it reads as if the authors are trying to just explain the mechanism of extra terminal branching observed in the Wgn-KO embryo and the this process is not critical to the overall growth of the tracheal system. What is the role of this extra terminal branching? Does it affect the organism/development?

In this work we focused on a role of Wgn that we identified and that was not previously described. We observed that Wgn is required to specify the number of terminal cells in the trachea and we focused on this particular aspect. We investigated the mechanisms by which Wgn regulates this process from a cellular and molecular point of view, and we reported on this analysis. Our analysis led us to propose an unexpected mechanism of activity for Wgn. The TNF/TNFR signalling plays critical roles during development, homeostasis and disease. Thus, we believe that focusing our analysis on the molecular mechanism of Wgn activity in the trachea may be relevant to gain a better understanding of the general mechanism/s of activity of TNF/TNFR.

Wgn regulates the number of terminal cells. Terminal cells are absolutely required for tracheal activity, as they are the cells in charge of oxygen exchange with target tissues. It has been shown that in the absence of terminal cells, larvae die due to inadequate oxygen supply (Guillemin et al., 1996). However, it has not been directly addressed previously what are the physiological consequences of having extra terminal cells. In an attempt to approach this interesting issue that the reviewer is pointing, we have performed a few preliminary experiments. Using hypoxic conditions, we find that larvae in which wgn is downregulated in the tracheal cells die while control larvae survive under the same conditions. This result may seem counterintuitive as one could expect a better performance of the airway tissue with extra terminal branches in conditions of oxygen need. Nevertheless, previous reports have already suggested that extra terminal branches can exhibit abnormal morphologies and arborisation (Guillemin et al., 1996; Jarecki et al., 1999), which could compromise the physiological activity. Thus, a detailed investigation of the physiological implications of having extra terminal cells would be required to understand how the organism is affected. We are encouraged by our preliminary results to undertake this type of analysis in collaboration with a lab expert in hypoxia that we have contacted. Nevertheless, this involves to carry out a whole new project on its own, which would represent a parallel work that we believe is beyond the scope of this manuscript at this stage.

While authors invoke molecular crosstalk between Wgn and Btl/FGFR, the alternative probabilities that can also explain Wgn-KO phenotypes are not carefully examined.

In this work we have addressed what is the mechanism by which Wgn manipulations affect the number of terminal cells in the embryonic tracheal system. The most plausible possibility we considered was that Wgn is activated by the unique TNF known in *Drosophila* (Egr) and that Wgn transduces the signal through the JNK pathway as previously proposed in the literature. We tested this possibility in our first submitted version of this manuscript (lines 133-183 and Fig 1 and 2, initial version). Our results indicated that Wgn is acting, at least largely in part, independently of Egr and the JNK pathway. In this revised version of the manuscript, we have added additional experiments (lines 211-213, 220-224, 246-270, and Fig 3F,K,L; Fig 4D-G,I-L and Fig S2I) that reinforce our hypothesis that Wgn is acting independently of Egr and the JNK pathway.

Because the tracheal defects upon Wgn manipulations affected the differentiation of terminal cells, and the differentiation of terminal cells has been shown to be regulated by the activity of the FGFR signalling pathway, we tested whether Wgn and FGFR interacted at some level. Our work is, therefore, the result of this analysis using genetic, cellular and biochemical approaches.

This is important, as the interactions between Btl and Wgn do not fully explain phenotypes.

We are not sure about which interactions is the reviewer referring because they are not detailed in this particular sentence. We assume that this is a general concern and we will address each particular point in the following responses.

To my understanding and based on the Introduction written, not much is known about TNF/TNFR expression and activity in tracheal development. So, understanding Wgn's role is a new direction in the field. However, the authors did not provide sufficient evidence of TNFs/TNFRs expression/roles in tracheal branching before starting to speculate signaling crosstalk.

Our paper focuses on the role of TNFR Wgn during tracheal formation. As the reviewer points, there were no previous reports about Wgn in tracheal formation (now indicated in lines 89-91 in the revised version). Thus, to investigate these issues, in our initial submission, we analysed its pattern of protein localisation (Fig 3, lines 185-217), the tracheal requirements (Fig 1A-L, Fig S1, and lines 97-131), and its interactions with other tracheal genes required for the same tracheal process (Fig 4, Fig 5A, Fig S2D-H, lines 220-328). Therefore, we provided a comprehensive analysis of the expression and role of Wgn in tracheal formation.

Because the TNF Egr is the only TNF identified in *Drosophila*, in our initial submission we addressed whether Egr loss and gain of function produced a similar phenotype to that of Wgn loss and gain of function, as expected if it was acting as the Wgn ligand. Because the phenotypes were not comparable, and because the genetic interactions were not consistent with this possibility (Fig 1M-O, Fig 2B, and lines 163-183 in our initial submission), we concluded that Wgn acts largely independent of Egr. As our focus was on Wgn role in terminal cell specification, we did not present further analysis of Egr. In this revised version of the manuscript, we are now providing a more comprehensive analysis of Egr during tracheal development, particularly the pattern of localisation (lines 246-253 and Fig 4D-G), additional phenotypic characterisation (lines 257-260 and Fig 4I,J), and additional genetic experiments (lines 260-266 and Fig 4K,L). Our new data further supports that Wgn acts independently of Egr.

Because, as indicated, our focus was the role of TNFR Wgn during tracheal formation, we did not address the role of other TNFR in the process, as we considered that this analysis was beyond the scope of this work and could represent, in itself, a topic for a parallel investigation. Nevertheless, we understand the interest of Reviewer #2 and Reviewer #1 in the other TNFR known in *Drosophila*, Grindelwald (Grnd), and we have performed several experiments to address its possible implication in tracheal development.

First, we analysed the pattern of accumulation of the receptor using antibody staining (antibody kindly provided by Prof Pierre Léopold, (Andersen et al., 2015)). The antibody indicated accumulation of Grnd in the apical membrane of hindgut and foregut (in agreement with the data of in situ hybridisation in BDGP, <https://insitu.fruitfly.org/cgi-bin/ex/insitu.pl>). At the stages of tracheal morphogenesis, we did not detect prominent and clear accumulation of the receptor in the tracheal tissue, however, on occasions we detected a very faint pattern of Grnd in the apical domain of the tracheal epithelium, which is difficult to distinguish from background. We then addressed the functional requirement of Grnd during tracheal development, analysing, with particular interest, the differentiation of the terminal cells. The analysis of grnd mutants did not reveal consistent and reproducible defects in the trachea. To circumvent possible maternal contributions and also possible non-autonomous effects of Grnd affecting tracheal development, we expressed Grnd-extra (which acts as a dominant negative form, (Andersen et al., 2015)) in tracheal cells. Again, we did not find gross defects in tracheal morphogenesis, and branching and branch fusion events proceeded correctly in these embryos. We detected a mild phenotype of extra terminal cells, in line with the defects observed by different loss of function conditions of JNK pathway. The overexpression of wild type Grnd in the trachea did not produce any clear defect that we could detect.

In summary, we did not detect any clear effect of Grnd in embryonic tracheal formation, although we observed a minor phenotype (in line with effects of JNK) that could be analysed in more detail in a parallel work. We are showing these results in Fig 2 in this document. Because we consider that these results do not provide relevant information related to the role of Wgn in tracheal formation and would add more data that could distract from the main point, we decided not to include these results in the revised version. Nevertheless, in case it is considered necessary, we could add these data.

A careful analysis of the autonomous and non-autonomous roles of Wgn and other TNFR and their ligands is a prerequisite to carefully thinking and designing experiments to speculate molecular/cellular mechanisms leading to a phenotype that originates far downstream from these events.

In our initial submission, we already addressed the autonomous-non autonomous requirements of Wgn in the tracheal system (presented in Fig S1G and lines 120-122), which is an important aspect as the reviewer points. We described that the defects of downregulating Wgn in the trachea (using btlGal4 driver that is expressed in the tracheal cells to express WgnRNAi) and of WgnKO mutants in terminal cell differentiation were comparable (Fig S1G). In this revised version we are showing this analysis in the main Fig 1 (Fig 1I), with the corresponding statistical analysis, which was not presented in the previous version. Our results indicate that Wgn is required in the tracheal cells to regulate the number of terminal cells. Overexpression of Wgn also in tracheal cells produce the opposite effect to that of the RNAi, further supporting the autonomous effect of Wgn.

In our analysis of Grnd, we are also investigating the autonomous role of Grnd in the tracheal system, by expressing wild type Grnd and Grnd-extra (detailed in the previous point).

As indicated in the previous point and in the manuscript (lines 92, 235-236, 505-506), TNF-Egr is the only TNF ligand identified in Drosophila. In this revised version of the manuscript we are now providing a more comprehensive analysis of Egr during tracheal development, particularly the pattern of localisation (lines 246-253 and Fig 4D-G), additional phenotypic characterisation (lines 257-260 and Fig 4I,J), and additional genetic experiments (lines 260-266 and Fig 4K,L).

Let me elaborate on some of these above comments going through line-by-line and by Figures.

2) In Line 24, sufficient evidence is not provided that Wgn works unconventionally. Still, Egr expression provided a phenotype. No data was shown of Egr or any other TNF binding/not-binding to Wgn.

Line 24 corresponded to the abstract. Abstracts are limited in words and it is difficult to be more explicit.

We described in the abstract that Wgn works unconventionally because it is required for terminal cell differentiation independently of its ligand and canonical pathway; these results are fully developed in the results section. Results on Egr, the sole TNF ligand in Drosophila, binding or not binding to Wgn are described in the results section (lines 246-253) and shown in Fig 4F,G in the revised version.

3) Line 25 and 26 repeat the concept of Wengen working unconventionally.

The word "unconventional" was used in line 25 of the initial submission. The following sentence was aimed to explain in more detail what we meant for "unconventional". In the revised version we have revised the abstract.

Secondly, it is confusing to read that Wgn represses terminal cell fates in one sentence, and then in the next sentence, it interacts with FGFR to activate terminal cell differentiation.

"Wengen forms a complex with Breathless (an FGFR), which responds to ligand by activating terminal cell differentiation." In this sentence, we meant that Breathless activates terminal cell differentiation.

In the revised version we have revised the abstract to clarify several points.

The abstract does not give any clue of the mechanism or significance, and it appears to report very preliminary observations.

The significance of this work is highlighted in the two first sentences and the last sentence of the abstract.

A general comment: I guess we do not know the exact mechanism that activates terminal cell differentiation (or any cell other fate specification), it would be ideal to

state that a certain genotype is required for the terminal cell fates. A statement like FGFR signaling activates terminal cell differentiation might be wrong, but it is true that FGFR is required for differentiation.

4) Line 68 – citation needed.

The citations of this part of the text can be found at the end of the paragraph (line 80)

Line 73-74 – the reason for investigating TNFR and FGFR needs to be clarified. What is known and what is unknown about the TNF-TNFR signaling in the tracheal is critical for the reader of a different field. Signaling cross-talks are not surprising, but why these two receptors but not others? Why not Hh or Dpp, which are also critical for the dorsal branch migration and terminal fates?

We have addressed this concern in the last paragraph of the introduction in the revised version (lines 82-106).

As indicated, the similarity of the phenotypes of Wgn with those of FGFR manipulations (FGFR is responsible for terminal cell specification in the trachea, (Jarecki et al., 1999; Lee et al., 1996; Nussbaumer et al., 2000; Samakovlis et al., 1996; Sutherland et al., 1996)) prompted us to investigate their possible interactions and cross-talks.

The roles of Hh and Dpp in tracheal formation have been previously described and none of these signalling pathways gives rise to defects in terminal cell differentiation comparable to those of Wgn. Dpp is critical for the fusion fate specification. As a consequence, manipulations of Dpp can give rise to defects in terminal cell specification (Steneberg et al., 1999). In our work we show that fusion cells are normally specified in wgn loss and gain of function, indicating a different activity. Hh has been shown to play a major role in directing the cell extensions that arise from terminal cells, but not to play a major role in terminal cell specification (Kato et al., 2004), which is the event that we investigate in this manuscript.

5) Figure 1: A: and Line 103, I think an introduction of all the tracheal branch names is helpful to general readers.

We agree with the reviewer and we are providing this information in Fig S1A

a. 1B: please provide the significance of terminal branching and what happens if more terminal branching occurs. What does it mean in terms of the organism? Do we have more cells that need oxygen?

Wgn regulates the number of terminal cells. Terminal cells are absolutely required for tracheal activity, as they are the cells in charge of oxygen exchange with target tissues (this is now stated in the revised version in lines 93-96). It has been shown that in the absence of terminal cells, larvae die due to inadequate oxygen supply (Guillemin et al., 1996). However, it has not been directly addressed previously what are the physiological consequences of having extra terminal cells. In an attempt to approach this interesting issue that the reviewer is pointing, we have performed a few preliminary experiments. Using hypoxic conditions, we find that larvae in which

wgn is downregulated in the tracheal cells die while control larvae survive under the same conditions. This result may seem counterintuitive as one could expect a better performance of the airway tissue with extra terminal branches in conditions of oxygen need. Nevertheless, previous reports have already suggested that extra terminal branches can exhibit abnormal morphologies and arborisation (Guillemin et al., 1996; Jarecki et al., 1999), which could compromise the physiological activity. Thus, a detailed investigation of the physiological implications of having extra terminal cells would be required to understand how the organism is affected. We are encouraged by our preliminary results to undertake this type of analysis, in collaboration with a lab expert in hypoxia that we have contacted. Nevertheless, this involves to carry out a whole new project on its own, which would represent a parallel work that we believe is beyond the scope of this manuscript at this stage.

b. Line 116 : quantification is based on only 2 embryos?

As indicated in lines 116-118 in the original submission, quantifications were based in 475 branches in the control and 415 branches in the WgnKO mutants. In this revised version we have added the number of embryos analysed (lines 132-135).

c. Since tracheal terminal branching involves a trachea-extrinsic control of branching from the target cells, and since wgn-KO affects all cells, it is critical to analyze autonomous and non-autonomous effects on trachea to correctly predict a molecular mechanism from the end-point genotype-phenotype analyses. For instance, Fig. 1D - wgnKO shows higher DSRF outside the trachea relative to control. Similarly, in Fig 1G, wgnRNAi expression in the trachea has fewer extra branches (a milder phenotype) and terminal cells in comparison to wgn-KO. There might also be a non-autonomous effect on the trachea, promoting terminal branches in the wgn-KO condition. Alternative non-autonomous effects are not shown.

In our original submission we already addressed the issue of non-autonomous effects, as we agree with the reviewer that this is an important point. The data related to this issue was presented in Fig S1G and lines 120-122.

To address this question, we expressed wgnRNAi (2 independent lines) in the tracheal system and analysed terminal cell differentiation. As it was shown in Fig S1G in the original submission, we found a comparable phenotype to that of wgnKO. In all 3 conditions analysed, wgnKO and the 2 independent RNAi lines, the percentage of branches with 2 or more terminal cells is around 90% and the percentage of terminal cells arising from the stalk is around 30%. In this revised version we are showing these results in the main Figures (Fig 1I), and we are also providing the statistical significance of the phenotypes to further demonstrate this autonomous effect. In particular, we are comparing the proportion of branches showing two or more terminal cells per DB and the proportion of terminal cells in the stalk, as different classes. One of the wgnRNAi lines (UASwgnRNAi-VSH330339) shows a comparable expressivity to that of wgnKO mutants, indicating that wgn is required in the tracheal cells to regulate terminal cell differentiation. The other line (UASwgnRNAi-Trip.HMC03962) shows a milder phenotype, indicating a weaker interference of wgn. From these results we conclude that the role of Wgn in terminal

cell differentiation depends on the activity of Wgn in the tracheal cells (lines 136-138).

We have changed the image in Fig 1G to present a representative example of a wgnRNAi embryo to be compared to wgnKO.

The reviewer refers to the levels of expression of DSRF in tissues other than the trachea. We would like to point that we were using the expression of DSRF in terminal cells to evaluate their differentiation (expression of DSRF=terminal cell, no expression of DSRF=no terminal cell), and we did not pretend to quantitatively analyse the levels in other tissues (nor in the trachea). The fact that the downregulation of wgn in the tracheal cells reproduces the defects of wgnKO in terminal cell differentiation indicates that, even in the case that there were differences in DSRF expression outside the trachea in wgnKO mutants, these would not cause the effect in terminal cell differentiation.

d. Fig. 1D, where is the dorsal trunk? Similarly, 1G, has a dorsal trunk broken in one place.

Authors mentioned that the early development is not affected by wgn-KO, but these images might suggest an effect and an important role of TNF/TNFR

In the original submission, Fig 1D showed a ventral vision of the embryo and the dorsal trunk (which is more dorsal) was not acquired. We fully agree that the image can be misleading. Thus, in our revised version we have changed the image in Fig 1D to show a comparable region to that of the control in Fig 1C.

In the original submission, Fig 1G showed an embryo with a non-broken dorsal trunk. However, the image was also misleading. The image showed a projection of several confocal sections. We normally focus in those planes that are necessary to show the terminal cells in the dorsal branches. Because the dorsal trunk is normally convoluted and in a different plane to that of the dorsal branches, the planes that would complete the dorsal trunk were not included, giving the impression that there was a break. In our revised version we have changed the image in Fig 1G to show a representative embryo with an unaffected dorsal trunk.

e. Line 122: "tracheal cells to regulate terminal cell differentiation". I think a statement like --'regulate terminal cell number' is more directly based on the data. Authors should avoid speculative sentences, especially on mechanisms, like wgn regulate differentiation. Based on the picture presented, wgn caused an increase in terminal cell number.

We have modified the text in our revised version

f. No mention of Fig. 1M-O in the text.

Figures 1M-O were mentioned in the text in our first submission (in lines 169-180).

In this revised version we have reorganised the text in the results section. Because we have added more data related to Egr, we are now providing a Figure that

contains all data related to Egr (Fig 4, lines 232-270). Previous Fig 1M-O can now be found in Fig 4A,C,H

g. Line 125: highly penetrant phenotype – please provide justification for 'highly' with quantitation.

The quantification of the phenotype of the overexpression of *wgn* was provided in FigS1H in our original submission.

In this revised version we are indicating this in the text (line 139-142), and we provide statistical analysis in Fig1J.

h. A conceptual problem that Figure1 presents - at this embryonic stage shown in figures, tracheal cells do not divide. So, if multiple terminal cells appear, where are they originating? More terminal cells could be due to the re-activation of cell division of a mother terminal cell or differentiation of the existing cells, or could it also be an early specification of terminal fates which get reorganized later into branched patterns? A number count of all cells in a branch would be informative. It is also important to examine an alternative possibility if *wgn*-KO can increase terminal cells by suppressing apoptosis of extra terminal cells that might be there before and that are not normally detected in wild type genotype-phenotype analyses. It is also possible that the apoptosis is blocked in the terminal branch recipient cells to which the branch will adhere, thereby stabilizing the extra branch that we can see now. Since *wgn* is involved in apoptotic pathway, this point needs to be clarified.

This is an interesting point that we have addressed experimentally.

We first analysed whether we could detect extra cell proliferation in the tracheal system. As the reviewer points, tracheal cells do not divide after stage 11 (Kondo and Hayashi, 2013). We used anti-phospho-Histone3 (pH3) as a marker for cell proliferation. We did not detect expression of pH3 in the tracheal cells at the stage when DSRF starts to be detected (stage 14), neither at earlier (stage 13), nor at later stages (stage 15). Thus, we find no extra cell proliferation in the trachea in *wgn* loss of function conditions, indicating that the extra terminal cells do not arise from division of a mother terminal cell. This information has been added in this revised version (lines 146-148, FigS1H).

To understand the origin of the extra terminal cells, and as suggested by the reviewer, we have counted the number of cells in DBs of metameres 4, 5 and 6 in control and *wgn* mutant embryos. We used *btlGal4>UASNLS-cherry* to visualise the nuclei for cell number quantification. Quantification of cell number did not reveal significant differences between control and *wgn* mutants (lines 148-150, Fig S1I). As we show in Fig 1K-N, we find that fusion cells are normally specified in *wgn* mutant conditions (lines 144-146, Fig 1M). Thus, the results indicate that the extra terminal cells derive from stalk cells that now acquire the terminal cell identity, expressing DSRF and forming extra terminal branches (lines 150-151).

The reviewer suggests another interesting possibility for the origin of extra terminal cells: those would arise from a *wgn*-dependent suppression of apoptosis of a pool of terminal cells that normally disappear in the wild type. We have also evaluated this possibility. First of all, we want to point out that in the wild type we do not detect

more than one terminal cell per DB at early stages, a pattern that would then be refined to a single terminal cell per DB at later stages. It has been previously shown that apoptosis has a certain contribution to tracheal morphogenesis (Baer et al., 2010). Thus, we tested whether blocking apoptosis has any effect in the differentiation of terminal cells by expressing p35 in the tracheal system. We found no significant increase of terminal cells compared to the control (6,8% of dorsal branches presented 1 extra terminal cell in *btlGal4>UASp35* embryos, n=190 branches, from 24 embryos, versus 7,0% in *btl>srcGFP* control embryos, n=799 branches from 71 embryos). This result indicates no specific contribution of cell death to terminal cell number. We did not add these experiments with p35 in our revised version because our experiments with cell counts already indicate that extra terminal cells arise from a change in cell identity.

As the reviewer indicates *Wgn* has been proposed to induce cell death in different cellular contexts (Kauppila et al., 2003). Thus, we also evaluated whether *wgn* induced cell death in our experimental model in the trachea. We compared the results of the overexpression of *wgn* (and *egr*) and the activation of JNK, which is also known to induce cell death in different cellular contexts. The expression of a constitutive form of *hep* (*hep^{CA}*) clearly promoted cell death in tracheal cells, as indicated by the expression of apoptotic markers and presence of apoptotic cells. In contrast, we did not detect these effects upon overexpression of *wgn* or *egr*. These results further confirm our hypothesis that *wgn* does not act through the JNK pathway during tracheal formation and that it does not regulate the number of terminal cells by regulating cell death. These results have been added to the revised version (lines 220-224, 257-260 and Fig 3K,L, Fig 4J)

I think a rigorous analyses of expression of TNFs and TNFRs and their binding (like shown for *Btl-Bnl*) needs to be verified. There are reagents available to do so. As nothing is known about TNF-TNFR in trachea, this characterization is critical before speculating complex mechanisms and interactions between the *wgn* and other pathways.

In our initial submission, we presented a comprehensive analysis of the expression and functional requirements of the TNFR-*wgn*.

We have now extended our initial characterisation to the unique TNF known in *Drosophila*, *Egr*, in relation to the tracheal system and to *Wgn*. We are now providing a more comprehensive analysis of the functional requirements of *egr*. In addition, we also provide analysis of the pattern of accumulation of *Egr* in relation to the trachea and to *Wgn* pattern. The new results further suggest that *Wgn* is required for terminal cell differentiation independently of its ligand *Egr*. These additions can be found in the revised version (lines 232-270, Fig 4).

As discussed in a previous point, we also evaluated the possible involvement of *Grnd* in tracheal formation (in Fig 2 in this document).

5. Fig.2 and related text:

a. Fig. 2B: significance of stats to be indicated.

In the revised version we provide the statistical analysis in Fig 3B (former Fig 2B), comparing the different conditions of JNK downregulation to the control and to the downregulation of Wgn in the tracheal cells.

b. “Downregulation of the pathway using different tools (bskDN, Tak1DN, hepRNAi, Traf2RNAi, or UASpuc) did not reproduce the TNFR-wgn loss of function phenotype.” – it is possible that wgn-KO condition is due to non-autonomous effect. The comparison between an entire embryo wgn-KO with tissue-specific knockdown of JNK signaling is inaccurate. Authors should compare KO of some of these genes with wgn-KO. Alternatively, compare all the conditions with btl>wgn-RNAi. Please examine thoroughly where else wgn is expressed and whether wgn can be complex with ligands.

As we have indicated in a previous response, we found that the role of Wgn in terminal cell differentiation depends on the activity of Wgn in the tracheal cells (lines 136-138, Fig 1I). Because the defects in terminal cell differentiation are comparable between wgnKO and btl>wgnRNAi, in our initial submission we used the results of wgnKO to compare to the conditions of JNK downregulation in the tracheal system. Nevertheless, we fully agree with the reviewer that this is not accurate. Therefore, we have corrected the graph (Fig 3B) to show comparisons of btl>wgnRNAi to the rest of the conditions.

c. Authors mentioned already focusing on a dorsal branch for simplicity, but why refer to another branch GB repeatedly without proper quantitation and reason?

This was meant to show that the effects observed in the dorsal branches were reproduced in other branches, as one would expect from our model. We think this is a valuable piece of information.

d. Since authors are talking about an interaction of FGFR and TNFR within the tracheal cells, they clarify that the effect they see in wgn-KO is not due to an effect from outside the trachea. Many of the results (.e.g, Fig 2J; can be explained by non-autonomous effects in the trachea). Expression of Egr in the trachea inducing more branches could be due to the non-autonomous effect from cells that received Egr.

We have addressed the point of the cell-autonomous/non cell-autonomous effects in previous answers. We show that downregulating Wgn in the tracheal cells produce the same defects than removing Wgn in all the embryo, indicating that Wgn is autonomously required in the tracheal cells to regulate the number of terminal cells (lines 136-138 Fig 1F,G,I).

We agree with the reviewer that theoretically the effects of Egr expression in the trachea could be due to a non-autonomous effect, however, we believe this is unlikely because we show that Wgn is required in the tracheal cells to regulate terminal cell number (RNAi experiments, lines 136-138 and Fig 1F,G,I). Remarkably, we also find that the phenotype of extra terminal cells produced by Egr expression in the trachea can be suppressed by co-overexpressing Wgn, strongly supporting that the effect of Egr in terminal cell specification is mediated by the Wgn activity in the tracheal cells (lines 263-266 and Fig 4K,L).

e. The way phenotypes are described is confusing. Based on the control image of DB, all branches robustly have 1 terminal cell, but more than 1 terminal cell in conditions like *egr* expression of *bskDN* is termed as not effective. They might not completely phenocopy a whole embryo KO phenotype, but they clearly produced more terminal cells than normal. In genetic interactions, it might be important not to focus on milder phenotypes, but the minor phenotypes often can give very important ideas of a molecular mechanism.

We apologise if our descriptions were confusing. We have tried to clarify the text (lines 201-207)

There is a certain variability in the number of terminal cells that form at the tips of dorsal branches. This is clearly evidenced by the minor effect on terminal cell differentiation shown in control conditions (Fig 3B). As indicated in the text, we detect a certain phenotype in terminal cell differentiation in different conditions of downregulation of JNK pathway or in *egr* mutants compared to control. However, these effects are clearly different from those of *wgn*KO or *btl*>*wgn*RNAi. As we indicate in the text, these quantitative and qualitative phenotypic differences strongly suggest that TNFR-Wgn regulates terminal cell differentiation independently of its ligand TNF-Egr and canonical JNK pathway, although we cannot completely discard a minor contribution, as we also indicate (lines 267-270). The conclusion that Wgn acts largely independent of the JNK and Egr is not only based on the gain and loss of function phenotypes observed, but also on genetic interactions and pattern of expression (for Egr) or pattern of activation (for the JNK pathway). In this work we have focused on the effect of Wgn on terminal cell differentiation that is independent of the canonical ligand and signalling cascade

f. Fig 2K/L: *wng*-flag expression reduced DSRF expression, but we still see the extra branch in L popping out from the dorsal trunk, and we see the abnormal fusion. Why is it so?

We assume that the Reviewer is referring to Fig 1K/L

The terminal cell that seemed to pop out from the DT was actually from another plane that in the stack projection seemed to arise from the DT. We have improved the image by removing the confocal planes that did not correspond to the DT and DB.

The fusion process is not a synchronised process, and some branches fuse slightly earlier than others. In Fig 1J-K some contralateral dorsal branches have already fused while others are in the process. However, we show that all fusion cells are specified in control, *wgn* mutants and *wgn* overexpression conditions, anticipating the fusion event.

Line 177-183: It is better not to assume what is physiological or not based on just knock-out and overexpression analyses. Wgn can have multiple roles through multiple pathways. However, classifying one as physiological but not the other based on phenotypic analyses of gene perturbation can potentially accumulate wrong information or the future. Secondly, if excess *egr* sequesters Wgn, an overexpression of *wgn* should suppress overexpression phenotypes. Show Egr/other TNF binding to TNFR, as the authors suggest that the binding is not critical for the

wgn-KO-induced phenotypes.

We agree with the reviewer and we have removed the term physiological. We performed the experiment that the reviewer suggested and we obtained the result that the reviewer was pointing, a suppression of the phenotype of Egr tracheal expression. We have added this experiment in the revised version (lines 263-266 Fig 4K,L).

As indicated in the text, TNF-Egr is the only TNF ligand identified in Drosophila. As indicated previously, we have addressed the contribution of Egr to terminal cell specification and its pattern of accumulation in more detail in the revised version (lines 232-270, Fig4).

Line 185: Please mention how native Wgn is detected in the text. How specific is this Wgn antibody, and which part of Wgn is probed? Why is Wgn not in other tissues in Fig. 3A, unlike what is mentioned in the text? This might be important.

Wgn is detected by immunofluorescence. We have added this information in the text (line 161). The antibody was generated (and kindly provided) in Dr. Konrad Basler's lab, and has been validated in several papers (Andersen et al., 2015; Palmerini et al., 2021).

Figure 2A shows a region focused on the tracheal system, so it is more difficult to see the pattern outside the trachea. Orange arrows in Fig 2A',B',I' point to expression of Wgn outside the trachea. In addition, Wgn pattern outside the trachea can also be seen in other Figs (Fig 4F,G, Fig 5G, Fig S2D, Fig S4B).

4) Fig.3A:

a. the ROI box on the trachea is not in focus. More importantly, please show a comparable stage with terminal cells turning away from the fusion cell. Wgn expression can be stage-specific as early-stage wgn-ko embryos have normal development of most primary branches.

The ROI in Fig 2A is in focus, but because it is a projection it may seem slightly blurred. Fig 2A shows a stage 13, as indicated in the Figure legend. At this stage no terminal cells turn away from fusion cells. The stage at which this happens is shown in Fig 2B.

Note that endogenous receptors can be faintly localized on the membrane, and often the strong signal in the intracellular vesicles under confocal can mask the detection of faint cell surface localization. Since this is the first report of TNFR expression in the trachea, better imaging (e.g., airyscan, if available, or non-permeabilized Wgn staining for surface distribution) might be critical for this paper as it talks a lot about the difference between surface and intracellular distribution of the receptor.

Our results with Rab5DN show that we can detect Wgn at the membrane under certain conditions. Therefore, if we do not detect it in wild type conditions it likely indicates that its accumulation at the membrane must be negligible compared with the pattern in vesicles.

It is worth pointing out that in other publications it has also been shown that Wgn accumulates in intracellular punctae and not at the membrane in other tissues (Palmerini et al., 2021).

The experiment the reviewer proposes of permeabilisation would be very interesting but it is not feasible in our case. The trachea is an internal structure in the embryo and therefore it is not exposed at the surface for such evaluation.

b. Also, please mark the cell/tissue outlines and focus on the parts required to be shown. This is important because the authors claim that Wgn apparently has little function in membrane-localized forms.

We have tried to outline the tracheal system in Fig 2A,B without compromising the information, as outlining the trachea may mask possible accumulation of Wgn at the membrane. We hope that the insets and arrows help to visualise Wgn protein in the tracheal system.

c. In Fig. 3B,C dorsal trunk, wgn is more in the fusion cells? Fig 3F, Rab7 is everywhere, and specific colocalization is hard to predict.

Rab7 associates with late endosomes and typically shows this pattern in punctae, which unfortunately makes difficult the analysis. To show another marker of late endosome stages we also provide Hrs in Fig 2E
Fig 2B shows endogenous Wgn while Fig 2C (3B,C in original submission) shows overexpression of Wgn (detected with anti-flag). In these stainings we did not counterstain with a fusion marker. However, we have not noticed in these or in other stainings higher levels of Wgn in the fusion cells of the dorsal trunk, and we have not detected branch fusion defects.

d. Fig.3I, a clear increase in membrane Wgn with Rab5DN condition and membrane localization of Wgn correlated with its loss of terminal cell repression activity. However, a control expt must be done. Is this a condition-specific for Wgn, or all membrane proteins, including Btl/BMPR, or anything else expressed in these branches? Internalization of all these receptors is critical for canonical signaling pathways. A recent paper suggests that Btl in membrane-localized form can also act as a cell adhesion molecule for the GPI-anchored Bnl and the CAM activity (which does not require activation of canonical MAPK) is required for terminal branching (Du et al. 2022). So, all these different pieces of information suggest that we have a limited understanding of terminal branching and to clearly understand the role of TNF/TNFR's in this, a thorough analysis of these new signaling proteins is a pre-requisite.

Here we investigate the differentiation of terminal cells. It is known that terminal cell differentiation depends on Btl activity (Lee et al., 1996; Nussbaumer et al., 2000; Sutherland et al., 1996). Thus, the relevant proteins to analyse in this context are Btl and Wgn. The reviewer is clearly right in arguing that the Rab5DN condition may affect many membrane proteins, as it is a general regulator of endocytic internalisation. However, we think this does not hamper the value of our experiment based on the following facts. 1- In the Rab5DN condition terminal cells form indicating that the Bnl-Btl signalling is active. 2- The overexpression of wgn is able to suppress terminal cell formation. 3- However, the overexpression of wgn in the Rab5-DN background does not suppress formation of terminal cells. Thus, directly or

indirectly, it is the Rab5DN condition that renders the overexpression of wgn not effective. As the double condition of overexpression of wgn/Rab5DN is associated with the localisation of Wgn at the membrane, we think we can claim that this experiment suggests that retention of Wgn at the membrane correlates with its loss of terminal cell repression.

e. "...traffics through the endocytic pathway and it is then degraded or recycled back to the membrane." - Authors previously mentioned Wgn, not in the membrane, but now concluded that Wgn recycled back to the membrane.

We apologise if we did not make this point clearer. As we were indicating in our original submission, our results suggested that Wgn has some ability to localise to the membrane. However, we described that in wild type conditions we do not detect it at the membrane.

We have revised the whole chapter to clarify these points (lines 159-193).

"...traffics through the endocytic pathway and it is then degraded or recycled back to the membrane." Is this just based on lysosomal and Rab4/7 localization? Where is strong evidence for degradation or recycling?

We detect Wgn colocalising with markers for different endocytic compartments, including the sorting and recycling endosomes. For a transmembrane receptor, this is considered strongly indicative of trafficking through the endocytic pathway. However, we agree with the reviewer that these observations do not conclusively prove this hypothesis. That is why we show that Wgn localisation changes upon compromising endocytic internalisation. This approach has been broadly used in Cell Biology to demonstrate endocytic recycling of transmembrane receptors.

We have clarified this point in the text (lines 173-186).

f. Wgn clearly is enriched at the branch tip cells compared to other cells in a branch. Authors should focus and display images properly. Fig 3E-H; which branch are we looking at, and what is the outline of the trachea?

Images in Fig 2E-H (Fig 3E-H in the original version) show dorsal branches. We have added this information in the figure legend. We have outlined the dorsal branches as suggested by the reviewer.

g. "Altogether, these results show that TNFR-Wgn is a highly dynamic protein that reaches
216 the membrane but is constantly internalised, preventing it to stably localise at
the
217 membrane."

h. Authors repeatedly suggested dynamic activities, but no data was shown to support dynamism. Dynamism is inferred from just fixed-tissue localization in some specific compartments.

The reviewer is completely right with this comment. We are not providing data on dynamism. We can only propose that the protein is trafficked, but we cannot imply

aspects related to dynamisms. We have addressed this point in the text throughout all the manuscript.

5) Fig. 4:

a. Since dpERK is a probe for terminal cells, even if Wgn-KO removes the terminal cell by an alternative route, it will not show dpERK-positive cells, as shown in Fig 4. How could authors conclude that Wgn-KO affected Btl-Bnl signaling without eliminating the other possibilities?

WgnKO does not remove terminal cells, it actually leads to extra terminal cells, as we show in Fig 1 and S1. In WgnKO conditions we observe extra terminal cells (as visualised by the expression of DSRF, Fig 5B) and these extra terminal cells also accumulate dpERK. Conversely, the overexpression of Wgn leads to lack of terminal cell and we do not detect cells at the tip of the branch accumulating dpERK (Fig 5C). The Bnl/Btl pathway is known to be the sole contributor for ERK-MAPK activation at the tips of the migrating tracheal branches (Gabay et al., 1997), and dpERK accumulates in the nuclei of the terminal cells upon Btl activation. dpERK enters the nucleus and activates the terminal cell program in the tip cell. Because we find that Wgn regulates dpERK accumulation, we propose that Wgn regulates Btl activity (lines 273-289).

b. Line: 244 – cite these reagents for validation.

We are now citing these reagents in the text and materials and methods

c. Fig. 4H: IP did with overexpression of both proteins in the salivary gland. Overexpression can cause false positive co-IP. This might not indicate a physiological interaction. Why is the salivary gland used? It is a non-essential organ in larvae.

Secondly, no clear idea of what is the IP control in panel H. Is it the membrane GFP co-IP with Wgn? Can there be a control overexpression membrane protein that does not co-IP with Wgn?

Unfortunately, co-IP assays cannot be performed with endogenous proteins in the trachea from total embryo protein extracts because their levels are too low. In addition, the Wgn antibody that we used in this work was a gift from the K.Basler lab, and we do not have enough antibody to perform co-IP experiments. We tried to produce our own Wgn antibody but we did not succeed. Therefore, we performed co-IP assays in salivary glands expressing wgn-flag and btl-GFP since we can isolate the tissue and obtain sufficient purified protein extracts to perform biochemical experiments. To avoid strong overexpression with the Gal4/UAS system, crosses and larvae were maintained at 22 °C.

Control in Fig 5H,I is the IP with a non-related antibody of the same isotype (here mouse IgGs), which is a negative control usually used in co-IP experiments to show that the protein is not non-specifically retained by the beads or IgGs.

Protein extraction was carried out in RIPA buffer, a highly stringent buffer that contains non-ionic (triton X-100) and ionic (sodium deoxycholate and SDS) detergents which minimize non-specific protein-protein interactions.

As the referee suggested, we have also performed control co-IP experiments with overexpression in the salivary glands of two other membrane proteins, DE-Cadherin and Sidekick, and we show that Wgn does not co-immunoprecipitate these proteins. These results indicate that the overexpression of Wgn and another membrane protein does not cause unspecific co-immunoprecipitations. These results have been added to the revised version (Fig S2G,H).

To find further evidence that Btl and Wgn form a complex, we performed new experiments that we are providing in this revised version. We used the trachea to perform PLA (proximity labelling assay) experiments. We found the presence of PLA spots in experimental conditions (expression of Wgn and Btl) and not in control conditions (Fig 5J,K). The analysis suggests that the two proteins reside at a maximum distance of 40 nm, consistent with our co-IP experiments and with the hypothesis that they participate in the formation of a complex (lines 225-229).

Salivary glands normally do not have tracheal supply and Btl/Bnl expression. Does it express TNF/TNFR? Do we get any overexpression phenotype due to Wgn expression in the salivary gland?

We have analysed salivary glands and we have detected low expression of Wgn. We have also analysed salivary glands that overexpress wgn (*fkhGal4>UASwgn-flag*) stained for an Adherens Junction marker (E-Cadherin) to visualise the apical domain of the cells. We did not detect any clear effect in salivary glands. Actually, we find that flies carrying *fkhGal4>UASwgn-flag* are viable. These results can be found in Fig 3 in this document

d. Fig S2H is important data and needs to be in the main Figure, and the graph should be accompanied by a representative Figure panel. Why are authors not showing dpERK (activity) if the concern is Btl activation? If the hypothesis presented in this section is correct - Do you see Wgn localization on the membrane and more Btl in the cytoplasm (required for MAPK) when is Egr overexpressed? I think the data here is not strong enough to be confident about the model proposed.

We have moved Fig S2H to the main figures (Fig 5N in the revised version). We have also added panels showing a representative example of Wgn and Btl accumulation in control and in *btl>egr* conditions. As it can be observed, we detected Wgn in vesicles. Thus, Egr is probably interfering with Wgn activity once they are internalised. It has already been shown that Egr can be internalised (Ruan et al., 2016).

As suggested by the reviewer, besides showing that *btl>egr* leads to extraterminal cells as assessed by DSRF expression (Fig 4H), we are now also showing that it leads to dpERK accumulation in these extra cells (lines 335-338, Fig S2I), indicating the activation of Btl.

e. Fig. 4K-L, is very important and needs to be supported by proper image panels. Secondly, the no. of *btl* puncta or levels could simply be a function of no. of cells in the tip or stalk or their size/shape/amount of membrane surface. A normalization/cell would be helpful to conclude.

Representative image panels for Fig 6A (former 4K) are provided in FigS3A-C. We used normalisation as described in materials and methods. A normalisation per cell would be very complicated because all cells express *btl*GFP and it is difficult to determine the exact extent of each individual cell.

f. Authors must examine whether the effect of Wgn is specific to Btl by examining control receptors/proteins, which might remain unchanged. Conclusions on degradation and so on are just based on Arl8 localization. Degradation needs to be verified with biochemical assay if this is important.

We fully agree with the reviewer that, to claim that Wgn promotes degradation of Btl, stronger biochemical evidence should be provided. In this manuscript we do not conclude, but suggest a role for Wgn in regulating Btl trafficking and degradation (lines 361-363). This suggestion is not only based on Arl8 localisation, but also, importantly, on the analysis of the levels of Btl in different *wgn* conditions. Thus, in conditions of Wgn overexpression we detect an increase of Btl vesicles positive for Arl8, which correlate with low levels of Btl and fewer Btl vesicles.

The reviewer asks to investigate whether the effect of Wgn is specific to Btl. We do not state in our manuscript that Wgn only affects Btl. Wgn could also affect the trafficking and/or degradation of other proteins. However, here we were asking how Wgn is regulating the number of terminal cells, which is controlled by Btl activity. In this regard, our genetic experiments (using *UASWgn;UAS λ Btl* and *wgn;UASbtlRNAi/bnl00857*, Figs 6F-K and S3D-G) and analysis of dpERK accumulation (Fig 5A-C) strongly indicate that Wgn regulates the number of terminal cells by regulating Btl activity. For this reason, we analysed a possible effect on degradation of Btl. Investigating whether the trafficking or degradation of other proteins is affected by Wgn is a very interesting project, which we believe is beyond the scope of this work.

h. Fig S2, authors must focus on showing the same stage of embryos where there are terminal cells differentiating from fusion cells and also to the point of vesicles and membrane localization wrt to terminal cells. I guess these Btl reagents are published (please cite) and well-verified, and showing a validation of these reagents is less important than validating Wgn localization/expression (which needs to be reinforced with data presented in the paper).

The images that the reviewer is asking are presented in Fig 2B, Fig 5G, Fig 7A,C (previously Figs 3B,4G and 5D in the original submission), so we consider that the pattern of Wgn expression in the trachea is extensively addressed in the manuscript. We are now citing these reagents in the text and materials and methods.

i) Line 301-315: In a previous section, authors already showed that RasV12 induced ectopic DSRF positive terminal cells, and this cannot be suppressed by Wgn overexpression, so Wgn must act in parallel to Ras pathway or upstream between FGFR and Ras. Since activated Btl has no ligand binding domain, and the intracellular Kinase domains are permanently dimerized, so constitutively inducing Ras-MAPK signaling, it is not surprising that the const. activated Ras pathway is not

suppressed by Wgn overexpression. All these phenotypes collectively suggest a parallel pathway of FGFR and Wgn, but not clear-cut interactions between the two pathways,

As the reviewer points, we indicated in our initial submission that Wgn may act in parallel or upstream of Ras, because it cannot suppress the effect of Ras activation. Different results, however, suggest that Wgn may act at the level of Btl receptor. First, our data indicate that Wgn and Btl form a complex. We also find that Wgn manipulations affect the accumulation of dpERK, a key MAPK signalling component, which is activated by Btl in terminal cell differentiation. Our genetic experiments reinforce our hypothesis that Wgn acts regulating Btl. In particular, we observe that the downregulation of Btl activity can revert the defects of wgn loss of function (lines 368-381, Fig 6F-H, S3D-G), and that the co-overexpression of wgn and λ btl rescues the lack of terminal cells of wgn overexpression and partially reverts the effects of λ btl (lines 392-400, Fig 6I-K). We consider that the results of these genetic experiments are consistent with our model.

j) Btl:GFP overexpression can induce MAPK signaling and tracheal responses shown by other groups. It is also known that Btl level and localization to the membrane are dependent on Bnl signaling, so overexpression of Btl:GFP did not cause an extra terminal branch, but overexpression of Bnl did seem to be not matched with any of the prior reports on Btl-Bnl signaling and the existing model. It sounds like the authors are suggesting that overexpression of Btl:GFP did not activate signaling.

The line UASBtl::GFP that we use in our manuscript has been used in 8 previous publications (Dammai et al., 2003; Fan et al., 2020; Hiramatsu et al., 2019; Hsouna et al., 2010; Mortimer and Moberg, 2009; Sato and Kornberg, 2002; Wang et al., 2010; Yan and Lin, 2007). It has been shown that this line can rescue the lack of endogenous btl (Yan and Lin, 2007), indicating that a functional Btl protein is produced. However, in none of the publications it is shown that Btl::GFP overexpression produces an excess of terminal cell number. Here we investigated this issue and we find that Btl::GFP overexpression does not lead to extra terminal cells. This suggests that either Btl::GFP is expressed at too low levels to promote extra terminal cells, and/or that the levels of Btl are not limiting to regulate terminal cell differentiation but the levels of another factor (e.g. the ligand bnl) are limiting (discussed in lines 481-491)

Interestingly, in two previously published papers it is stated:
Dammai et al., 2003: "In the wild-type genetic background, the Btl-GFP fusion protein is barely detectable in the tracheal cells (Fig. 5A), showing that expression of Btl-GFP by itself does not lead to overaccumulation of the fusion protein. This indicates that the expression level of btl-GFP is likely under the same stringent feedback control as for the endogenous btl (Ohshiro and Saigo 1997). Significantly, at high magnification Btl-GFP-containing particles are seen internalized in all tracheal cells of wild-type embryo (Fig. 5A–C), consistent with the notion that active internalization from the membrane domain prevents surface accumulation of Btl-

GFP. This is further exemplified by lack of tracheal abnormality in wild-type embryos expressing Btl-GFP."

Hsouna et al., 2010: "A *btl::GFP* fusion protein gene was expressed from the *UAS* enhancer under the control of the *btl* promoter-driven *Gal4*. As shown in Fig. Fig.66 A, in wild-type (*1-eve-1*) embryos, Btl::GFP is expressed at a low level on the cell periphery, in contrast to the high-level, generalized cytoplasmic expression of *UAS-GFP* from the same *btl-Gal4* driver (Fig. (Fig.6A').6A'). This indicates that the posttranscriptional expression level and localization of the receptor are tightly regulated."

The results we present with overexpression of Bnl fully match with all previous literature on the subject: Bnl overexpression leads to extra terminal cells (Gervais and Casanova, 2011; Nussbaumer et al., 2000; Steneberg et al., 1999; Sutherland et al., 1996).

5) Fig 5:

a. Fig 5D, I do not understand if a Btl:GFP or Btl overexpression alone did not show an extra terminal branch. Why did authors need to examine something upstream to Btl? It is important that the authors focus on the cross-talks based on the available BtlDN (lacks intracellular kinase domains), loss of Btl (RNAi clones), and gain of Btl (clones). E.g. *btl*-RNAi clone and see what happens to Wgn. Unless there is an increase in the number of Bnl source cells in *wgn* KO condition (which is not verified), that can extrinsically control terminal cell fates. Also, a direct measure of MAPK signaling is required if activation of Btl is the critical component in the pathway predicted. A Btl-Bnl endosome might not always mean signaling activation. So a direct measure is more important than an indirect measure to prove an outstanding claim.

As requested by Reviewer #2 and #3, we have analysed whether *wgn* is affecting the pattern of Bnl, which could account for the phenotypes observed. Our results indicate that the pattern of Bnl, analysed using *bnl-lacZ*, is comparable in control and mutant embryos, and quantitative PCRs indicate no significant differences in Bnl transcriptional levels in control and *wgn* mutants (lines 294-298 and Fig S2B-D). These results, combined with the results where we show that downregulation of *wgn* in tracheal cells reproduce the same effects in terminal cell specification than those observed in the *wgn*KO mutants, indicate that Wgn is autonomously required in the tracheal cells to regulate terminal cell number.

In our manuscript we show a direct measure of MAPK activity upon Wgn manipulations using dpERK (Fig 5A-C)

We apologise if our descriptions of the effects of Btl-GFP overexpression were confusing. In this revised version we have tried to clarify the results (lines 381-389, Fig S3I) and the text (lines 487-491) to show that Btl-GFP overexpression does not lead to extra terminal cells.

As suggested by Reviewer #2 and #3, we have performed new genetic experiments. In our initial submission we were already showing interactions between

overexpression of *wgn* and λ *btl* (lines 392-400, Fig 6I-K). We have now tested whether decreasing *btl* activity reverts the effects of *wgn*. We tested 3 different conditions to approach this question: *wgn*KO; *btl*DN; *wgn*KO;*btl*RNAi and *wgn*KO;*bnl*⁰⁰⁸⁵⁷

In our hands, the expression of *btl*DN (Reichman-Fried and Shilo, 1995) in the embryonic tracheal system did not produce any detectable phenotype, in branching or in the specification of terminal cells, suggesting that this line is not sufficiently active in our experimental set-up. In agreement with this low activity, we found that *btl*DN was not able to suppress the effects of *wgn*KO mutants and extra terminal cells were detected.

In contrast, we found that the expression of 2 independent *btl*RNAi lines (namely BL-60013 *y v*; P(Trip.HMS05005) attP40 and BL-40871 *y sc v sev*; P(Trip.HMS02038)attP2) in the tracheal system produced an extreme phenotype of lack of terminal cells, indicating that no sufficient levels of *btl* are present in these conditions to activate the pathway to form terminal cells. We found that the expression of *btl*RNAi reverted the phenotype of *wgn*KO. In this case, we compared and quantified the number of terminal cells in *wgn*KO; *btl*Gal4>*lacZ* mutants and in their siblings *wgn*KO; *btl*Gal4>UAS*btl*RNAi. The suppression of the phenotype was very clear and consistent.

bnl has been shown to be an haploinsufficient locus (Jarecki et al., 1999; Sutherland et al., 1996). We tested the effect of reducing 1 copy of *bnl* quantifying the presence of terminal cells in the dorsal branches in *bnl*⁰⁰⁸⁵⁷ (an amorphic allele) heterozygotes. We found that most dorsal branches lack the terminal cell. Thus, we then tested whether reducing the dosage of *bnl* reverted the *wgn*KO phenotype. We compared and quantified the number of terminal cells in *wgn*KO;*lacZ*/+ mutants and in their siblings *wgn*KO; *bnl*⁰⁰⁸⁵⁷/+. The suppression of the phenotype was very clear and consistent. The reduction of *bnl* decreases the activity of the pathway because less receptor is activated (as the ligand is limiting, (Sutherland et al., 1996)). Interestingly, in this case, we found that reducing the dose of *bnl* mostly reverted the *wgn* phenotype to a wild type number of terminal cells (1 terminal cell per DB), while reducing *Btl* levels by RNAi produced mostly absence of terminal cells. These results suggest dosage-sensitive interactions.

We have added the results of *btl*RNAi (line BL-60013) and *bnl*⁰⁰⁸⁵⁷ in the text and figures (lines 369-381 and Fig 6F-H and S3D-G).

We believe that these genetic interactions further strengthen our hypothesis that *Wgn* regulates the number of terminal cells through the regulation of *Btl* activity.

b. Fig. 3H, All the conditions, like overexpressing a highly potent ligand like *Bnl* in the trachea, destroy tracheal branch growth. So, not sure what the authors are really comparing.

We assume that the reviewer referred to Fig 5H. In this experiment we overexpressed *Bnl*. This leads to clear defects, but terminal cells form and branching is observed under conditions of mild activation of the Gal4/UAS system (Fig 7F,

S3J). Under these conditions we detected an increased accumulation of Wgn, mostly in vesicles, as compared to the control situation (Fig 7E).

c. Line 333: Intracellular Wgn more in terminal cells - quantitation? Authors suggest degradation of Wgn based on more signal detected in the vesicular form inside. More signals could be due to more expression.

As suggested by the reviewer we have quantified the percentage of Wgn vesicles at the tip and at the base. This data can be now found in lines 417-420 and in a representative image in Fig 7C

In the text we are discussing the point that the reviewer is making and we suggest that regulated degradation likely contributes to Wgn pattern, although other mechanisms (like increased transcription) could also contribute. We have tried to make the point clearer (lines 420-431).

d. Line 338- what is shrub?

Shrub encodes Vps32/Snf7, a subunit of the ESCRT III complex. We now provide a more complete description of shrub in the text (lines 423-426).

e. Line 339: "using shrub-GFP 33,34. We found the presence of large vesicles containing TNFR-Wgn at both tip and proximal regions (Fig 5F), suggesting that in normal conditions TNFR

Wgn is differentially processed throughout the dorsal branch, and likely degraded faster at the proximal region, probably contributing to the TNFR-Wgn pattern." what is the basis of such an important conclusion about differential processing?

We apologise if we gave the impression that we were making an important conclusion from our results. As we state in the text we suggest a differential processing based on our results ".... suggesting that in normal conditions TNFR-Wgn is differentially processed throughout the dorsal branch, and likely degraded at the proximal region. Thus, our results suggest that regulated degradation likely contributes to TNFR-Wgn pattern, although other mechanisms may also contribute."

The basis for this suggestion is the following: in wild type conditions we detect increased accumulation of Wgn vesicles at the tips, and only a few vesicles in the stalk. shrub-GFP acts as a dominant negative and it has been shown that when expressed in the tracheal cells it interferes with endosome maturation towards lysosome degradation giving rise to swollen endosomes (Dong et al., 2014; Mathew et al., 2020). We find the presence of Wgn in these swollen endosomes in the tip but also in the stalk of dorsal branches. This result suggests that Wgn is also present in the stalk, but is normally degraded so we do not detect it. We have tried to clarify this point in the text (lines 417-431).

f. Feedback loop? The evidence is not strong enough to support the feedback loop.

Too many claims without showing convincing data and analyzing the roles of TNF/TNFRs in tracheal very carefully!

We show that wgn regulates Btl (Fig 342-451) and we show that Btl activity regulates Wgn accumulation (Fig 7E,F text lines 432-438), pointing to a feed-back loop.

6) Discussion:

353 "We evidence a model in which two different receptors, each acting by a different mechanism, regulate one physiological event." It would be best to first show a mechanism of Wgn, before claiming different mechanisms at the molecular level.

We would like to suggest that this point is a matter of styles. We would like to point that we propose a mechanism for Wgn which is based on phenotypic analysis, genetic experiments, cellular analysis and biochemical data.

358: "In addition, our cellular analysis shows that while TNFR-Wgn can localise at the membrane, in normal conditions it is constantly and rapidly internalised into intracellular vesicles." - There were no experiments showing dynamics or kinetics and many normal conditions in the paper.

We fully agree with the reviewer that because we have not performed kinetics experiments we cannot refer to dynamisms. We have addressed this point throughout the whole manuscript. However, as discussed in previous points, our results do indicate that Wgn is internalised.

394: MAPK signaling needs to be shown under wgn conditions. A direct measure of btl activation is more important than Btl-Bnl colocalization. Can Btl activation be non-canonical, induced by TNFR, without Bnl?

In the results section we analyse MAPK activity and signalling (using both dpERK accumulation and the expression of the target gene DSRF) in wgn conditions (Fig 1C-H, Fig 5A-C).

We have addressed experimentally the point that the reviewer indicates, and we have tested whether in the absence of Bnl, Btl can be activated depending on Wgn. Mutants for bnl show a phenotype of lack of tracheal migration, and no terminal cells are specified (Sutherland et al., 1996). In contrast, mutants for wgn produce extra terminal cells. Double mutants for wgn;bnl show the same phenotype as bnl mutants, lack of tracheal migration and terminal cells. This result indicates that in the absence of Bnl, the Btl receptor cannot signal, indicating that it cannot be modulated non-canonically by wgn.

We are showing in Fig 4 in this document a representative example of a double mutant

Line 404: nothing was shown about Egr binding to Wgn in the trachea. How about Grnd expression in the trachea?

These points have been experimentally addressed and extended in the revised manuscript and are discussed in previous responses to this reviewer's concerns

I think similarly, TNFR-Btl interaction if any needs to be carefully verified with proper experiments. Also, please remove all the claims of temporal dynamics which were not tested.

As indicated in previous responses, we agree with the reviewer that we have not addressed protein dynamics and we have removed these claims. The Wgn-Btl interaction has been experimentally addressed throughout this manuscript using genetic, cellular and biochemical analysis.

Reviewer 3:

Letizia and colleagues report an unconventional TNFR signaling in *Drosophila* tracheal development. *Drosophila* Egr-JNK signaling is conserved and plays various roles in immune response, apoptosis and cell growth. In stark contrast to this canonical TNF pathway, Wengen, a TNFR, is in a complex with Btl (FGFR), and repress the differentiation of the tracheal terminal cell. Mechanistically, Wengen undergoes constant internalization. It colocalizes with Btl in intracellular vesicles and regulates the intracellular transport and turnover of Btl. This study also reveals a feedback loop in which FGF/Bnl stabilizes Wgn protein.

The messages in this manuscript are noteworthy and informative. The conclusions are experimentally addressed. This manuscript would alert the field the involvement of TNFR in RTK-FGFR signaling-dependent tubulogenesis. It would be of help if authors could clarify some points below.

We are glad to see that the reviewer finds our work noteworthy and informative.

1 The phenotype of extra terminal branches and/or terminal cells is very striking. The mechanism in which Wgn interferes Btl is reasonable. There is also possibility although unlikely that Wgn alters FGF, Bnl. It would be even more convincing, if the authors could show that expression of bnl is unaltered, by bnl-lacZ, Bnl staining, or RT-PCR?

We have addressed the point that the Reviewer #3 (and also Reviewer #2) is making. We have used bnl-lacZ and we have not detected differences between the pattern in control embryos and in wgnKO mutants. RT-PCR analysis also indicated no significant differences in bnl transcriptional levels in control versus mutant wgnKO. These results, combined with the results where we show that downregulation of wgn (wgnRNAi) in tracheal cells reproduce the same effects in terminal cell differentiation than those observed in the wgnKO mutants, indicate that

Wgn is autonomously required in the tracheal cells to regulate terminal cell number. We have added these results in the text and figures (lines 294-298, Fig S2B-D)

2 Authors show that Wgn colocalizes with Rab4, Rab7 and Arl8 and that perturbation of Rab5 suppresses TNFR-Wgn-dependent phenotype. Wonder if depletion of other endosomal components (e.g. Rab4) generates similar phenotype?

We have performed the experiment that the reviewer proposes. We have found that the expression of Rab4DN in tracheal cells does not suppress the effect of Wgn overexpression. Thus, we detect a phenotype of lack of terminal cells, indicating that Wgn is still able to prevent terminal cell differentiation. Rab4 is involved in a fast recycling pathway to direct cargoes (including, presumably, Wgn) from the endosomes to the plasma membrane (Bhuin and Roy, 2014; Wandinger-Ness and Zerial, 2014). Thus, we looked at the pattern of accumulation of Wgn in Rab4DN conditions, and we found that Wgn still localised in vesicles, indicating that while Wgn trafficking may be modulated, its internalisation is not prevented under this condition. This correlates with the capacity of Wgn to prevent terminal cell differentiation. These results are in agreement with our hypothesis that Wgn activity depends on its ability to internalise. The results with Rab4DN are shown in Fig 5 in this document. We have not added them to the manuscript to avoid adding too much information. Nevertheless, we can add these results in case it is considered necessary.

Rab5 is involved in the traffic from the membrane to the early endosomes (Bhuin and Roy, 2014; Wandinger-Ness and Zerial, 2014). In contrast to Rab4DN, Rab5DN produce a clear change in the pattern of Wgn localisation, which now localises at the membrane. This correlates with a loss of activity of Wgn, which now cannot prevent terminal cell differentiation (page 8, Fig 2J). Again, these results are in agreement with our hypothesis that Wgn activity depends on its ability to internalise and traffic.

3 In addition to anti-Flag, is it possible to perform the co-IP experiment with anti-Wgn antibody to assay the interaction between Wgn and Btl at physiological level?

Unfortunately, these co-IP assays cannot be performed with endogenous proteins in the trachea from total embryo protein extracts because the levels are too low. In addition, the Wgn antibody that we used in this work was a gift from the K.Basler lab, and we do not have enough antibody to perform co-IP experiments. We tried to produce our own Wgn antibody but we did not succeed. Therefore, we performed co-IP assays in salivary glands expressing wgn-flag and btl-GFP since we can isolate the tissue and obtain sufficient purified protein extracts to perform biochemical experiments. To avoid strong overexpression with the Gal4/UAS system, crosses and larvae were maintained at 22 °C.

To gain further evidence of our hypothesis that Wgn and Btl participate in the formation of a complex, in this revised version we are providing data based on in situ PLA experiments (proximity ligation assay). We found the presence of PLA interactions in the trachea when we expressed Wgn-flag and Btl-GFP, compared to the control experiments (lines 325-329 and Fig 5J,K). In this case, we could not use

either the antibodies recognising the endogenous Wgn or Btl proteins, as we do not have sufficient Wgn antibody and our Btl antibody does not work for immunolabelling. Our PLA experiments suggest that the two proteins reside at a maximum distance of 40 nm, consistent with the hypothesis that they form a complex.

4 It has been reported that Awd (abnormal wing discs) regulates the endocytosis of Btl. Wonder if authors have genetic interaction between awd, wgn and btl.

As the reviewer points, awd, the *Drosophila* nm23 homolog, was shown to participate in the dynamin mediated endocytosis of Btl, regulating Btl levels and as a consequence tracheal migration (Dammai et al., 2003). Therefore, considering the model we propose, a possibility is that wgn and awd interact in some way, participating in a common pathway, for instance.

It has not been previously addressed, however, whether awd regulates DSRF expression, which is the Btl activity that we are investigating in this manuscript. Therefore, we first analysed whether awd loss and gain of function (using a null mutant for awd (*awd^{j2A4}*) and UAS-awd-HA, kindly provided by Dr. Cavaliere, University of Bologna) affected terminal cell number by assessing DSRF expression. *awd^{j2A4}* mutants did not show consistent defects in the number of terminal cells, although we detected migration defects as previously described (Dammai et al., 2003). The overexpression of awd did not produce defects in the number of terminal cells either. Altogether, these results indicate that even that Awd regulates Btl levels and activity (Dammai et al., 2003), it does not affect the ability of Btl receptor to signal to activate the terminal identity in the tracheal cells.

In spite of this lack of effect on terminal cell differentiation, we have performed different experiments to identify possible genetic interactions with Wgn. In particular, we performed the following experiments:

1) overexpression of awd in wgn mutants: to test whether increased awd rescued the extra terminal cell phenotype of wgn mutants. We did not find a rescue.

2) co-overexpression of awd and wgn: to test whether increased awd exacerbated the lack of terminal cells produced by overexpression of wgn. We did not find an increased effect.

3) double mutants *wgn;awd*: to test whether absence of awd exacerbated the extra terminal cell phenotype of wgn mutants. We did not find an increased effect.

4) overexpression of wgn in awd mutants: to test whether the absence of awd rescued the lack of terminal cells produced by overexpression of wgn. We did not find a rescue.

Altogether our experiments suggest that wgn and awd act in different parallel pathways to regulate the levels and activity of Btl.

The results with Awd can be found in Fig 6 in this document. We have not added them to the manuscript to avoid adding too much information. Nevertheless, we can add these results in case it is considered necessary.

5 The authors show that overexpression of Egr reduces Btl/Wgn vesicle. Because TNF/Egr triggers apoptosis in various tissues, it is good to know if this happens in tracheal system?

As the reviewer points, it has been shown that TNFR signalling and JNK pathway activation can promote cell death in different cellular contexts (Igaki et al., 2002; Moreno et al., 2002). Thus, as suggested by the reviewer, we have investigated whether this is also the case in the embryonic tracheal system.

In our initial submission we showed that the expression of a constitutively active form of hep in the tracheal cells (hepCA) induced the expression of puc-lacZ (a reporter of JNK activity), indicating that hepCA activates the pathway. In this revised version we now analysed cell death under these conditions. We detected massive cell death using Dcp1 as a marker and presence of many apoptotic cells. These results indicate that the activation of the JNK pathway in the trachea triggers apoptosis.

We expressed Wgn and Egr in the tracheal cells to investigate whether they also promote cell death. We have found that they do not activate puc-lacZ and cell death markers, and tracheal cells did not display signs of massive apoptosis. These results indicate that Wgn and Egr are not activating the JNK pathway in the trachea and that they do not promote cell death.

These results can now be found in the text and figures (pages 220-224, 257-260 and Fig 3I-L and Fig 4I,J)

6 In line 326-327, the authors claim that Wgn restricts the maintenance of Bnl/Btl complexes in terminal cells. They also show that Wgn is in a complex with Btl. Are those three together in a complex? Or Wgn competes for Btl, and thus prevents its activation by Bnl? Please clarify.

The reviewer raises an interesting point that we have tried to address by performing new co-IP experiments. We obtained extracts of salivary glands that express the three proteins, wgn-Flag, btl and bnl-HA, in two different buffers: RIPA buffer, a highly stringent buffer which minimises non-specific protein-protein interactions (as used in co-IP assays shown in Fig 5H,I and S2G,H, results not shown), and a less stringent buffer to preserve more labile interactions (see Fig 7 in this document). In both cases α Flag co-immunoprecipitated Btl but not Bnl-HA (Fig 7, left panels). This could reflect that the three proteins are not together in a complex but it could also indicate that the interaction between Btl and Bnl does not last long enough to be revealed by co-IP. To discriminate between the two possibilities, extracts of salivary glands expressing btl and bnl-HA were immunoprecipitated with α btl antibodies. No Bnl-HA was observed in the immunoprecipitated material (Fig 7, right panels). These results indicate that Btl-Bnl interaction cannot be analysed by co-IP. Therefore, we cannot answer the reviewer's question using co-IP experiments. Different

experimental approaches should be carried out to determine whether Wgn is in a complex with Btl-Bnl or competes for Btl preventing its activation by Bnl. Our results indicating a stabilisation of Wgn accumulation upon Bnl overexpression could suggest that Bnl/Btl/Wgn form a complex rather than Wgn and Bnl compete. Future experiments in the lab will try to address this particular question.

Minor points:

1 It would be informative to provide p value and/or statistical significances in Fig. 2B, since the authors claim that *egr* mutants and JNK downregulation gave low penetrant phenotype in line 169, 170.

In the revised version we provide the statistical analysis in Fig 3B (former Fig 2B), comparing the different conditions of JNK downregulation to the control and to the downregulation of *wgn* in the tracheal cells.

2 The terminal cell phenotypes of *wgn* mutants can be suppressed by expression of *btIRNAi* or *btIDN*?

As suggested by Reviewer #3 and also Reviewer #2 we have analysed whether decreasing *btl* levels/activity suppressed *wgn* loss of function defects, as it would be expected if *Wgn* regulates *Btl* activity.

We tested 3 different conditions to approach this question: *wgnKO*; *btIDN*; *wgnKO*; *btIRNAi* and *wgnKO*; *bnl⁰⁰⁸⁵⁷*

In our hands, the expression of *btIDN* in the tracheal system did not produce any detectable phenotype, in branching or in the specification of terminal cells, suggesting that this line is not sufficiently active in our embryonic experimental set-up. In agreement with this low activity, we found that *btIDN* was not able to suppress the effects of *wgnKO* mutants and extra terminal cells were detected.

In contrast, we found that the expression of 2 independent *btIRNAi* lines (namely BL-60013 *y v*; P(Trip.HMS05005) *attP40* and BL-40871 *y sc v sev*; P(Trip.HMS02038) *attP2*) in the tracheal system produced an extreme phenotype of lack of terminal cells, indicating that no sufficient levels of *btl* are present in these conditions to activate the pathway to form terminal cells. We found that the expression of *btIRNAi* reverted the phenotype of *wgnKO*. In this case, we compared and quantified the number of terminal cells in *wgnKO*; *btIGal4>lacZ* mutants and in their siblings *wgnKO*; *btIGal4>UASbtIRNAi*. The suppression of the phenotype was very clear and consistent.

bnl has been shown to be an haploinsufficient locus (Jarecki et al., 1999; Sutherland et al., 1996). We tested the effect of reducing 1 copy of *bnl* quantifying the presence of terminal cells in the dorsal branches in *bnl⁰⁰⁸⁵⁷* (an amorphic allele) heterozygotes. We found that most dorsal branches lacked the terminal cell. Thus, we then tested whether reducing the dosage of *bnl* reverted the *wgnKO* phenotype. We compared and quantified the number of terminal cells in *wgnKO*; *lacZ/+* mutants and in their siblings *wgnKO*; *bnl⁰⁰⁸⁵⁷/+*. The reversion of the phenotype was very clear and consistent. The reduction of *bnl* decreases the activity of the pathway because less

receptor is activated (as the ligand is limiting, (Sutherland et al., 1996)). Interestingly, in this case, we found that reducing the dose of *bnl* mostly reverted the *wgn* phenotype to a wild type number of terminal cells (1 terminal cell per DB), while reducing *Btl* levels by RNAi produced mostly absence of terminal cells. These results suggest dosage-sensitive interactions.

We have added the results of *btl*RNAi (line BL-60013) and *bnl*⁰⁰⁸⁵⁷ in the text and figures (lines 369-381 and Fig 6F-H and S3D-G)

We believe that these genetic interactions further strengthen our hypothesis that *Wgn* regulates the number of terminal cells through the regulation of *Btl* activity.

3 In line 258, the authors claim that *btl* is not expressed in salivary gland. This statement is not needed for this co-IP experiment. The expression of *btl*-Gal4 in salivary gland is observed.

We thank the reviewer to point this out. We have removed this statement.

4 It would be of help if authors could provide more insights regarding how the membrane receptor, *Wgn*, promotes the degradation of *Btl* in the discussion?

We have extended the discussion to add some insights into how *Wgn* could regulate *Btl* degradation (lines 473-479)

- Andersen, D.S., J. Colombani, V. Palmerini, K. Chakrabandhu, E. Boone, M. Rothlisberger, J. Toggweiler, K. Basler, M. Mapelli, A.O. Hueber, and P. Leopold. 2015. The *Drosophila* TNF receptor Grindelwald couples loss of cell polarity and neoplastic growth. *Nature*. 522:482-486.
- Baer, M.M., A. Bilstein, E. Caussin, A. Csiszar, M. Affolter, and M. Leptin. 2010. The role of apoptosis in shaping the tracheal system in the *Drosophila* embryo. *Mech Dev*. 127:28-35.
- Bhuin, T., and J.K. Roy. 2014. Rab proteins: the key regulators of intracellular vesicle transport. *Exp Cell Res*. 328:1-19.
- Dammai, V., B. Adryan, K.R. Lavenburg, and T. Hsu. 2003. *Drosophila* *awd*, the homolog of human *nm23*, regulates FGF receptor levels and functions synergistically with *shi*/dynamin during tracheal development. *Genes Dev*. 17:2812-2824.
- Dong, B., E. Hannezo, and S. Hayashi. 2014. Balance between apical membrane growth and luminal matrix resistance determines epithelial tubule shape. *Cell Rep*. 7:941-950.
- Fan, S.J., B. Kroeger, P.P. Marie, E.M. Bridges, J.D. Mason, K. McCormick, C.E. Zois, H. Sheldon, N. Khalid Alham, E. Johnson, M. Ellis, M.I. Stefana, C.C. Mendes, S.M. Wainwright, C. Cunningham, F.C. Hamdy, J.F. Morris, A.L. Harris, C. Wilson, and D.C. Goberdhan. 2020. Glutamine deprivation alters the origin and function of cancer cell exosomes. *EMBO J*. 39:e103009.
- Gabay, L., R. Seger, and B.Z. Shilo. 1997. MAP kinase in situ activation atlas during *Drosophila* embryogenesis. *Development*. 124:3535-3541.
- Gervais, L., and J. Casanova. 2011. The *Drosophila* homologue of SRF acts as a boosting mechanism to sustain FGF-induced terminal branching in the tracheal system. *Development*. 138:1269-1274.
- Guillemin, K., J. Groppe, K. Ducker, R. Treisman, E. Hafen, M. Affolter, and M.A. Krasnow. 1996. The pruned gene encodes the *Drosophila* serum response factor and regulates cytoplasmic outgrowth during terminal branching of the tracheal system. *Development*. 122:1353-1362.

- Hiramatsu, N., T. Tago, T. Satoh, and A.K. Satoh. 2019. ER membrane protein complex is required for the insertions of late-synthesized transmembrane helices of Rh1 in *Drosophila* photoreceptors. *Mol Biol Cell*. 30:2890-2900.
- Hsouna, A., G. Nallamotheu, N. Kose, M. Guinea, V. Dammai, and T. Hsu. 2010. *Drosophila* von Hippel-Lindau tumor suppressor gene function in epithelial tubule morphogenesis. *Mol Cell Biol*. 30:3779-3794.
- Igaki, T., H. Kanda, Y. Yamamoto-Goto, H. Kanuka, E. Kuranaga, T. Aigaki, and M. Miura. 2002. Eiger, a TNF superfamily ligand that triggers the *Drosophila* JNK pathway. *EMBO J*. 21:3009-3018.
- Jarecki, J., E. Johnson, and M.A. Krasnow. 1999. Oxygen regulation of airway branching in *Drosophila* is mediated by branchless FGF. *Cell*. 99:211-220.
- Kato, K., T. Chihara, and S. Hayashi. 2004. Hedgehog and Decapentaplegic instruct polarized growth of cell extensions in the *Drosophila* trachea. *Development*. 131:5253-5261.
- Kaupilla, S., W.S. Maaty, P. Chen, R.S. Tomar, M.T. Eby, J. Chapo, S. Chew, N. Rathore, S. Zachariah, S.K. Sinha, J.M. Abrams, and P.M. Chaudhary. 2003. Eiger and its receptor, Wengen, comprise a TNF-like system in *Drosophila*. *Oncogene*. 22:4860-4867.
- Kondo, T., and S. Hayashi. 2013. Mitotic cell rounding accelerates epithelial invagination. *Nature*. 494:125-129.
- Lee, T., N. Hacohen, M. Krasnow, and D.J. Montell. 1996. Regulated Breathless receptor tyrosine kinase activity required to pattern cell migration and branching in the *Drosophila* tracheal system. *Genes Dev*. 10:2912-2921.
- Mathew, R., L.D. Rios-Barrera, P. Machado, Y. Schwab, and M. Leptin. 2020. Transcytosis via the late endocytic pathway as a cell morphogenetic mechanism. *EMBO J*. 39:e105332.
- Moreno, E., M. Yan, and K. Basler. 2002. Evolution of TNF signaling mechanisms: JNK-dependent apoptosis triggered by Eiger, the *Drosophila* homolog of the TNF superfamily. *Curr Biol*. 12:1263-1268.
- Mortimer, N.T., and K.H. Moberg. 2009. Regulation of *Drosophila* embryonic tracheogenesis by dVHL and hypoxia. *Dev Biol*. 329:294-305.
- Nussbaumer, U., G. Halder, J. Groppe, M. Affolter, and J. Montagne. 2000. Expression of the blistered/DSRF gene is controlled by different morphogens during *Drosophila* trachea and wing development. *Mech Dev*. 96:27-36.
- Palmerini, V., S. Monzani, Q. Laurichesse, R. Loudhaief, S. Mari, V. Cecatiello, V. Olieric, S. Pasqualato, J. Colombani, D.S. Andersen, and M. Mapelli. 2021. *Drosophila* TNFRs Grindelwald and Wengen bind Eiger with different affinities and promote distinct cellular functions. *Nature communications*. 12:2070.
- Reichman-Fried, M., and B.Z. Shilo. 1995. Breathless, a *Drosophila* FGF receptor homolog, is required for the onset of tracheal cell migration and tracheole formation. *Mech Dev*. 52:265-273.
- Ruan, W., A. Srinivasan, S. Lin, I. Kara k, and P.A. Barker. 2016. Eiger-induced cell death relies on Rac1-dependent endocytosis. *Cell death & disease*. 7:e2181.
- Samakovlis, C., N. Hacohen, G. Manning, D.C. Sutherland, K. Guillemin, and M.A. Krasnow. 1996. Development of the *Drosophila* tracheal system occurs by a series of morphologically distinct but genetically coupled branching events. *Development*. 122:1395-1407.
- Sarabipour, S., and K. Hristova. 2016. Mechanism of FGF receptor dimerization and activation. *Nature communications*. 7:10262.
- Sato, M., and T.B. Kornberg. 2002. FGF is an essential mitogen and chemoattractant for the air sacs of the *drosophila* tracheal system. *Dev Cell*. 3:195-207.
- Steneberg, P., J. Hemphala, and C. Samakovlis. 1999. Dpp and Notch specify the fusion cell fate in the dorsal branches of the *Drosophila* trachea. *Mech Dev*. 87:153-163.
- Sutherland, D., C. Samakovlis, and M.A. Krasnow. 1996. branchless encodes a *Drosophila* FGF homolog that controls tracheal cell migration and the pattern of branching. *Cell*. 87:1091-1101.
- Wandinger-Ness, A., and M. Zerial. 2014. Rab proteins and the compartmentalization of the endosomal system. *Cold Spring Harb Perspect Biol*. 6:a022616.
- Wang, Q., M. Uhlirva, and D. Bohmann. 2010. Spatial restriction of FGF signaling by a matrix metalloprotease controls branching morphogenesis. *Dev Cell*. 18:157-164.
- Xue, L., T. Igaki, E. Kuranaga, H. Kanda, M. Miura, and T. Xu. 2007. Tumor suppressor CYLD regulates JNK-induced cell death in *Drosophila*. *Dev Cell*. 13:446-454.
- Yan, D., and X. Lin. 2007. *Drosophila* glypican Dally-like acts in FGF-receiving cells to modulate FGF signaling during tracheal morphogenesis. *Dev Biol*. 312:203-216.

Coomassie blue staining of total protein extracts (left panel) and the soluble fraction (right panel) from *E. Coli* cells expressing either GST or GST-btl fusion proteins containing the transmembrane and/or the intracellular domains of btl (TM+IC and IC respectively). * indicate the recombinant proteins

Figure 1

Lateral views of *btI^{GFP}* stage 14 embryos stained with Grnd (magenta and grey) and with GFP to visualise the tracheal cells. Note the accumulation of Grnd in the hindgut (yellow arrows) and in lateral chordotonal organs (blue arrows). Grnd is not clearly detected in the tracheal cells (A), however, sometimes we detect a faint apical accumulation (red arrow in B and B'). B' corresponds to the inset in B. Scale bar A,B 20 μ m; B' 10 μ m

Lateral views of representative embryos expressing *grnd^{extra}* or full length *grnd* in the tracheal system. Embryos were stained with DSRF to visualise the terminal cells and with CBP to visualise the tracheal pattern. Note the presence of an extra terminal cell in *grnd^{extra}* conditions. Scale bar 20 μ m

Quantification of the percentage of dorsal branches that show the indicated phenotypes (presence of 2 terminal cells or lack of terminal cell) in the indicated genotypes. The expression of *grnd^{extra}* in the trachea leads to a weak phenotype of extra terminal cells compared to the control. Control embryos correspond to *btIGal4>srcGFP*. Number of branches analysed: *btIGal4>srcGFP*, 799 branches (from 71 embryos), *btI>grnd^{extra},srcGFP*, 306 branches (from 41 embryos), *btI>grnd,srcGFP*, 216 branches (from 27 embryos). The Error bars indicate standard deviation (s. d.). * $p < 0.05$, ns not significant, non-parametric Mann-Whitney test.

Figure 2

DE-Cadherin/Flag

Wgn/Trachea/Salivary gland

Images show salivary glands. The overexpression of Wgn in salivary glands (A) does not produce a detectable phenotype. Low levels of Wgn are detected in salivary glands (B). Note the pattern of Wgn in vesicles also in the salivary glands (white arrows)
Scale bars 10 μ m

Figure 3

DSRF/trachea

Lateral view of a stage 14/15 stained with DSRF to visualise the terminal cells (green) and to visualise the tracheal system (magenta). Note that in these double mutants no tracheal cells express DSRF

Figure 4

Lateral view of a stage 14 *btlGal4>UASwgn-flag; UASRab4^{DN}-YFP* embryo stained with Flag (grey) to visualise Wgn, DSRF (magenta and grey) to visualise the terminal cells and with GFP to visualise the tracheal cells (green). Note the accumulation of Wgn in vesicles and the lack of terminal cells (yellow arrow points to the only DSRF expressing cells in the dorsal branches).
Scale bar 20 μ m

Quantification of the percentage of dorsal branches that lack the terminal cell in the indicated genotypes. Note that compromising Rab4 activity does not suppress the effect of wgn overexpression. n indicates the number of branches analysed, and in brackets the number of embryos. The Error bars indicate standard deviation (s. d.). ns not significant, unpaired t test with Welch's correction.

Figure 5

awd mutants do not show defects in terminal cell number

awd overexpression does not produce defects in terminal cell number

■ Extra terminal cells

The overexpression of awd does not revert the phenotype of extra terminal cells of wgn mutants

■ Lack of terminal cell

The co-overexpression of awd does not exacerbate the phenotype of lack of terminal cells of wgn overexpression

■ Extra terminal cells

The loss of awd activity does not exacerbate the phenotype of extra terminal cells of wgn mutants

■ Lack of terminal cell

The loss of awd activity does not revert the phenotype of lack of terminal cells of wgn overexpression

Images show lateral views of a stage 14/15 stained with DSRF to visualise the terminal cells (green) and to visualise the tracheal cells (magenta).

Scale bars 20 μm

Quantifications show the percentage of dorsal branches that show the indicated phenotype in the indicated genotypes. n indicates the number of branches analysed, and in brackets the number of embryos. The Error bars indicate standard deviation (s. d.). ns not significant, ****p < 0.0001.

Figure 6

Western blot using α HA (upper panels), α Btl (middle panels) and α Flag (lower panel) of salivary gland extracts from third-instar larvae that express either TNFR-wgn-flag, FGFR-btl and bnl-GFP-HA (left panels) or FGFR-btl and bnl-GFP-HA (right panels) that were immunoprecipitated with antibodies indicated on top of the Figure. Input correspond to 10% of the immunoprecipitated material.

Figure 7

REVIEWER COMMENTS

Reviewer #1 (Remarks to the Author):

The presented revision of the article by Letizia et al. is clearly improved and a number of different ambiguities have been resolved. The central message of the article, which is that there is a physical Wengen/activated Btl interaction that prevents appropriate btl signaling downstream of Btl and thus interferes with terminal cell genesis, is still elusive.

The new results suggest that in the terminal cell system TNF/TNFR signaling does not play a role and that Wengen acts exclusively as an interaction partner for activated Btl.

I still have some ambiguities that remain.

In Fig. 5, pERK staining is shown as a marker of FGF/FGFR activity. In B, pERK signals are seen not only in tracheal cells but also in many other cells - but not in the WT context and in Wg overexpression. Surely this should be the case in exactly the opposite way.

The evidence for a direct interaction between wgn and Btl is still weak, as is the hypothesis that increased degradation of activated Btl occurs.

Reviewer #2 (Remarks to the Author):

The revised manuscript is significantly improved and has addressed most of my major concerns. The non-canonical activity of Wgn regulating FGFR levels during tracheal terminal branching is a significant finding, presenting a new crosstalk mechanism between TNFR and FGFR pathways.

I have a few additional points for the revised manuscript:

1) Line 164: if the Wgn antibody binds to the extracellular portion of Wgn, the non-permeabilized immunostaining (similar to extracellular Dpp, reported before) might provide improved detection of cell surface Wgn levels. This would strengthen the point presented here.

Line 175: Please clarify how Hrs localization is detected. Does Fyve:GFP (Fig2E) a reporter for Hrs? While the overall intensity and number of puncta increased in the mutant conditions, it is unclear from the images whether there is any change between the membrane vs. intracellular distributions with and without endocytic uptake. Zoomed-in images might help to understand. If possible, comparing the ratio of surface to intra-cellular levels between the control and test might clearly show the required phenotypes and strengthen this point.

2) Line 186: Fig. 2J: Clearly, DSRF is present in the Wgn overexpressing branch shown, but does this phenotype differ from Fig 1N, where DSRF was found in some of the branches without affecting endocytosis? The no. of DSRF-positive branches relative to total branches, with and without endocytosis, under the Wgn gain-of-function condition might strengthen the point.

3) Line 191: the Rab5 downregulation phenotypes support that the Wgn internalization is required for terminal branch-inducing activities. Similarly, the change in membrane localization under the conditions might suggest that Wgn internalization limits its surface distribution. However, this line gives an impression of showing constitutive internalization of molecules. Rewording some of these sentences might be better.

4) Line 325-328: Additional PLA experiment supported the model of Wgn-Btl interactions. It is important to present a brief idea of the PLA experiment (without going into the method), especially explaining the two antibodies selected for the intracellular interactions between the chimeric proteins and why <40 nm proximity is suggested. Secondly, why in Fig 5K, the dorsal trunk is shown rather than the dorsal branch can be explained. Please indicate the tracheal outlines in Fig. K, L, M for better

visibility.

It would be wonderful to do this assay with the native Btl and Wgn without overexpressing them in the trachea, but suitable reagents might be unavailable.

5) Line 297: There is no doubt that Wgn has an autonomous role in the trachea, but Fig. S2D-E also indicates a change in bnl-LacZ. The highly expressing cells at the tracheal tips are reduced; instead, more bnl cells appear at the lateral sides of the dorsal branch. This might not have anything to do with Wgn, but the possibility of a non-autonomous activity cannot be completely ruled out.

6) Line 105: The prediction of the signaling gradient might be irrelevant. Wgn modulation changed the number of terminal branch cells containing more Btl/Bnl puncta and nuclear dpERK. However, the gradient of Btl signaling activity (i.e., dpERK) is hard to be judged from the images. Rewording might help.

Reviewer #3 (Remarks to the Author):

The authors have addressed all the questions. The authors performed substantial experiments in response to reviewers' questions, especially those common ones. I am satisfied with the responses and improvements and support its publication.

POINT BY POINT RESPONSE TO REVIEWER'S COMMENTS

Reviewer #1 (Remarks to the Author):

The presented revision of the article by Letizia et al. is clearly improved and a number of different ambiguities have been resolved. The central message of the article, which is that there is a physical Wengen/activated Btl interaction that prevents appropriate btl signaling downstream of Btl and thus interferes with terminal cell genesis, is still elusive.

The new results suggest that in the terminal cell system TNF/TNFR signaling does not play a role and that Wengen acts exclusively as an interaction partner for activated Btl.

I still have some ambiguities that remain.

In Fig. 5, pERK staining is shown as a marker of FGF/FGFR activity. In B, pERK signals are seen not only in tracheal cells but also in many other cells - but not in the WT context and in Wg overexpression. Surely this should be the case in exactly the opposite way.

To visualise pERK we used an antibody that recognises a dual phosphorylated form of ERK-MAPK (dpERK). The antibody reveals the pattern of phosphorylated-ERK dependent on the activation of different Receptor Tyrosine Kinases (RTKs) in different tissues (Gabay et al. 1997. Dev 124:3535). Thus, the staining reflects the activity of Btl but also from other RTKs in tissues other than the trachea. In addition, as published, the antibody shows a very dynamic pattern that reflects RTKs' activity. The antibody is a tricky one and it is often difficult to obtain good stainings, and there is a certain variability in the background-signal ratio in different experiments. The reviewer points that there is more signal outside the trachea in *wgn* mutant conditions than in control or *btl>UASwgn* conditions in the images shown in Fig 5. However, we think this is likely due to certain variability in the stainings and to the dynamic nature of pERK activation. We are showing below panels with examples of dpERK staining in the different conditions where expression outside the trachea can be observed in control and *btl>wgn* conditions, or where no clear expression of dpERK is observed outside the trachea in *wgn* mutants.

Images show examples of control, UAS-wgn and wgnKO mutants stained for dpERK (red), DSRF to mark the terminal cells (green) and tracheal branches (blue). Note the pattern of dpERK outside the tracheal system in the control and UAS-wgn examples (white arrows), and low levels of dpERK signal outside the tracheal system in wgnKO mutants

However, we cannot discard a role for Wgn in regulating the activity of other RTKs. Actually, this is one of the lines of investigation that we are currently following, trying to determine whether Wgn can act in a comparable manner to that with Btl to regulate the activity of other RTKs. If that was the case, we could expect an increase of dpERK accumulation in different tissues in wgn mutant conditions. Nevertheless, even in the case that Wgn regulated other RTKs-dependent pERK accumulation, this should not contribute to the effects we describe on tracheal terminal cell differentiation, as we show that Wgn is required in the tracheal cells to control the number of terminal cells.

The evidence for a direct interaction between wgn and Btl is still weak, as is the hypothesis that increased degradation of activated Btl occurs.

We would like point that, while in our experiments we did not address a direct interaction between Btl and Wgn, the co-IP and PLA experiments provide strong evidence indicating that they form a complex. On the other hand, the phenotypic analysis of wgn mutant conditions indicates the regulation of Btl activity, and the genetic experiments further support that Wgn acts by regulating Btl activity. Altogether, we reveal an unconventional function of a TNFR in regulating the activity of a FGFR. Both TNFRs and FGFRs are involved in a wide variety of developmental and homeostatic events, and their misregulation leads to fatal diseases. Thus, our work provides significant new insights into the understanding of the function and activation mechanisms of these families of important receptors. Future work in the lab will focus to disentangle the exact mechanism by which Wgn may be regulating Btl trafficking and degradation.

Reviewer #2 (Remarks to the Author):

The revised manuscript is significantly improved and has addressed most of my major concerns. The non-canonical activity of Wgn regulating FGFR levels during tracheal terminal branching is a significant finding, presenting a new crosstalk mechanism between TNFR and FGFR pathways.

We are happy to see that the reviewer finds our work a significant finding.

I have a few additional points for the revised manuscript:

1) Line 164: if the Wgn antibody binds to the extracellular portion of Wgn, the non-permeabilized immunostaining (similar to extracellular Dpp, reported before) might provide improved detection of cell surface Wgn levels. This would strengthen the point presented here.

The experiment the reviewer proposes, based on non-permeabilised immunostaining, would be very useful to detect possible cell surface Wgn levels, but unfortunately it is not feasible in our case. The trachea is an internal structure in the embryo and therefore it is not exposed at the surface for such evaluation.

Line 175: Please clarify how Hrs localization is detected. Does Fyve:GFP (Fig2E) a reporter for Hrs? While the overall intensity and number of puncta increased in the mutant conditions, it is unclear from the images whether there is any change between the membrane vs. intracellular distributions with and without endocytic uptake. Zoomed-in images might help to understand. If possible, comparing the ratio of surface to intra-cellular levels between the control and test might clearly show the required phenotypes and strengthen this point.

Fyve-GFP identifies the Fyve domain of the ESCRT-0 component Hrs, thus representing a reporter for Hrs. We apologise for not mentioning this in the manuscript. We have added this information in the figure legend (Fig legend 2, line 1013).

We find that compromising the endocytic uptake leads to a change in the localisation of Wgn, as can be observed comparing Fig 2C,D with 2J and is described in the text (lines 181-187). As suggested by the reviewer, to help to visualise this we are now adding zoom-in images in the revised Figure 2. In addition, as suggested by the reviewer, we have analysed the ratio of Wgn accumulation in the basal membrane and in the cytoplasm in conditions of Wgn overexpression and in conditions of Wgn overexpression in which the endocytic uptake is compromised. In particular, we have analysed 15 dorsal branches from 5 *btGal4>UASWgn-flag* embryos and 15 dorsal branches from 6 *btGal4>UASWgn-flag; UASRab5^{DN}* embryos. We have measured the levels in the basal membrane of the branches (as an average of 3 different measurements in each branch) by manually drawing a 5-pixel line that follows the membrane, and the levels in the cytoplasm by manually drawing the region (as an average of 3 different measurements in each branch). We have obtained the ratio of cytoplasm/membrane for each branch. The results are shown below, as well as an example of the branches used. The results confirm our observation

Images show 1 dorsal branch of the indicated genotypes stained with Flag to visualise Wgn and in green to visualise the tracheal cells. Note the accumulation of Wgn in the membrane when endocytic uptake is compromised. Scatter plot quantifying the ratio of the levels of Wgn in membrane/cytoplasm in the different conditions. Mean \pm SD is shown. **** $p < 0.0001$. Two-tailed unpaired t test with Welch's correction

2) Line 186: Fig. 2J: Clearly, DSRF is present in the Wgn overexpressing branch shown, but does this phenotype differ from Fig 1N, where DSRF was found in some of the branches without affecting endocytosis? The no. of DSRF-positive branches relative to total branches, with and without endocytosis, under the Wgn gain-of-function condition might strengthen the point.

As indicated in Fig 1J, in conditions of Wgn overexpression a large percentage of dorsal branches lack the terminal cell (77,4% of DB have 0 terminal cells, n=736 dorsal branches from 82 embryos). In contrast, in conditions of Wgn overexpression and Rab5 downregulation, exemplified in Fig 2J, none of the dorsal branches show absence of terminal cell (0% of DB have 0 terminal cells, n=62 dorsal branches from 8 embryos). We have added a clarification in the text (line 191-192)

3) Line 191: the Rab5 downregulation phenotypes support that the Wgn internalization is required for terminal branch-inducing activities. Similarly, the change in membrane localization under the conditions might suggest that Wgn

internalization limits its surface distribution. However, this line gives an impression of showing constitutive internalization of molecules. Rewording some of these sentences might be better.

We have revised the paragraph and changed the text as suggested by the reviewer (lines 183, 195)

4) Line 325-328: Additional PLA experiment supported the model of Wgn-Btl interactions. It is important to present a brief idea of the PLA experiment (without going into the method), especially explaining the two antibodies selected for the intracellular interactions between the chimeric proteins and why <40 nm proximity is suggested.

We have revised the section describing the PLA experiments in the results sections providing more information (lines 335-343). The PLA signal is produced if the 2 proteins are closer than 40 nm according to the Duolink In Situ Red Starter Kit that we used.

Secondly, why in Fig 5K, the dorsal trunk is shown rather than the dorsal branch can be explained. Please indicate the tracheal outlines in Fig. K, L, M for better visibility.

To perform the PLA experiment we have used dissected larval trachea, for which we have adapted the protocol in the lab, as we cannot use whole mount embryos due to problems with permeabilisation/penetration of reagents. We obtain the dorsal trunks attached to the cuticle during the process of dissection, and typically the dorsal branches (which are finer and more fragile branches) are broken and lost. The tracheal outlines have been added in the Figure

It would be wonderful to do this assay with the native Btl and Wgn without overexpressing them in the trachea, but suitable reagents might be unavailable.

We agree with the reviewer that it would be wonderful to perform the PLA experiment with the native Btl and Wgn, however, we cannot perform this experiment. On the one hand, the Wgn antibody that we used in this work was a gift from the K.Basler lab, and we do not have enough antibody to perform the PLA experiments. We tried to produce our own Wgn antibody but we did not succeed. On the other hand, we have a Btl antibody generated in Dr. Casanova's lab that works well for WB but is not good enough to detect the endogenous Btl in immunostainings (although it can detect overexpressed Btl) (Lebreton and Casanova, 2016. Dev Dyn 245; 372). For this reason we performed the PLA experiments in trachea that express wgn-flag and btl-GFP.

5) Line 297: There is no doubt that Wgn has an autonomous role in the trachea, but Fig. S2D-E also indicates a change in bnl-LacZ. The highly expressing cells at the

tracheal tips are reduced; instead, more *bnl* cells appear at the lateral sides of the dorsal branch. This might not have anything to do with *Wgn*, but the possibility of a non-autonomous activity cannot be completely ruled out.

In FigS2B,C we used a *bnl-lacZ* reporter (*bnl*⁰⁶⁹¹⁶). Unfortunately in our hands this reporter is not very strong, which makes it difficult to compare in different conditions. To further test the pattern of *bnl* in control and *wgn* mutants we decided to use the *bnl-lexA/lexO-CAAXmcherry* system, kindly provided by S. Roy lab, that nicely and strongly reports the pattern of *bnl* (Du et al. 2017. *Dev Biol* 427 (1): 35). We could not detect clear differences between the control and *wgn*KO mutants in *bnl* expression pattern. To better document this observation we have changed the images in Fig S2B,C, now showing representative examples using the *bnl-lexA/lexO-CAAXmcherry* system of embryos at stage of terminal cell differentiation. In addition, we show below a panel of the pattern of *bnl* expression at stage 13, right before terminal cell specification, showing no clear differences between control and mutant. Nevertheless, *bnl* expression is very dynamic and complex (Du et al. 2017. *Dev Biol* 427 (1): 35), and we cannot completely discard minor effects of *wgn* on *bnl* pattern. We are now discussing this in the text (lines 301-306)

Images show stage 13 embryos labeled to show the pattern of *bnl*. Note the comparable expression pattern. *wgn* mutant embryo is slightly tilted dorsally compared to control, and for this reason the ventral pattern (orange arrow) is less visible

6) Line 105: The prediction of the signaling gradient might be irrelevant. *Wgn* modulation changed the number of terminal branch cells containing more *Btl*/*Bnl* puncta and nuclear dpERK. However, the gradient of *Btl* signaling activity (i.e., dpERK) is hard to be judged from the images. Rewording might help.

We agree with the reviewer in this point and we have reworded the sentence (line 105)

Reviewer #3 (Remarks to the Author):

The authors have addressed all the questions. The authors performed substantial experiments in response to reviewers' questions, especially those common ones. I am satisfied with the responses and improvements and support its publication.

We wish to thank the reviewer for the support

REVIEWERS' COMMENTS

Reviewer #1 (Remarks to the Author):

The authors answered all relevant questions appropriately. I am now satisfied with the current version of the manuscript.

Reviewer #2 (Remarks to the Author):

The authors have addressed my concerns/suggestions.

POINT BY POINT RESPONSE TO REVIEWER'S COMMENTS

Reviewer #1 (Remarks to the Author):

The authors answered all relevant questions appropriately. I am now satisfied with the current version of the manuscript.

We wish to thank the reviewer for the constructive criticisms

Reviewer #2 (Remarks to the Author):

The authors have addressed my concerns/suggestions.

We wish to thank the reviewer for the constructive criticisms